# Single-cell transcriptome profiles of *Drosophila fruitless*-expressing neurons from both sexes

**Colleen M Palmateer[1], Catherina Artikis[1], Savannah G Brovero[1], Benjamin Friedman[1], Alexis Gresham[1], Michelle N Arbeitman[1,2]***

[1]Department of Biomedical Sciences, Florida State University, College of Medicine, Tallahassee, United States; [2]Program of Neuroscience, Florida State University, Tallahassee, United States

**Abstract** *Drosophila melanogaster* reproductive behaviors are orchestrated by *fruitless* neurons. We performed single-cell RNA-sequencing on pupal neurons that produce sex-specifically spliced *fru* transcripts, the *fru P1-expressing* neurons. Uniform Manifold Approximation and Projection (UMAP) with clustering generates an atlas containing 113 clusters. While the male and female neurons overlap in UMAP space, more than half the clusters have sex differences in neuron number, and nearly all clusters display sex-differential expression. Based on an examination of enriched marker genes, we annotate clusters as circadian clock neurons, mushroom body Kenyon cell neurons, neurotransmitter- and/or neuropeptide-producing, and those that express *doublesex*. Marker gene analyses also show that genes that encode members of the immunoglobulin superfamily of cell adhesion molecules, transcription factors, neuropeptides, neuropeptide receptors, and Wnts have unique patterns of enriched expression across the clusters. In vivo spatial gene expression links to the clusters are examined. A functional analysis of *fru P1* circadian neurons shows they have dimorphic roles in activity and period length. Given that most clusters are comprised of male and female neurons indicates that the sexes have *fru P1* neurons with common gene expression programs. Sex-specific expression is overlaid on this program, to build the potential for vastly different sex-specific behaviors.

**\*For correspondence:**
michelle.arbeitman@med.fsu.edu

**Competing interest:** The authors declare that no competing interests exist.

## Editor's evaluation

This study presents a valuable single-cell sequencing dataset of fruitless-expressing neurons in the male and female *Drosophila* nervous system. The quality data and convincing analyses allowed the authors to conclude that most neuronal types are present in both *Drosophila* sexes, suggesting that sex-specific versions of the transcription factor Fruitless can modify neural function in a sex-specific way without completely altering core neural identity. This work will be of interest to developmental biologists and neuroscientists with a focus on sex-specific differences.

## Introduction

A current goal of neuroscience research is to understand molecular differences at the single cell/neuron level to better understand the diverse cellular components of the nervous system, with the ultimate goal of understanding how diverse cells work together to generate behavior (*Ngai, 2022*). *Drosophila* is an excellent model for this approach, given there are defined and experimentally tractable sets of neurons that generate the potential for complex behaviors. Indeed, a *Drosophila* single-cell RNA-seq (scRNA-seq) atlas has been generated to understand neurons underlying adult circadian

biology, with the characterization of the adult core circadian clock neurons (*Ma et al., 2021*). In addition, there are atlases that have been generated to understand the cellular components of the brain and ventral nerve cord (VNC) during development and adult stages, using single-cell, genome-wide approaches (*Allen et al., 2020*; *Croset et al., 2018*; *Davie et al., 2018*; *Konstantinides et al., 2018*; *Kurmangaliyev et al., 2020*; *Li et al., 2017*; *Li et al., 2022*; *McLaughlin et al., 2021*; *Özel et al., 2021*; *Simon and Konstantinides, 2021*; *Xie et al., 2021*). Here, we present a scRNA-seq study to understand how the potential for sexually dimorphic adult reproductive behaviors are specified in the developing nervous system in males and females. We gain insight into how neurons that arise from a sex-shared developmental trajectory can underlie vastly different behaviors in males and females (*Ren et al., 2016*).

In *Drosophila*, the neuronal substrates that direct reproductive behaviors are specified by the sex-specific transcription factors (TFs) encoded by *fruitless* (*fru; fru P1* transcript isoforms) and *doublesex* (*dsx*) (*Figure 1A*), produced as an outcome of alternative pre-mRNA splicing by the sex determination hierarchy (reviewed in *Andrew et al., 2019*; *Cline and Meyer, 1996*). Sex-specific splicing at the 5' end of *fru* transcripts produced under control of the most distal promoter (*P1* promoter; *fru P1* transcripts) results in the production of male-specific (Fru$^M$) isoforms. Fru$^M$ isoforms have an additional 101 amino acid region on the amino terminus that is not present in common Fru isoforms. In females, *fru P1* transcripts encode for a short peptide that is predicted to be nonfunctional. *fru* transcript isoforms are also alternatively spliced at the 3' end, resulting in products with different DNA-binding domains (*Gramates et al., 2022*; *Ito et al., 1996*; *Ryner et al., 1996*). The identification of master regulatory TFs has provided an unprecedented molecular inroad into a behavioral question, allowing for high-resolution genomic and genetic interrogation, microscopic visualization, and the physiological manipulation of neurons directing behavior. *dsx* and *fru P1*-expressing neurons (*fru P1* neurons hereafter) are present in both sexes and each set arises from sex-shared developmental lineages (*Ito et al., 1996*; *Lee et al., 2002*; *Manoli et al., 2005*; *Ren et al., 2016*; *Robinett et al., 2010*; *Ryner et al., 1996*; *Sanders and Arbeitman, 2008*; *Stockinger et al., 2005*). However, these neurons direct dramatically different innate behaviors in the sexes due to differences in morphology, connectivity, physiology, and number. Males display an intricate courtship display that includes chasing the female, tapping her with his leg, and singing a song by wing vibration. The female will either accept or reject the male's courtship advances. Once the female has mated, she shows a broad range of post-mating changes including changes in her receptivity to subsequent courtship displays (reviewed in *Anholt et al., 2020*; *Auer and Benton, 2016*; *Dauwalder, 2011*; *Greenspan and Ferveur, 2000*; *Manoli et al., 2006*; *Peng et al., 2021*; *Villella and Hall, 2008*; *Yamamoto et al., 2014*).

Here, we focus on *fru P1* neurons that are found in both males and females. Earlier studies of *fru P1* neurons showed that there are not large sex differences in their number or cell body positions (*Manoli et al., 2005*; *Stockinger et al., 2005*). *fru P1* was initially thought to be important for male courtship based on analyses of different *fru P1* mutant allele combinations (*Anand et al., 2001*; *Gailey and Hall, 1988*; *Ryner et al., 1996*; *Villella et al., 1997*). Additional studies showed that *fru P1* neurons are both necessary and largely sufficient for male courtship behaviors, based on experiments where *fru P1* neurons were either activated or silenced and behavior was examined, and by studies where Fru$^M$ was produced in females in *fru P1* neurons (*Clyne and Miesenböck, 2008*; *Demir and Dickson, 2005*; *Manoli et al., 2005*; *Stockinger et al., 2005*). Female receptivity was later shown to be directed by *fru P1* neurons using neuronal silencing approaches (*Kvitsiani and Dickson, 2006*). *fru P1* neurons make up ~2–5% of all neurons, in both sexes, with expression found in the periphery, brain, and VNC (*Lee et al., 2000*; *Manoli et al., 2005*; *Stockinger et al., 2005*). *fru P1* neurons in peripheral structures are important for detecting con-specific mates. In the central nervous system (CNS), *fru P1* is in regions important for higher-order processing of experience/sensation and regions involved in motor outputs for reproductive behaviors (reviewed in *Auer and Benton, 2016*).

The spatial positions of *fru P1* neurons in the brain and VNC have been assigned to named spatial clusters, with the idea that neurons in close proximity may share functions (*Billeter and Goodwin, 2004*; *Lee et al., 2000*; *Manoli et al., 2005*; *Stockinger et al., 2005*). However, it is not clear if all neurons within a spatial cluster are functionally and/or molecularly similar, which we address here. Furthermore, while the position and number of neurons are similar between the sexes, there is dimorphism in cell number within several spatial clusters, male-specific clusters, sex differences in neuronal projections, and differences in their physiology (reviewed in *Auer and Benton, 2016*; *Billeter and*

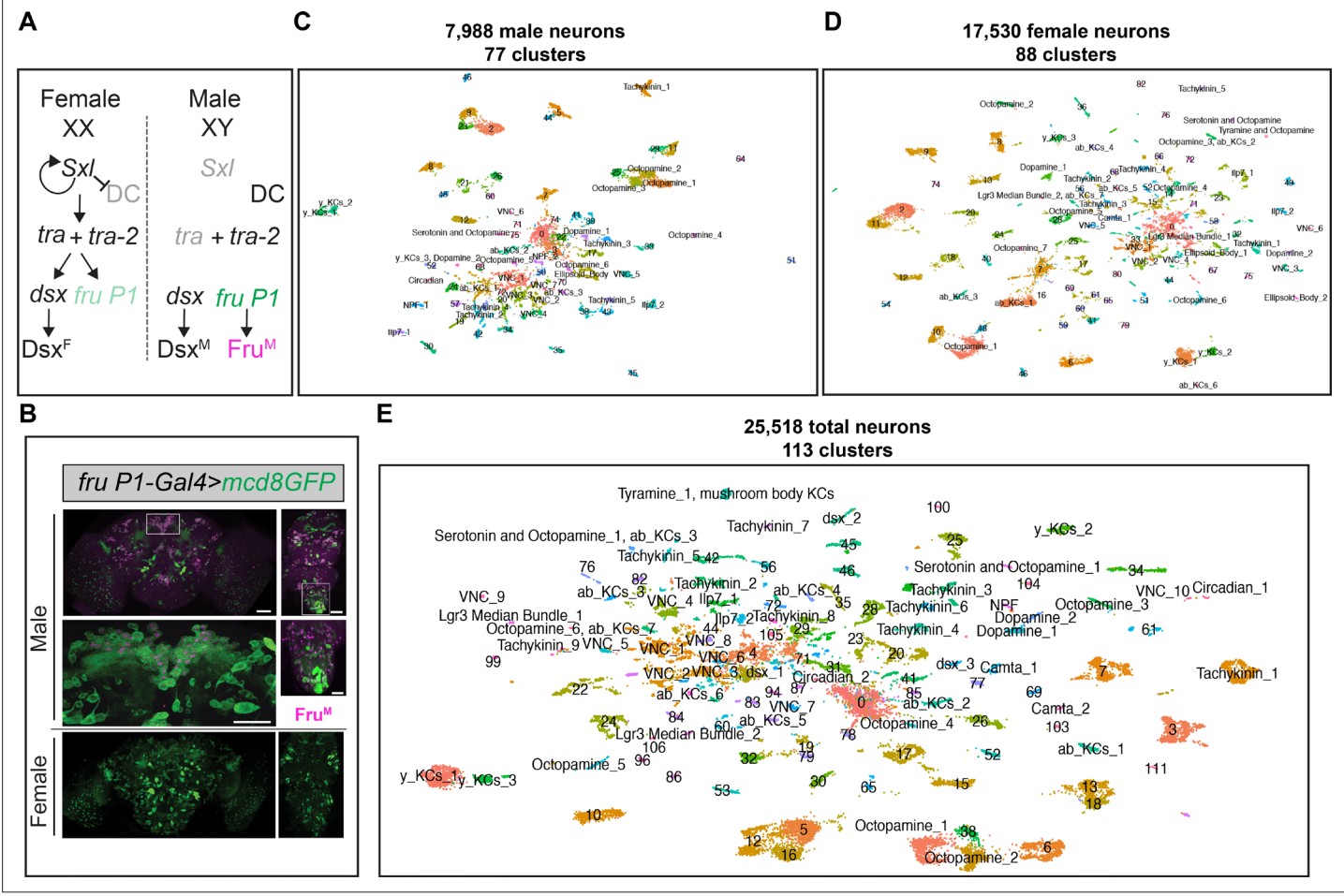

**Figure 1.** scRNA-sequencing of *fru P1* neurons from males and females at 48-hr after puparium formation (APF). (**A**) The *Drosophila* sex determination hierarchy is an alternative pre-mRNA splicing cascade that generates sex-specific transcription factors that regulate somatic sexual differentiation. The pre-mRNA splicing regulators are encoded by *sxl*, *tra*, and *tra-2*. Early production of Sxl in females results in continued production of functional Sxl and Tra in females. Tra and Tra-2 regulate the sex-specific splicing of *doublesex* (*dsx*) and *fruitless* transcripts from the *P1* promoter (*fru P1*). In females, female-specific Dsx (Dsx$^F$) is produced. In males, male-specific (Dsx$^M$) and male-specific Fru (Fru$^M$) are produced due to default pre-mRNA splicing. Sxl also regulates dosage compensation (DC) (reviewed in ***Andrew et al., 2019***; ***Cline and Meyer, 1996***). (**B**) Confocal maximum projections of 48-hr APF male (top) and female (bottom) brains and ventral nerve cords (VNCs) expressing membrane-bound GFP (*mcd8::GFP*) in *fru P1*-expressing neurons (green). Male tissues are immunostained for male-specific Fru$^M$ (magenta). Magnification of the boxed regions in male brain and VNC are below the male tissues (×40). Scale bars = 50 µm. (**C**) Uniform Manifold Approximation and Projection (UMAP) plot of 7988 *fru P1* cells, from male central nervous system (CNS) at 48-hr APF, grouped into 77 clusters (male data analysis). (**D**) UMAP plot of 17,530 *fru P1* cells, from female CNS at 48-hr APF, grouped into 88 clusters (female data analysis). (**E**) UMAP plot of 25,518 total *fru P1* neurons, from both sexes (full data set), grouped into 113 clusters. For all UMAP plots, the annotations shown were determined using marker gene expression (***Source data 3***). Clusters with numbers have additional annotations listed in ***Source data 3***.

The online version of this article includes the following figure supplement(s) for figure 1:

**Figure supplement 1.** Data filtering and replicate overlap in Uniform Manifold Approximation and Projection (UMAP) space.

**Figure supplement 2.** Jackstraw plots used to determine number of significant principal components (PCs) to use for dimensional reduction and clustering.

**Figure supplement 3.** Test of differing cluster resolutions for male and female *fru P1* neuron data sets.

**Figure supplement 4.** Test of differing cluster resolutions for full *fru P1* neuron data set and cluster defining marker gene expression for all data sets.

**Figure supplement 5.** Seurat data integration of full data set and comparison to merged analysis.

**Figure supplement 6.** Visualization of GFP, *fru*, *roX1*, and *roX2* expression.

**Figure supplement 7.** Gene ontology (GO) enrichment analysis for marker genes lists.

**Figure supplement 8.** Visualization of *DIP-iota* expression.

*Goodwin, 2004*; *Cachero et al., 2010*; *Clowney et al., 2015*; *Datta et al., 2008*; *Kallman et al., 2015*; *Kimura, 2008*; *Kimura et al., 2005*; *Ruta et al., 2010*; *Yu et al., 2010*). Sex differences in *fru P1* neurons begin to be established during metamorphosis, coinciding with when Fru$^M$ is at its peak expression in males, during the mid-pupal phase (*Arthur et al., 1998*; *Belote and Baker, 1987*; *Chen et al., 2021*; *Kimura et al., 2005*; *Lee et al., 2000*; *Stockinger et al., 2005*). Therefore, a major remaining question is how does *fru* establish these sex differences and ultimately the potential for sex-specific behavior during development. This scRNA-seq study addresses what are the different molecular classes of *fru P1* neurons that arise during development in each sex.

Work from our laboratory and others have examined gene expression in *fru P1* neurons, Fru$^M$ regulated expression, Fru$^M$ binding, chromatin modifications in *fru P1* neurons, and open chromatin regions in *fru P1* neurons (*Brovkina et al., 2021*; *Dalton et al., 2013*; *Goldman and Arbeitman, 2007*; *Neville et al., 2014*; *Newell et al., 2016*; *Palmateer et al., 2021*; *Vernes, 2014*). Among these studies, only two have evaluated the mid-pupal stage (*Neville et al., 2014*; *Palmateer et al., 2021*). However, these studies used bulk sequencing approaches, thereby lacking the resolution to examine differences in gene expression across individual neurons. To determine the gene expression signatures for individual *fru P1* cells, we performed single-cell RNA-seq during a critical developmental stage for establishing the potential for sex-specific behavior. This produced three cell atlases containing: male *fru P1* neurons, female *fru P1* neurons, and a full data set of male and female *fru P1* neurons co-analyzed. We focus our analyses on the full data set of 25,518 neurons from both males and females that formed 113 molecular clusters after dimensionality reduction using Uniform Manifold Approximation and Projection (UMAP) and clustering. We examine sex differences in gene expression and neuron number within these clusters. We also manually annotated several clusters using the enriched gene expression signatures. We annotate clusters based on gene expression consistent with producing fast-acting neurotransmitters, neuropeptides, receptors, TFs, and several previously characterized *fru P1* neuronal populations. We compare our cluster annotations to those in recent *Drosophila* neuronal scRNA-seq studies, by examining marker gene overlap. Additionally, we examine the in vivo spatial patterns of overlapping expression with *fru P1* and several marker genes to provide anatomical links to our analyses. Our identification of a population of neurons that express genes that regulate circadian behavior led to our discovery of a set of *fru P1* neurons that regulate activity and period length in a sex-specific manner. Altogether, our data sets show the molecular and functional heterogeneity of *fru P1* neurons, revealing the diversity of neurons required to build reproductive behaviors in males and females. While the data reveal sex differences in gene expression within molecularly defined clusters, the observation that nearly all clusters are comprised of male and female neurons indicates that male and female *fru P1* neurons also share common gene expression repertoires, with sex-specific information overlaid on these core patterns.

## Results and discussion

### scRNA-seq of *fru P1*-expressing cells in the pupal CNS reveals cell type diversity

To gain insight into the molecular and functional differences across *fru P1*-expressing neurons (*fru P1* neurons) in males and females, we performed scRNA-seq, using the 10× Genomics platform. This approach allows us to examine the transcriptomes of individual *fru P1* neurons. To enrich for *fru P1* neurons, fluorescence-activated cell sorting (FACS) was performed on dissociated neurons from pupae CNS tissue, at 48 hr after puparium formation (APF), that expressed membrane-bound GFP in *fru P1* neurons (*fru P1>mCD8::GFP*) (*Manoli et al., 2005*; *Figure 1B*). This is the pupal stage where Fru$^M$ is at peak expression in males, based on immunostaining, and a critical period for when *fru P1* establishes the potential for behavior (*Arthur et al., 1998*; *Belote and Baker, 1987*; *Chen et al., 2021*; *Lee et al., 2000*). After 10× library construction, Illumina sequencing, read mapping, and processing with the 10× Genomics CellRanger pipeline, we obtained gene-barcode matrices for further analyses.

These matrices were filtered to retain high-quality cells: cells were removed that had >5% mitochondrial transcripts (dying and/or stressed cells),<200 genes detected (empty droplets), those expressing more than 4000 genes (nfeatures), and/or possessing more than 20,000 unique molecular identifiers (UMIs; a metric for transcript counts) (potential multiplets) (*Figure 1—figure supplement 1*). After filtering, there were 7988 male cells and 17,530 female cells in the pooled replicates (*Figure 1—figure*

*supplement 1*). The pooled replicates from males have 1653 genes and 4712 UMIs per cell (median values). The pooled replicates from females have 1729 genes and 5186 UMIs per cell (median values) (*Source data 1*). The combined filtered data from both sexes contain 25,518 *fru P1*-expressing cells (hereafter called the full data set). In the full data set, the cells have a median of 1706 genes and 5046 UMIs detected per cell, which is on par with or better coverage than other single-cell atlas studies that examine the *Drosophila* CNS (*Allen et al., 2020*; *Avalos et al., 2019*; *Croset et al., 2018*; *Davie et al., 2018*; *Li et al., 2021*; *Source data 1*). As expected, given that *fru P1* is expressed in neurons, 99% of the cells in the full data set are classified as neurons (25,509 cells). This is based on detecting expression of at least one of three genes with known neuronal expression (*embryonic lethal abnormal vision*, *elav*; *neuronal Synaptobrevin*, *nSyb*; *long non-coding RNA*, *noe*; *Source data 2*; *DiAntonio et al., 1993*; *Kim et al., 1998*; *Robinow et al., 1988*). Given this, the cells in the scRNA-seq data set will be called neurons, hereafter. There is minimal cell-cycle impacts on gene expression in neurons in this study, given that during the pupal stage we examined, the majority of neurons are post-mitotic, with only four neuroblasts continuing to divide into the late pupal stages (*Ito and Hotta, 1992*). Furthermore, we observe that a large percentage of neurons express the post-mitotic markers genes *elav* (77%) and *nSyb* (93%) (*Source data 2*). We also searched the data set for expression of *string (stg)* that encodes a tyrosine protein phosphatase required for cell-cycle progression, with elevated expression in mitotic neuroblasts. *stg* expression was only found in 0.3% of neurons in our data set (*Source data 2*), which also indicates that nearly all the neurons examined are post-mitotic.

We next scaled and normalized the expression data from the full data set, and also the male and female data sets separately, using a Seurat workflow (*Stuart et al., 2019*). To generate UMAP plots for visualization and analysis, we selected the number of significant principle components from the expression data, based on Jackstraw analyses (*Stuart et al., 2019*) (p < 0.05 for PCs; *Figure 1—figure supplement 2*). The male and female replicates show large overlap in UMAP space, indicating that there are minimal batch effects between replicates within sex (*Figure 1—figure supplement 1G–I*). To identify neurons that exhibit similarities in their gene expression profiles, we performed graph-based clustering analysis on each data set. To examine gene expression differences between the clusters in each data set, we identified marker genes, using the Seurat 'FindAllMarkers' function and the Wilcox rank sum test (min.pct = 0.25, logfc.threshold = 0.25; which resulted in all called marker genes having p < 0.05). Marker genes are those that have significantly higher expression in one cluster compared to all other clusters within each analysis (*Source data 3*).

Here, we present an in-depth examination of the UMAP and clustering from the full data set and provide links to the sex-specific data sets, in some sections. The male data set has 77 clusters (*Figure 1C*, *Figure 1—figure supplement 3A*), the female data set has 88 clusters (*Figure 1D*, *Figure 1—figure supplement 3C*), and the full data set has 113 clusters (*Figure 1E*, *Figure 1—figure supplement 4*). We observed a high correlation in the number of neurons in each cluster between the replicates from each sex (male replicates *r* = 0.93; female replicates *r* = 0.88) (*Source data 2*), indicating small technical variation in the number of neurons per cluster. Given that *fru P1* is expressed in ~2000 neurons (*Lee et al., 2000*), it is possible we are examining the full repertoire of *fru P1* neurons (~3.9× coverage for male cells and ~8.8× coverage for female cells), assuming there are not biases in the ability to capture different *fru P1* neurons in the experimental workflow. In addition to the full data set analysis on the merged male and female data, we also performed an analysis on integrated male and female data, using the Seurat integration workflow. These results were highly concordant in terms of overlap (*Figure 1—figure supplement 5C, D*, *Source data 2*), providing evidence that the overall conclusions are independent of this step in the data processing (see methods for additional comparative details). We also include the integrated analysis for comparison (*Figure 1—figure supplement 5*, *Source data 2 and 3*).

Next, we evaluate the molecular differences across the clusters by examining marker genes. There is a range in the number of marker genes for each cluster: 48–501 in the male data set, 74–555 in the female data set, and 61–610 in the full data set (*Source data 3*). In total, we find 3582 marker genes in the male data set, 3203 in the female data set, and 3662 in the full data set (*Source data 3*). Gene expression heatmaps show the top five marker genes, based on log fold-change, for each cluster (*Figure 1—figure supplement 4C–E*). The robust expression differences of the marker genes across clusters further confirm that the clusters are different at the molecular level and that the resolution chosen for the clusters results in biologically meaningful separations. To understand the functions of

neurons in the different clusters, we provide detailed annotation of the clusters, with a primary focus on the full data set (113 clusters, *Figure 1E*; *Source data 3*). To gain insight into which genes to focus on for annotation, we determined which gene functional groups are enriched in the marker genes lists, by performing gene ontology (GO) analyses. An examination of the top 10 overrepresented terms for three GO categories: 'molecular function', 'biological process', and 'cellular component' (*Figure 1—figure supplement 7A–C*, *Source data 4*), shows that they are shared across the three data set analyses (*Figure 1—figure supplement 7A–C*), and are characteristic of developing neurons. In addition, we also performed protein domain and pathway enrichment analyses for these marker gene lists and find that the top enriched terms are also shared across the three data sets (*Source data 4*). We annotate each cluster by indicating the distribution of the marker genes from some of the enriched categories (GO, protein domain, and pathways) and from a curated selection of genes that are biologically relevant (*Source data 3*). Additional annotation of the clusters is based on published expression data, GO enrichments of marker genes from each cluster, and overlap with other single-cell studies in the CNS (*Source data 3*; *Allen et al., 2020*; *Avalos et al., 2019*; *Croset et al., 2018*; *Davie et al., 2018*; *Ma et al., 2021*). We provide a name for 46 clusters and information at the level of marker gene expression for the remaining clusters.

## Examination of *fru P1* scRNA-seq clusters with sex differences in neuron number

An examination of the full data set UMAP plot shows separation of male and female neurons within each cluster, with some clusters having more neurons from one sex (*Figure 2A*, *Figure 1—figure supplement 1I*). This separation of the male and female cells is not due to the highly expressed male-specific long non-coding *RNA on the X* 1 and/or 2 (*roX1* and *roX2*), based on a full Seurat analysis performed when these genes are removed (*Figure 1—figure supplement 6I, J*). We note that some of the separation may be due to developmental differences between males and females, given that females have a shorter pupal phase than males (*Bakker and Nelissen, 1963*). The results indicate that male and female neurons have shared gene expression, with some expression differences that drive the separation in UMAP space. This is consistent with our previous studies that showed male and female *fru P1* neurons have shared gene expression profiles, using a cell-type-specific RNA-sequencing approach called Translating Ribosome Affinity Profiling (TRAP) (*Newell et al., 2016*; *Palmateer et al., 2021*; *Thomas et al., 2012*). When we examine the overlap in marker genes from the three scRNA-seq data sets with gene lists from our previous TRAP study that examined gene expression enriched in *fru P1* neurons at 48-hr APF, we find significant overlap between the gene lists (*Source data 3*; *Palmateer et al., 2021*), providing further validation of the scRNA-seq approach. The idea that *fru P1* neurons from males and females would share gene expression repertoires also fits well with the observation that homologously positioned *fru P1* neurons in males and females arise from a shared developmental trajectory (*Ren et al., 2016*), with cell bodies that reside in similar anatomical positions (*Manoli et al., 2005*; *Stockinger et al., 2005*). Additionally, studies using genetic intersectional strategies to examine subpopulations of *fru P1* neurons find homologously positioned neurons in both sexes, across a large range of molecular expression tools (e.g., see *Brovero et al., 2021*; *Cachero et al., 2010*; *Palmateer et al., 2021*; *Yu et al., 2010*).

We next evaluated the clusters that have differences in neuron number between the sexes in our full data set. Given there are 2.19× more female neurons in the full data set we did several tests to determine the impact of this on clustering. After normalizing the number of neurons in each female cluster by dividing by 2.19×, we find that 52% of the clusters show a sex bias in their neuron numbers. 24% show a >twofold difference (sex bias), 28% show >four-fold difference (strong sex bias), and 3.5% of the clusters are sex specific (*Figure 2B*, *Source data 5*). All references to sex-biased clusters in the full data set analysis are on normalized data values (*Source data 5*). The workflow to obtain *fru P1* neurons may have biases in the neurons captured, though our data comparing replicates indicate the purification protocol is highly reproducible given the correlation of cell numbers per cluster between the two replicates (*Source data 2*).

We next determined if this difference is maintained when equal numbers of neurons between the sexes are present in the UMAP, by removing random subsets of female neurons to equal the number of male neurons, which was performed three independent times. We find that six clusters change their status after random removal across some of the independent cell removal analyses

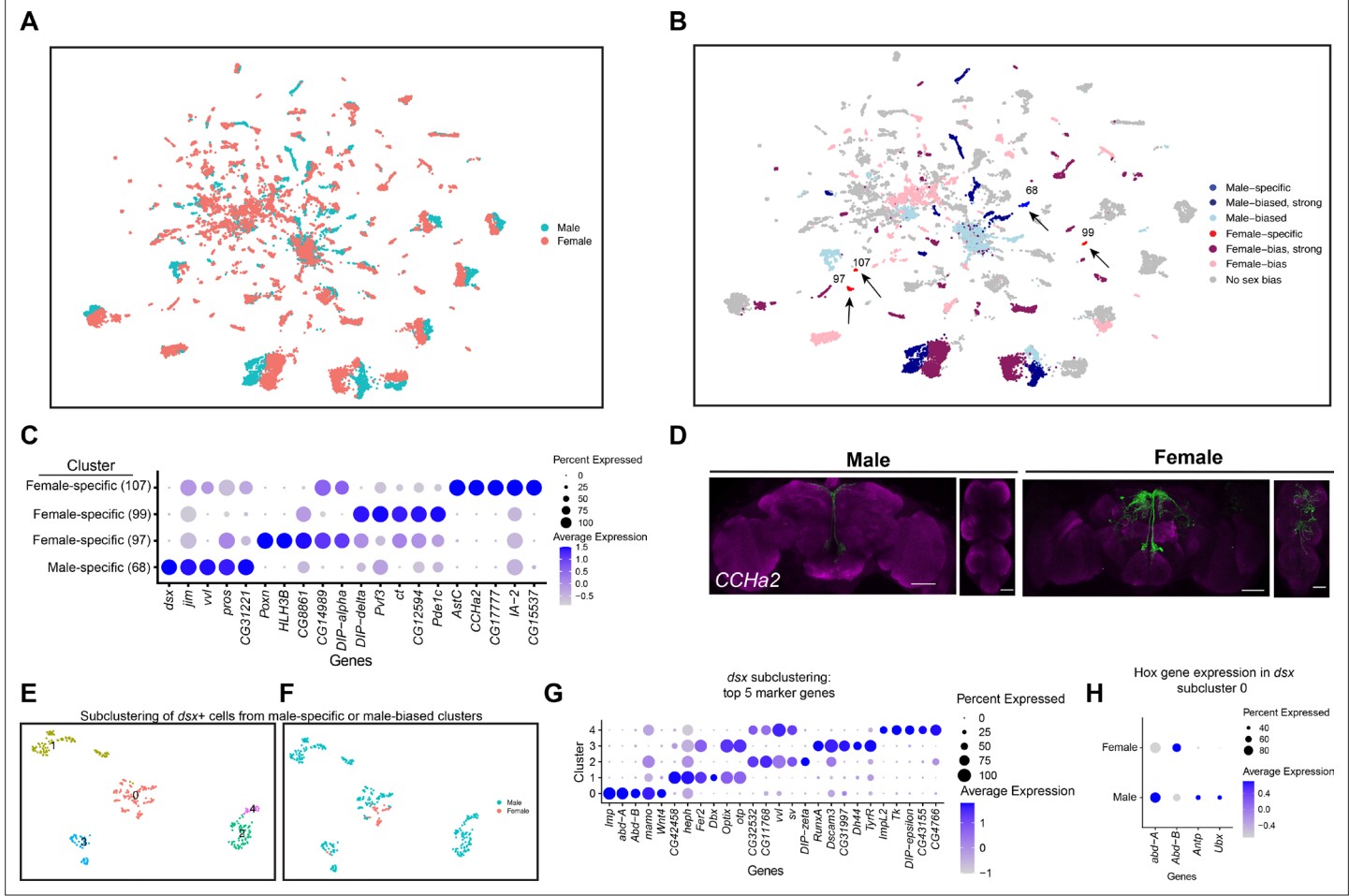

**Figure 2.** Sex differences in single-cell clustering. (**A–B**) Uniform Manifold Approximation and Projection (UMAP) plots of the 25,518 *fru P1* neurons from males and females (full data set) with all cells labeled by sex (**A**) and by sex bias in cell number (**B**) (*Source data 5*). (**B**) Sex-biased classification is based on normalized number of female cells (legend on right, *Source data 5*). Cluster numbers and black arrows indicate female-specific clusters (red) and the male-specific clusters (blue). Female- (light pink) and male biased (light blue) have two-fold more neurons compared to the other sex. Female- (purple) and male-biased, strong (dark blue) have four-fold more neurons compared to the other sex. (**C**) Dot plot of top 5 marker genes in each sex-specific cluster based on log fold-change in expression (*Source data 3*). (**D**) Brain and ventral nerve cord (VNC) confocal maximum projections from 0- to 24-hr adults for *CCHa2 ∩ fru P1*-expressing neurons, with intersecting expression shown in green for males (left) and females (right). Scale bars = 50 μm. (**E, F**) Subclustering analyses of neurons from male-specific and male-biased clusters that express *dsx* defines five subclusters (clusters are indicated by color and number). (**F**) The five subclusters are shown with sex indicated by color, with male cells in blue and female cells in pink. Female cells are present only in subclusters 0 and 3. (**G**) Dot plot of top 5 marker genes in each *dsx* subcluster based on log fold-change in expression (*Source data 3*). (**H**) Dot plot of Hox genes with known VNC expression in *dsx* subcluster 0, for both sexes. For all dot plot (**C, G, H**) panels the dot size indicates the percentage of neurons expressing each gene per cluster (Percent Expressed). Average normalized expression level is shown by color intensity (Average Expression).

The online version of this article includes the following figure supplement(s) for figure 2:

**Figure supplement 1.** Random removal and downsampling of female cells.

**Figure supplement 2.** Gene expression in sex-specific and sex-biased clusters.

(*Figure 2—figure supplement 1A*, *Source data 5*). Further, we repeated the full Seurat workflow by first removing female cells at random (downsampling) and generating new UMAP plots and clustering analysis for three independent downsampled analyses, with equal numbers of male and female cells. To examine the similarity of gene expression between the clusters, we use Pearson correlations in clustifyr (*Fu et al., 2020*). We match 80–85% of the clusters in the downsampled analyses to those found in the full data set, and of those there is high concordance of sex-specific and sex-biased clusters (*Figure 2—figure supplement 1B*, *Source data 5*). However, fewer female-specific clusters are present in the downsampled analyses and three additional male-specific clusters are resolved (*Figure 2—figure supplement 1B*, *Source data 5*). Though differences are revealed in the analyses

when female neurons are randomly removed to equal the number of male cells, most sex-biased clusters are preserved. Taken together, we proceed with the analysis of the clusters in the full data set, as the additional female cells provide information.

## Sex-specific clusters

In the full data set there are four sex-specific clusters (three female-specific and one male-specific), with two maintained after the downsampling analysis (female-specific cluster 107 and male-specific cluster 68) (*Figure 2B*, arrows). One of the top marker genes for the female-specific cluster 107 is *CCHamide-2* (*CCHa2*), which encodes a neuropeptide hormone that stimulates feeding (*Ren et al., 2015*). Cluster 107 neurons may be from the VNC, given that cluster 107 marker genes include Hox segment identity genes *Antennapedia* (*Antp*) and *Ultrabithorax* (*Ubx*). *Antp* and *Ubx* define VNC T1–T3 and T2–T3 thoracic segments, respectively (*Baek et al., 2013*; *Jarvis et al., 2012*). To determine if there are female-specific *CCHa2* neurons that also express *fru P1* (*CCHa2 ∩ fru P1*) in the VNC, we used a genetic intersectional approach to visualize *CCHa2 ∩ fru P1*-expressing neurons. The approach relies on the two-component Gal4/UAS system that drives expression of a Myc-tagged membrane-bound reporter protein (*Nern et al., 2015*). Expression of the UAS-reporter requires Flippase (FLP)-mediated removal of a stop cassette. The expression of FLP is directed by *fru P1* regulatory elements (*Yu et al., 2010*). Therefore, reporter gene expression is limited to *fru P1* neurons that also express Gal4. Throughout this study, we examine expression at 48-hr APF and 0- to 24-hr adults, given that the intersectional approach might be slow to report on expression given the requirement of a genomic, FLP-mediated, recombination event and production of several proteins. Additionally, *fru P1* expression is most thoroughly characterized in adults (*Billeter and Goodwin, 2004*; *Cachero et al., 2010*; *Kimura et al., 2005*; *Lee et al., 2000*; *Manoli et al., 2005*; *Stockinger et al., 2005*; *Yu et al., 2010*). *CCHa2* is also a marker gene for additional clusters, so we expect the expression pattern to include neurons not in cluster 107. There is no detectable reporter gene expression at 48-hr APF. In 1-day-old adults, we find female-specific *CCHa2 ∩ fru P1*-expressing neurons in both the brain and VNC, and *CCHa2 ∩ fru P1*-expressing neurons in the median bundle in both sexes (*Figure 2D*). There are no male *CCHa2 ∩ fru P1*-expressing neurons in the VNC. This observation indicates that there are female-specific *CCHa2 ∩ fru P1* neurons in the VNC, as suggested by the cluster analysis.

The top marker gene for the male-specific cluster 68 is the sex hierarchy gene *dsx*. This is consistent with the observation that there are male-specific populations of *dsx ∩ fru P1* neurons in the brain (pC1, pC2, SN, and SLN) and VNC (TN1 and abdominal ganglion) (*Billeter et al., 2006*; *Ishii et al., 2020*; *Lee et al., 2002*; *Rideout et al., 2007*; *Sanders and Arbeitman, 2008*). Given that cluster 68 does not express Hox segment identity genes that are expressed in the VNC, or other genes with known VNC expression, these neurons are likely from the brain (*Figure 2—figure supplement 2B*). There are two additional clusters in the full data set that have *dsx* as a marker gene: one strongly male-biased (4× more male cells after normalization) and one male-biased cluster (2× more male cells after normalization) (*Figure 2—figure supplement 2C*, *Source data 3 and 4*). To analyze these clusters more deeply, we subclustered these neurons, selecting for those with detectable *dsx* expression and identified marker genes for each subcluster (*Figure 2B*, *Figure 2—figure supplement 2C*). This reanalysis resolved five subclusters, retaining the original male-biased cluster as one cluster (subcluster 0), and generating two clusters from the original male-specific cluster (subclusters 2 and 4) and two clusters from the original strongly male-biased cluster (subclusters 1 and 3; *Figure 2E, F*, *Figure 2—figure supplement 2D*).

We propose that the male-biased subcluster 0 contains neurons from the VNC, given these neurons also express *Antp*, *Ubx*, abdominal A (*abd-A*), and *Abdominal B* (*Abd-B*) (*Figure 2E, F*, *Figure 2—figure supplement 2E–H*, *Source data 3*), and the most significant Berkeley *Drosophila* Genome Project (BDGP) enrichment term for the marker genes is 'ventral nerve cord', based on analysis in Flymine portal (*Lyne et al., 2007*). In the VNC, both males and females have *dsx ∩ fru P1* neurons in the abdominal ganglion, and there are male-specific *dsx ∩ fru P1* neurons in the metathoracic ganglion called TN1 and TN2 neurons (*Billeter et al., 2006*; *Lee et al., 2002*; *Rideout et al., 2007*; *Sanders and Arbeitman, 2008*). When we examine the expression of Hox genes in subcluster 0, many of the male and female neurons show expression of *abd-A* and *Abd-B* (*Figure 2H*). Given this, we propose these neurons are the *dsx ∩ fru P1* neurons in the abdominal ganglion. There are also neurons in subcluster 0 only in males, with expression of *Antp* and *Ubx* (*Figure 2H*) that we propose

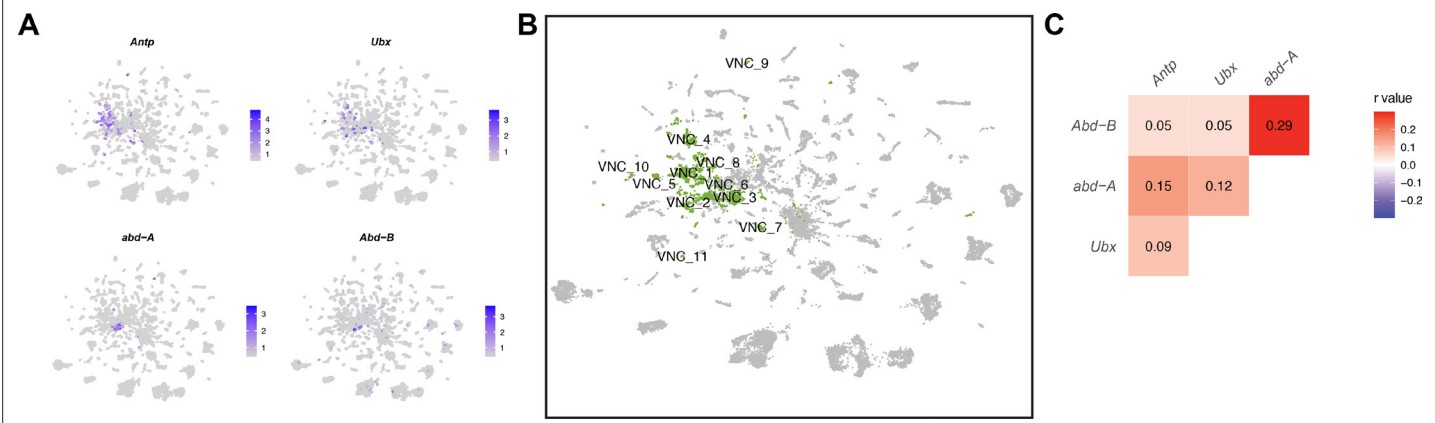

**Figure 3.** Ventral nerve cord (VNC) cluster annotations based on *Hox* gene expression. (**A**) Gene expression feature plots showing neurons that express *Hox* genes with known expression in the VNC (*Antp*, *Ubx*, *abd-A*, and *Abd-B*), in the full data set Uniform Manifold Approximation and Projection (UMAP). The gene expression feature plots show gene expression levels in purple in the UMAP, with color intensity proportional to the log-normalized expression levels. (**B**) UMAP plot showing annotated VNC clusters in the full data set analysis. The VNC clusters are numbered and highlighted in green (*Source data 3*). (**C**) Heatmap of correlation of *Hox* gene expression in single neurons in the full data set. Pearson's *r* values denoted on heatmap with color, according to legend (right).

The online version of this article includes the following figure supplement(s) for figure 3:

**Figure supplement 1.** Ventral nerve cord (VNC) cluster annotations based on *Hox* expression for male and female data sets.

are the male-specific metathoracic TN1 or TN2 neurons. Consistent with this, we previously showed TNI and TN2 neurons underwent cell death in females by 48-hr APF (*Sanders and Arbeitman, 2008*).

We propose that the strongly male-biased and male-specific *dsx* and *fru P1*-expressing neuron clusters contain neurons from the brain, based on expression of Hox genes (*Figure 2—figure supplement 2E–H*). Previous studies have shown *dsx* ∩ *fru P1* neurons in the brain in PC1 and PC2 regions, as well as small sets of neurons called SN and SLN (*Lee et al., 2002*; *Rideout et al., 2007*; *Sanders and Arbeitman, 2008*). The strongly male-biased cluster split into two subclusters, one that has two female neurons (subcluster 3), and the other is now male specific (subcluster 1). The male-specific cluster (*Figure 2—figure supplement 2C*) split into subclusters 2 and 4 that are positioned closely together in the UMAP space (*Figure 2—figure supplement 2D*). A recent study using genetic intersectional approaches found 1–2 pC1/pC2 female *dsx* ∩ *fru P1* neurons (*Ishii et al., 2020*), whereas we did not detect any (*Sanders and Arbeitman, 2008*), leaving open that possibility that neurons in subcluster 3 are from either pC1 or pC2 brain regions, given the detection of female cells. Also, it has been proposed that male pC2 *dsx*-expressing neurons are comprised of distinct populations, based on fasciculation patterns from either medial and lateral positions (pC2 renamed pC2m and pC2l) (*Robinett et al., 2010*). Thus, subclusters 1, 2, 3, and/or 4 may contain neurons that are from the pC1 or pC2, SN, SLN *dsx*-expressing regions in brain, given there are no known molecular markers that would distinguish these populations and there are predicted to be several distinct populations based on morphological and spatial features. An examination of marker genes unique to each subclusters suggests there may be functional differences (pathway enrichment, Benjamini–Hochberg p < 0.1; *Source data 6*).

## Hox gene expression identifies *fru P1* neurons from the VNC

To annotate clusters from the VNC, we examined expression of Hox genes *Antp*, *Ubx*, *abd-A*, and *Abd-B* that are known to be expressed in an anterior to posterior pattern in the VNC (*Baek et al., 2013*). These genes show restricted expression in a subset of clusters (*Figure 3A*, *Figure 3—figure supplement 1A–H*). Additionally, we find several have more than one of these Hox genes as a marker gene (*Source data 3*). For example, male data set cluster 1 has all four genes as markers, which indicates that neurons from several thoracic and abdominal VNC segments may be present in some clusters. This also indicates that neurons from different segments may have shared gene expression profiles. Therefore, any cluster with one or multiple Hox marker genes was annotated as a 'VNC' cluster (*Source data 3*), resulting in seven VNC clusters in both the male and female data set, and

11 in the full data set (*Figure 1D, F, G*, *Figure 3B*, *Figure 3—figure supplement 1I–J*, *Source data 3*). In addition, we performed a correlation analysis to examine co-expression of *Antp*, *Ubx*, *abd-A*, and *Abd-B* within single neurons. All the data sets have significant positive correlations for expression of these genes, with the highest correlations found between *abd-A* and *Abd-B*, consistent with these genes having the strongest overlap in their spatial expression (*Baek et al., 2013*; *Figure 3C*, *Figure 3—figure supplement 1K, L*). This finding differs from scRNA-seq on the adult VNC, which showed significant anti-correlations in expression between all pairings, with the exception of *abd-A* and *Abd-B* (*Allen et al., 2020*). This is consistent with the observation that during development there is a refinement of Hox gene expression that limits co-expression in VNC neurons in adults (*Allen et al., 2020*; *Baek et al., 2013*).

### Identification of *fru P1* mushroom body Kenyon cell neuron populations

The mushroom body is a structure in the brain that is critical for *Drosophila* learning and memory (reviewed in *Davis, 1993*; *Heisenberg, 2003*). The mushroom body is comprised of intrinsic neurons called the Kenyon cells (KCs), as well as extrinsic input and output neurons (*Aso et al., 2014*; *Li et al., 2020*). *fru P1* is expressed in the mushroom body γ and αβ KCs, whereas there are no reports of expression in the α′β′ KCs. Furthermore, previous work has demonstrated that when *fru P1* function is reduced by RNAi in γ or αβ KCs there are deficits in courtship learning (*Manoli et al., 2005*). To gain insight into *fru P1* MB KCs, we annotated clusters that are KC subtypes, based on marker gene

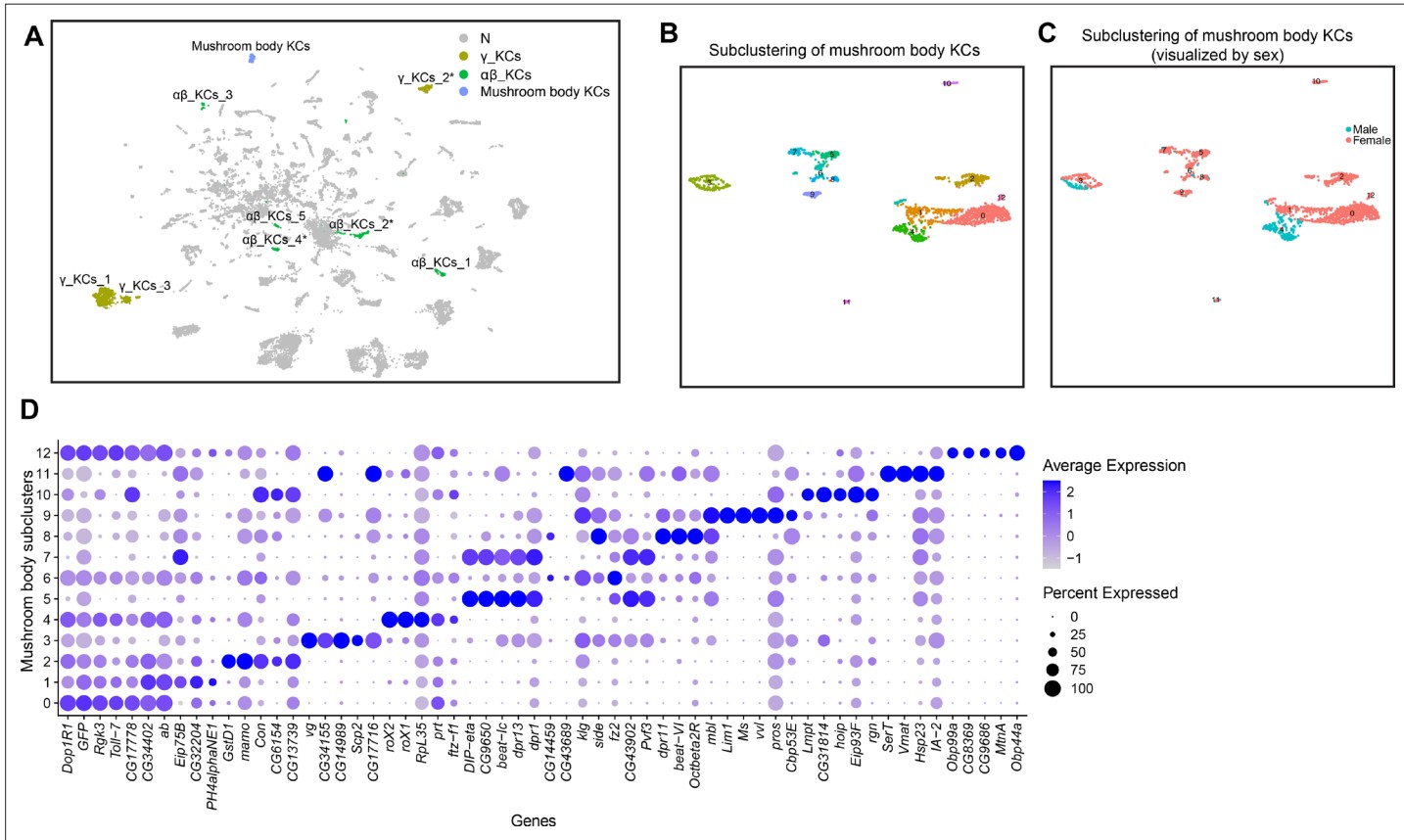

**Figure 4.** Annotation of mushroom body Kenyon cells (KCs). (**A**) Uniform Manifold Approximation and Projection (UMAP) plot highlighting KC cluster annotations in the full data set analysis (*Source data 3*). N (gray) indicates clusters not meeting marker gene criteria used to annotate KC clusters (see methods and *Figure 4—figure supplement 1A*). (**B**) Subclustering analysis of neurons from mushroom body KC clusters (colored clusters from panel **A**), creating 13 subclusters. (**C**) The 13 subclusters are shown with sex indicated by color, with male cells in blue and female cells in pink. (**D**) Dot plot of top 5 marker genes in each mushroom body KC subcluster based on log fold-change in expression (*Source data 3*). Dot size indicates the percentage of neurons expressing each gene per cluster (Percent Expressed). Average normalized expression level is shown by color intensity (Average Expression).

The online version of this article includes the following figure supplement(s) for figure 4:

**Figure supplement 1.** Mushroom body Kenyon cell (KC) annotations.

analysis, using criteria similar to a previous scRNA-seq study of the *Drosophila* midbrain (*Croset et al., 2018*; *Figure 4A*, *Source data 3*). The marker gene criteria are schematically summarized (*Figure 4—figure supplement 1A*). Clusters that contain KCs can be identified based on having *eyeless* (*ey*) and/or *Dopamine 1-like receptor 2* (*Dop1R2*, also known as *damb*) as marker genes (*Han et al., 1996*; *Kurusu et al., 2000*; *Source data 3*). To identify clusters that contain αβ KCs subtypes, we additionally required that *short neuropeptide precursor F* (*sNPF*), and/or *Fasciclin 2* (*Fas2*) are marker genes (*Cheng et al., 2001*; *Crittenden et al., 1998*; *Johard et al., 2008*). This identified three in the full data set analyses (*Source data 3*). The clusters that contain γ KCs were identified as those that have *trio* as a marker gene, in addition to the marker genes described for identifying αβ KCs. *trio* encodes a Rho guanine nucleotide exchange factor that has high expression restricted to the γ KCs, at 48-hr APF (*Awasaki et al., 2000*). This allowed us to confidently annotate two γ KC clusters in each data set analysis (*Source data 3*). We also annotate potential γ KCs and αβ KC clusters based on having marker gene combinations consistent with containing KCs (indicated by *; *Source data 3*). For example, we find one potential γ KC cluster in the female data set analysis that has only *Dop1R2* and *trio* marker genes. We also find several potential αβ KC clusters where *sNPF* and *Fas2* are marker genes, and there is expression of *ey* and/or *Dop1R2*, but these are not marker genes (see methods, *Figure 4—figure supplement 1*). We could not confidently identify clusters containing *fru P1*-expressing α′β′ KCs, if they exist, as there are no known marker gene patterns that would distinguish this population at 48-hr APF.

As several studies have implicated the αβ and γ lobes as being important for courtship memory in males, we next wanted to further examine the six high confidence KC clusters identified in the full data set analysis by subclustering and assessing marker genes. This resulted in 13 subclusters (*Figure 4B*), where the original three αβ KC clusters separate into five subclusters and the two γ KC clusters separate into seven subclusters (*Figure 4—figure supplement 1J*). All subclusters identified contain cells from each replicate (*Figure 4—figure supplement 1K*). The GO enrichment analyses for these subclusters are consistent with learning and memory functions (*Source data 4 and 6*). We also find marker genes in subclusters that have previously been shown to be expressed in the mushroom body in a sex-differential manner (*Obp99a*, *Obp44a* in females) (*Crocker et al., 2016*), other genes known to be highly expressed in KC neurons (*Dop1R1*) (*Croset et al., 2018*), and genes indicative of serotonin production (*Vmat* and *SerT*) (*Figure 4C, D*, *Source data 6*). Given the large number of KC neurons and subtypes previously defined using Gal4 driver tools (*Aso et al., 2014*), these results point to molecular differences that need be further validated in vivo to gain insight into the different morphological categories. Altogether, this analysis suggests the existence of *fru P1* KC subtypes that might have a role in learning during reproductive behaviors.

## Identification of *fru P1* neurons that produce fast-acting neurotransmitters

We next identify the *fru P1* neuronal clusters that produce the fast-acting neurotransmitters (FANs) acetylcholine, glutamate, and GABA. To annotate clusters, we used marker genes that encode products in an FAN biosynthetic pathway or encode transporters for the neurotransmitters, as previously done (*Allen et al., 2020*; *Avalos et al., 2019*; *Croset et al., 2018*; *Davie et al., 2018*). *Vesicular acetylcholine transporter* (*VAChT*) and *Choline acetyltransferase* (*ChAT*) were used to identify cholinergic neurons. *Vesicular glutamate transporter* (*VGlut*) was used to identify glutamatergic neurons. *Glutamic acid decarboxylase 1* (*Gad1*) and *Vesicular GABA Transporter* (*VGAT*) were used to identify GABAergic neurons (*Figure 5A*, *Figure 5—figure supplement 1A, H*). The majority of clusters that have neurons that produce FANs are annotated as cholinergic. For example, in the full data set analysis there are 32 cholinergic clusters, 18 GABAergic, and 16 glutamatergic (*Figure 5A*, *Source data 3*). Only one full data set cluster has marker gene expression for more than one FAN (cholinergic and glutamatergic), and the male and female data set each have one cluster that has marker gene expression for more than one FAN (GABAergic and glutamatergic) (*Figure 5A*, *Figure 5—figure supplement 1A, H*, *Source data 3*).

To determine if individual neurons express genes that would be indicative of a neuron producing multiple FANs, we examined expression of *VAChT* (cholinergic), *Gad1* (GABAergic), and *VGlut* (glutamatergic) at the neuron level, in each data set (*Figure 5B*, *Figure 5—figure supplement 1B, I*). Similar to other single-cell studies in the *Drosophila* CNS, this analysis revealed that *fru P1* neuronal

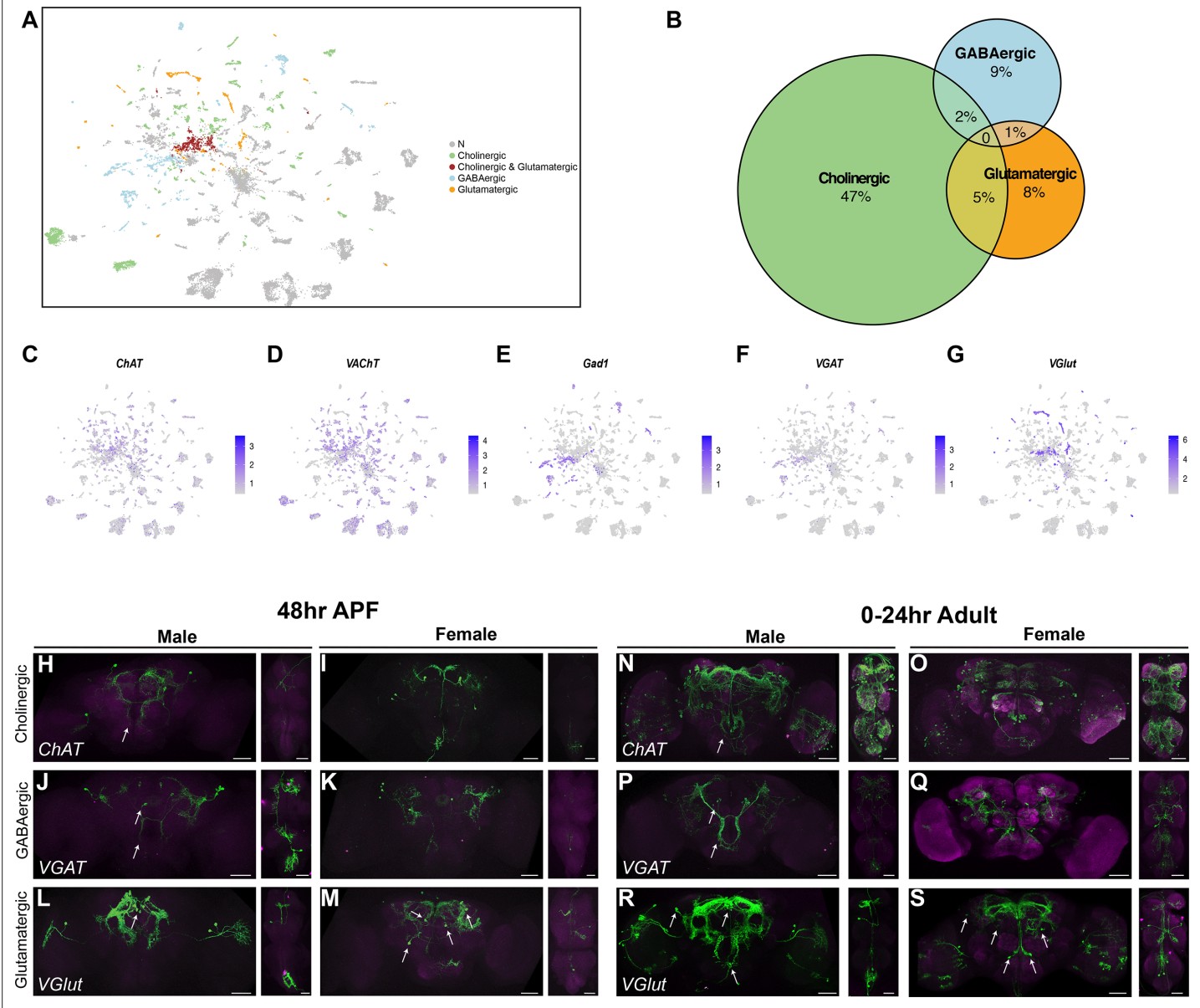

**Figure 5.** Annotation of neurons that produce fast-acting neurotransmitters (FANs). (**A**) Uniform Manifold Approximation and Projection (UMAP) plot showing annotated clusters with neurons that produce FANs, in the full data set analysis (***Source data 3***). In the color legend on right, N (gray) indicates clusters with no FAN marker gene expression. (**B**) Size proportional Euler diagram showing the percentage of cells with overlapping expression of genes indicative of cholinergic (*VAChT*), GABAergic (*Gad1*), and glutamatergic (*VGlut*) neurons, in the full data set. (**C–G**) Gene expression feature plots showing neurons that express genes indicative of FAN production: acetylcholine (*ChAT and VAChT*), GABA (*Gad1* and *VGAT*), and glutamate (*VGlut*), in the full data set UMAP. (**H–S**) Brain and ventral nerve cord (VNC) confocal maximum projections for FAN-producing ∩ *fru P1*-expressing neurons, with intersecting expression shown in green for males and females, as indicated. *ChAT* ∩ *fru P1* neurons, *VGAT* ∩ *fru P1* neurons, and *VGlut* ∩ *fru P1* neurons in 48-hr after puparium formation (APF) and 0- to 24-hr adult are shown. Arrows point to regions with sexually dimorphic projections or cell bodies. Scale bars = 50μm.

The online version of this article includes the following figure supplement(s) for figure 5:

**Figure supplement 1.** Expression of genes consistent with production of fast-acting neurotransmitters (FANs).

populations largely produce only one FAN and therefore are mostly exclusive (***Figure 5B***, ***Figure 5— figure supplement 1B, I***; ***Allen et al., 2020***; ***Avalos et al., 2019***; ***Croset et al., 2018***). In contrast to these other scRNA-seq studies, no neurons in our data set express genes indicative of production of three FANs, and ≤8% of neurons co-express two genes indicative of production of two FANs (***Allen et al., 2020***; ***Avalos et al., 2019***; ***Croset et al., 2018***). The expression of these FAN biosynthetic

pathway or transporter genes at the neuron level are also shown (*Figure 5C–G*, *Figure 5—figure supplement 1C–G, J–N*), indicating that 62% are cholinergic (*VAChT* and/or *ChAT*), 19% are GABAergic (*Gad1* and/or *VGAT*), and 14% are glutamatergic (*VGlut*) (*Source data 2*).

To visualize the spatial patterns of *fru P1* neurons that produce different FANs we use a genetic intersectional strategy that relies on Gal4 transgenes that report on *ChAT* (cholinergic), *VGAT* (GABAergic), and *VGlut* (glutamatergic) (*Deng et al., 2019*; *Nern et al., 2015*; *Yu et al., 2010*). We can detect the three different *fru P1*-expressing FAN classes at 48-hr APF in both sexes, based on reporter gene expression that labels the neuronal membrane of each FAN ∩ *fru P1*-expressing neuronal population. However, we do not find broad expression of *ChAT* at 48-hr APF, which is not consistent with our single-cell data that shows extensive expression of *ChAT* (*Figure 5C*, *Figure 5—figure supplement 1C, J*). Examination of *ChAT* ∩ *fru P1*-expressing neuronal populations in 0- to 24-hr adults does reveal broader expression of *ChAT* compared to *VGAT* and *VGlut* (*Figure 5N–S*).

We find sexual dimorphisms in the morphology of *fru P1* neurons in the FAN ∩ *fru P1*-expressing neuronal populations, some that have been previously described (*Figure 5H–S*; *Cachero et al., 2010*; *Kimura et al., 2005*; *Yu et al., 2010*). For example, in *CHAT* ∩ *fru P1* neurons, we observe male-specific expression in the tritocerebral loop (*Yu et al., 2010*). In *VGAT* ∩ *fru P1* neurons, we observe the sexually dimorphic mAL neurons, which have previously been shown to be GABAergic and suppress male courtship behavior in adults (*Figure 5J, P*; *Kallman et al., 2015*; *Kimura et al., 2005*; *Koganezawa et al., 2010*). In addition, we find that the *VGlut* ∩ *fru P1* expression patterns exhibit sex differences in cell body positioning (*Figure 5L, M, R, S*). In the male brain, the majority of *VGlut* ∩ *fru P1* cell bodies are exclusively positioned near the Pars intercerebralis (PI), whereas in females, the cell bodies are additionally positioned throughout the midbrain (*Figure 5L, M, R, S*, white arrows).

There are 17 *VGlut* ∩ *fru P1* clusters, without a corresponding number of spatially distinct *VGlut* ∩ *fru P1* in the CNS in males and females (*Figure 5A, L, M, R, S, Source data 3*). This suggests that neurons that reside in similar spatial position may have different molecular identities, as has previously been found in other single-cell studies of the *Drosophila* CNS (*Ma et al., 2021*). To interrogate gene expression differences, we further examined the top marker genes for these clusters. We found that *prospero* (*pros*), a gene which encodes a homeobox TF that promotes neural differentiation (*Doe et al., 1991*), is a marker gene in three clusters (*Source data 3*). When we examined *pros* expression across all neurons in each glutamatergic cluster we find five additional clusters where *pros* is not a marker gene but is expressed in a high percentage of cells (*Figure 5—figure supplement 1O*). This analysis also showed that eight clusters exhibit little to no *pros* expression (*Figure 5—figure supplement 1O*).

Next, we determined if neurons that are spatially close together are different at the molecular level, by examining Pros protein. In males, *VGlut* ∩ *fru P1* brain neurons are close together spatially, with the majority of the cell bodies positioned together near the Pars intercerebralis region of the brain. These neurons are different molecularly, as only some have nuclear Pros staining, based on immunofluorescence analyses (*Figure 5—figure supplement 1P–P'*, white arrows, *Figure 5—figure supplement 1Q–U*). Earlier in development, *pros* has a role in neuroblast differentiation (*Doe et al., 1991*). However, by 48-hr APF only mushroom body KC neuroblasts are thought to be dividing (*Ito and Hotta, 1992*). The *VGlut* ∩ *fru P1* neurons are not KC neurons, leaving open the role of *pros* in the *fru P1* neurons evaluated here. Overall, the cluster analyses reveals the molecular heterogeneity in closely positioned neurons in the brain, especially in the well-defined spatial clusters that the field has defined (*Lee et al., 2000*; *Manoli et al., 2005*; *Stockinger et al., 2005*).

## Aminergic populations of *fru P1* neurons

Subsets of *fru P1* neurons produce the biogenic amines dopamine, serotonin, tyramine, and octopamine (*Billeter and Goodwin, 2004*; *Certel et al., 2010*; *Certel et al., 2007*; *Jois et al., 2018*; *Lee and Hall, 2001*; *Lee et al., 2001*; *Yu et al., 2010*; *Zhang et al., 2016*). These aminergic *fru P1* neurons have roles regulating male mating drive (dopamine, *Zhang et al., 2016*), mate discrimination (octopamine/tyramine, *Certel et al., 2010*; *Certel et al., 2007*), and copulation duration (serotonin, *Jois et al., 2018*; *Lee and Hall, 2001*). To identify clusters with aminergic-releasing properties, we use marker gene expression of *vesicular monoamine transporter* (*Vmat*), that encodes a transporter for vesicular packaging of biogenic amines (*Greer et al., 2005*). *Vmat* is a marker gene in three clusters in the full data set, two clusters in the female data set, and is not detected as a marker gene in

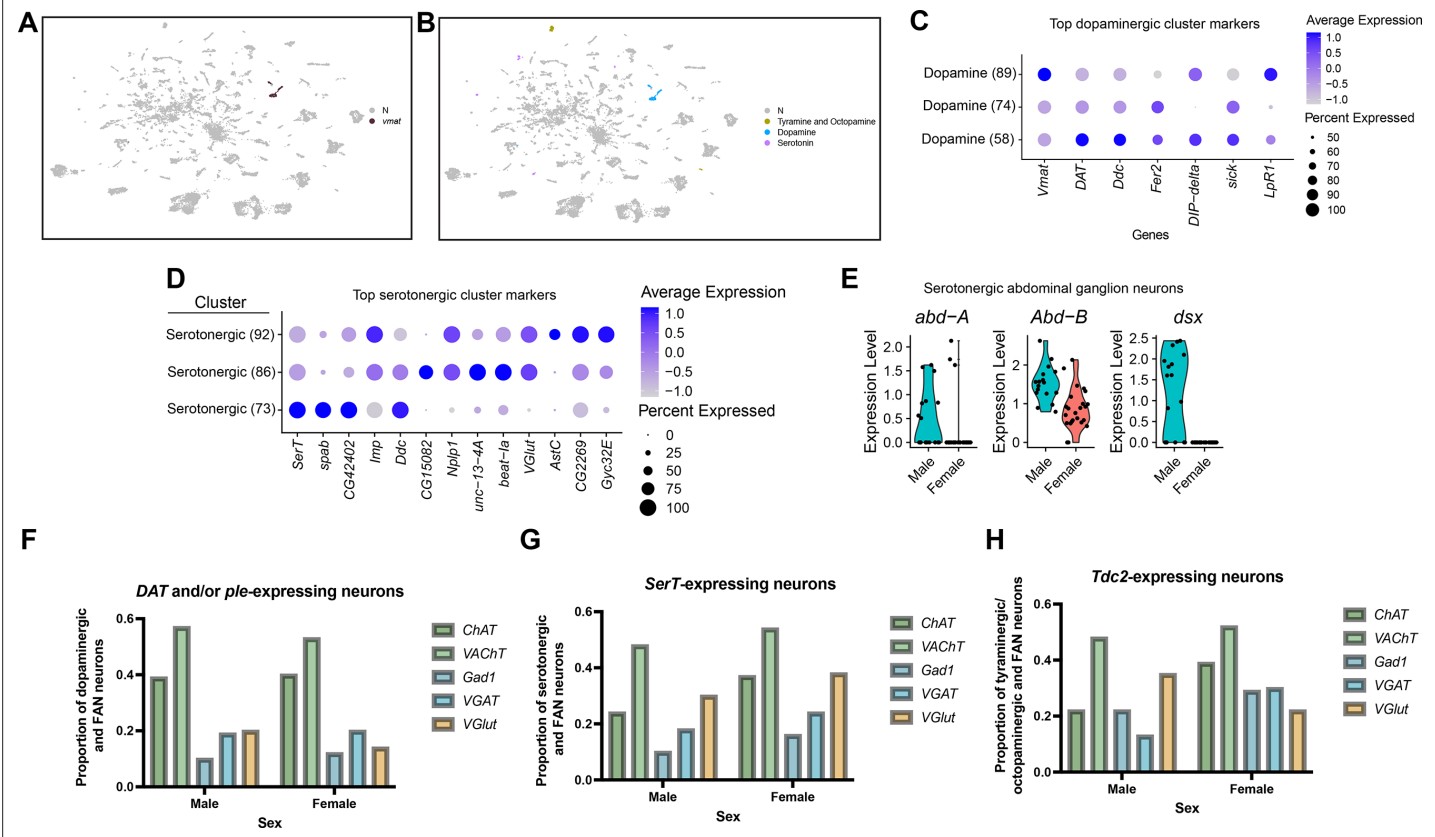

**Figure 6.** Annotation of neurons that produce aminergic neurotransmitters. (**A**) Uniform Manifold Approximation and Projection (UMAP) plot showing clusters where *Vmat* is a marker gene. (**B**) UMAP plot showing clusters of neurons that produce aminergic neurotransmitters are identified based on marker gene expression: Tyramine and Octopamine (*Tdc2*), Dopamine (*Dat* and/or *ple*), Serotonin (*SerT*), in the full data set (**Source data 3**). The N (gray) indicates clusters with no aminergic marker genes expression. (**C–D**) Dot plots of top 5 genes for the three dopaminergic clusters (**C**) and three serotonergic clusters (**D**), based on log fold-change in gene expression. Dot size indicates the percentage of neurons expressing each gene per cluster (Percent Expressed). Average normalized expression level is shown by color intensity (Average Expression). (**E**) Violin plots showing expression of *abd-A*, *Abd-B*, and *dsx* in male and female serotonergic cells predicted to be from the abdominal ganglion ventral nerve cord (VNC) neurons. These cells were identified by *SerT*, *abd-A*, and/or *Abd-B* expression, resulting in 41 neurons (17 males and 24 females). (**F–H**) Proportion of *fru P1* dopaminergic neurons (*DAT* and/or *ple*-expressing) (**F**) serotonergic neurons (SerT-expressing) (**G**), and octopaminergic neurons (*Tdc2*-expressing) (**H**) that co-express genes indicative of fast-acting neurotransmitter (FAN) expression in males and females. The proportion that are aminergic and cholinergic (*ChAT and VAChT*) are shown with green bars, GABAergic (*Gad1* and *VGAT*) are shown with blue bars, and glutamatergic (*VGlut*) are shown by orange bars. Legend to right of each plot.

The online version of this article includes the following figure supplement(s) for figure 6:

**Figure supplement 1.** Expression of genes consistent with production of aminergic neurotransmitters.

the male data set (***Figure 6A***, ***Source data 3***), though it is expressed in restricted clusters of neurons in the male data set analysis (***Figure 6—figure supplement 1A***). This restricted expression pattern of *Vmat* is consistent with other scRNA-seq analyses (***Allen et al., 2020***; ***Croset et al., 2018***; ***Davie et al., 2018***).

Next, we examined the spatial expression patterns of *Vmat ∩ fru P1* neurons (***Figure 6—figure supplement 1D–G***). At 48-hr APF in both sexes, we find limited expression in the brain and no expression in the VNC (***Figure 6—figure supplement 1D–E***). In 0- to 24-hr adults, we also observe limited expression in the brains of both sexes, with projections around the mushroom body γ lobes (***Figure 6—figure supplement 1F, G***). Further, in 0- to 24-hr adults, there is expression in the VNC of both sexes, consistent with previous reports of *Vmat* expression in the 5-day adult VNC (***Figure 6—figure supplement 1F, G***; ***Allen et al., 2020***). The restricted *Vmat ∩ fru P1* expression is consistent with the limited expression in the clusters.

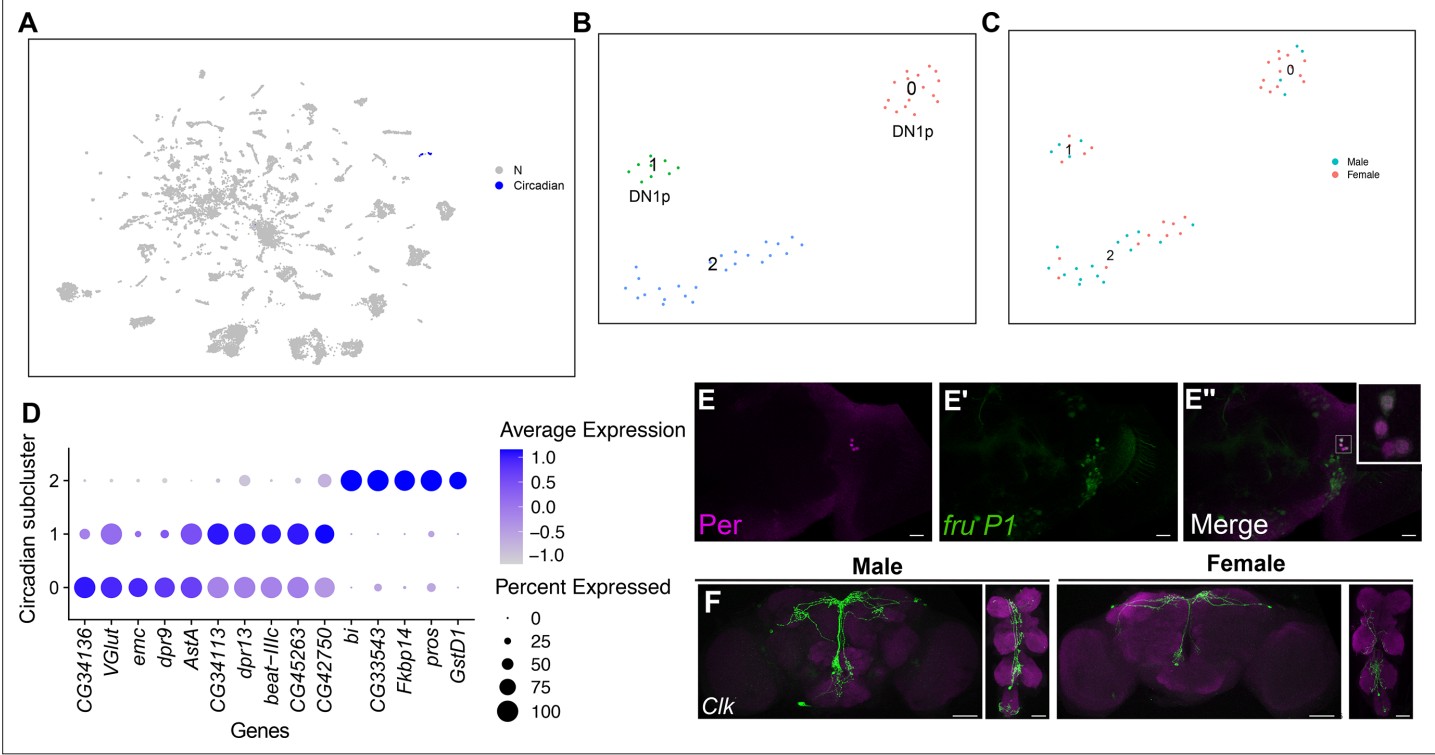

**Figure 7.** Annotation of neurons that express circadian clock genes. (**A**) Uniform Manifold Approximation and Projection (UMAP) plot showing clusters annotated as circadian clock neurons (blue), in full data set (*Source data 3*). (**B**) Subclustering analysis of neurons from annotated circadian clusters (colored clusters from panel **A**), defining three subclusters. (**C**) The three subclusters are shown with sex indicated by color. The male cells are in blue and female cells are in pink. (**D**) Dot plot of top 5 marker genes in each circadian subcluster based on log fold-change gene expression (*Source data 3*). Dot size indicates the percentage of neurons expressing each gene per cluster (Percent Expressed). Average normalized expression level is shown by color intensity (Average Expression). (**E–E''**) Confocal maximum projections of brains immunostained for Per (magenta) and *fru P1>mcd8::GFP* neurons, from 48-hr after puparium formation (APF) male pupae, at ×40 magnification. (**E''**) contains zoomed inset of *fru P1>mcd8::GFP* and Per (magenta) neurons in upper right. The right brain hemisphere is shown. (**F**) Brain and ventral nerve cord (VNC) confocal maximum projections for *Clk856 ∩ fru P1* expression in 0- to 24-hr adult males (left) and females (right). Scale bars = 50 µm.

The online version of this article includes the following figure supplement(s) for figure 7:

**Figure supplement 1.** Visualization of circadian clock gene expression.

**Figure supplement 2.** Activating *Clk856 ∩ fru P1* neurons effects locomotor activity in light:dark (LD).

**Figure supplement 3.** Activating *Clk856 ∩ fru P1* neurons impacts circadian period length and strength in DD.

We additionally annotate the aminergic clusters based on marker gene expression for genes that encode biosynthetic enzymes or transporters of specific aminergic transmitters, described below (*Figure 6B*, *Figure 6—figure supplement 1H, I*, *Source data 3*). Dopaminergic neurons were classified by marker gene expression of *Dopamine transporter* (*DAT*) and/or *pale* (*ple*). Serotonergic neurons were defined by maker gene expression of *Serotonin transporter* (*SerT*). Tyraminergic and octopaminergic were defined by marker gene expression of *Tyrosine decarboxylase 2* (*Tdc2*). Expression of these genes was also visualized at the single neuron level (*Figure 6—figure supplement 1A–C*), revealing that these genes are also expressed beyond where they are considered cluster marker genes.

## Dopaminergic populations of *fru P1* neurons

The three *Vmat* clusters from the full data set analysis have *DAT* and *ple* among their top enriched marker genes. *Dat* and *ple* are expressed >96% of neurons in these clusters, so these three clusters are annotated as dopaminergic neurons (clusters 58, 74, and 89, *Figure 6B*, *Figure 6—figure supplement 1C*, *Source data 3*). All three dopaminergic clusters annotated here have marker genes previously shown to be in adult midbrain and VNC dopaminergic neurons: *PDGP- and VEGF-related growth factor* (*Pvf3*), *kekkon 1* (*kek1*), and *homothorax* (*hth*) (*Allen et al., 2020*; *Croset et al., 2018*).

The three clusters do not have Hox marker genes that are expressed in the VNC, however a small population of neurons in clusters 74 and 89 have *Abd-B* expression (*Figure 6—figure supplement 1J*). The top marker genes are largely overlapping between the three clusters, with *Vmat*, *DAT*, and *ple* among the top 5 (*Figure 6C*). Clusters 58 and 74 also share the IgSF gene *DIP-delta* as a top non-unique marker gene (*Figure 6C*, *Source data 3*). We showed that *DIP-delta ∩ fru P1* neurons in the adult brain have restricted spatial expression pattern, with broader expression in the VNC (*Brovero et al., 2021*). This indicates that some *DIP-delta ∩ fru P1* neurons are likely dopaminergic and may be those previously visualized in the brain (*Brovero et al., 2021*). Further, when we examine *Vmat ∩ fru P1* expression patterns in the brain of both sexes (*Figure 6—figure supplement 1D–G*), their expression pattern is reminiscent of *DIP-delta ∩ fru P1* neurons, with projections around the mushroom body γ lobes (*Brovero et al., 2021*).

It is known that male dopaminergic *fru P1* neurons have a role in mating drive and copulation duration (*Jois et al., 2018*; *Zhang et al., 2016*), so we next evaluated male dopaminergic clusters identified in the male data set analysis (male clusters 56 and 67, *Figure 6—figure supplement 1H*, *Source data 3*). A small population of neurons in both clusters express *Abd-B* and *Ubx*, suggesting that they are from the VNC (*Figure 6—figure supplement 1K*), potentially capturing previously identified neurons involved in male copulation duration (*Jois et al., 2018*). Unlike the dopaminergic clusters identified in the full data set analysis, these two male clusters show little overlap in their marker genes with only five shared marker genes (*Source data 3*). To compare these clusters, we visualized expression of their top marker genes, revealing that cluster 56 has *DIP-beta* (*Figure 6—figure supplement 1L*, *Source data 3*). In addition, we find cluster 67 has *VGlut* as a top marker gene, suggesting that populations of male glutamatergic *fru P1* neurons are also dopaminergic (*Figure 6—figure supplement 1L*).

## Serotonergic populations of *fru P1* neurons

Next, to identify clusters with serotonergic neurons, we examined the clusters that have *SerT* as a marker gene in the full data set. We identify three serotonergic clusters (clusters 73, 86, and 92). However, all three clusters also contain a substantial number of neurons without *SerT* expression, with all clusters having less than 70% of their neurons expressing *SerT* (cluster 73: 26%, cluster 86: 26%, and cluster 92: 70%) (*Figure 6—figure supplement 1A–C*, *Source data 3*). These clusters have additional marker genes that have been previously identified in serotonergic neurons in the adult midbrain and VNC (*Allen et al., 2020*; *Croset et al., 2018*), which provides further evidence that they should be considered serotonergic clusters. All three serotonergic clusters have *IGF-II mRNA-binding protein* (*Imp*), and two have *Jim Lovell* (*lov*), as marker genes (*Croset et al., 2018*). We also find marker genes encoding TFs *ventral veins lacking* (*vvl*) and *Lim3* for clusters 86 and 92, as well as *juvenile hormone inducible 21* (*JhI-21*) as a marker gene in all three clusters. These three genes have previously been shown to be marker genes in adult VNC serotonergic neurons (*Allen et al., 2020*). Further, our data suggest that neurons from two serotonergic clusters have autoregulatory control, as one cluster has marker gene expression of 5HT receptors *5-hydroxytryptamine (serotonin) receptor 1*A and B (*5-HT1A* and *5-HT1B*, cluster 73) and another has *5-hydroxytryptamine (serotonin) receptor 2B* (*5-HT2B*, cluster 92) (*Source data 3*), as previously shown in adults (*Allen et al., 2020*).

To examine expression across these serotonergic clusters, we evaluate the top five marker genes for each serotonergic cluster. We find that all three have *SerT*, *Imp*, *Dopa decarboxylase* (*Ddc*), and *CG2269* as marker genes that are expressed in a high percentage of neurons (*Figure 6D*, *Source data 3*). The serotonergic clusters 86 and 92 share enriched expression of *Neuropeptide-like precursor 1* (*Nplp1*) and *VGlut*, suggesting that these serotonergic neurons are also glutamatergic and peptidergic (*Figure 6D*, *Source data 3*). This analysis also reveals specific and highly expressed markers for these neuron populations: cluster 86 expresses *CG15082* and cluster 92 expresses the neuropeptide-encoding gene *Allatostatin C* (*AstC*) (*Figure 6D*, *Source data 3*), indicating molecular differences.

Serotonergic *fru P1* neurons in the abdominal ganglion of the VNC are sexually dimorphic, with ~8–10 male-specific neurons in this region that have a role in sperm transfer (*Billeter and Goodwin, 2004*; *Lee and Hall, 2001*; *Lee et al., 2001*). To identify these neurons in our data set we searched for *SerT*-expressing neurons that also expresses *abd-A* and/or *Abd-B*, from the full data set analysis, resulting in 41 neurons (17 male and 24 female). To investigate sex-specific molecular differences, we examined sex-differentially expressed genes in these 41 neurons (*Source data 3*). *dsx* is a top marker gene in male neurons (*Source data 3*). 11 of the 17 male serotonergic abdominal ganglion neurons

identified here express *dsx*, whereas the 24 female cells exhibit no *dsx* expression (*Figure 6E*). *dsx* expression is required for the formation of male-specific *fru P1* serotonergic neurons in the abdominal ganglion (*Billeter et al., 2006*), some of which are involved in sperm transfer (*Tayler et al., 2012*), suggesting that we may have identified these male-specific neurons in our data set.

## Tyraminergic and octopaminergic populations of *fru P1* neurons

Tyraminergic and/or octopaminergic *fru P1* neurons have been shown to have a role in mate discrimination. Artificially activating octopaminergic *fru P1* neurons or using RNA-mediated interference against *fru P1* transcripts in octopaminergic neurons increases male–male courtship (*Certel et al., 2010*; *Certel et al., 2007*). The gene-encoding *tyrosine decarboxylase 2* (*Tdc2*), an enzyme involved in synthesizing the amino acid tyrosine to tyramine, is expressed in both tyraminergic and octopaminergic neurons (*Roeder, 2005*). We identify tyraminergic/octopaminergic neurons using the *Tdc2* marker gene. This results in two tyraminergic/octopaminergic clusters in the full data set and one in the female data set (*Source data 3*). *Tdc2* is not detected as a marker gene in the male data set but is expressed in a small population of male cells (*Figure 6—figure supplement 1A*). Interestingly, in the full data set, one of the tyraminergic/octopaminergic clusters, cluster 51, is also identified as a mushroom body KC cluster (*Source data 3*). This suggests that some previously identified *Tdc2*-expressing mushroom body neurons may also include *fru P1*-expressing neurons (*Aso et al., 2014*). However, to our knowledge, no reports of *fru P1* and *Tdc2* expression have been shown in the mushroom body.

We did not detect *Tdc2* ∩ *fru P1* neurons using the genetic intersectional strategy, with a recently published *Tdc2-Gal4* driver (*Deng et al., 2019*). There are previous reports of overlap based on Fru^M antibody staining and a different *Tdc2-Gal4* driver (*Certel et al., 2010*; *Certel et al., 2007*), suggesting that there may be limitations to our visualization approach. One additional possibility is that there are *Gal4* expression differences between these *Tdc2-Gal4* driver lines. Further, this may be a Fru^M-expressing population where *fru^{FLP}* is not expressed, given there are Fru^M-expressing neurons that are *fru^{FLP}* negative (*Yu et al., 2010*).

## *fru P1* neurons that produce both aminergic and FANs

We find that one male dopaminergic cluster and two serotonergic clusters have *VGlut* marker gene expression (*Figure 6D*, *Figure 6—figure supplement 1G*, *Source data 3*). This suggests that there are neurons that may be both aminergic and FAN producing. This prompted us to evaluate at the neuron level co-expression of genes indicative of biogenic amine and FAN production or transport (*Figure 6F–H*). Here, aminergic neurons were classified based on *DAT* and/or *ple* (dopaminergic), *SerT* (serotonergic), or *Tdc2* (tyraminergic/octopaminergic) expression and we asked about their overlapping expression with genes we previously used to identify FAN-producing neurons. For the three classes of aminergic neurons the majority are also cholinergic, based on their co-expression of either *ChAT* or *VAChT* (*Figure 6F–H*, green bars). The male tyraminergic/octopaminergic neurons have a higher proportion of glutamatergic (*VGlut*) co-expressing neurons compared to GABAergic (*VGAT* or *Gad1*), whereas the opposite is seen in females (*Figure 6H*). Overall, we find a range of *fru P1* aminergic neurons show co-expression with genes indicative of producing/transporting FANs (10–54% of aminergic neurons). This suggests that populations of *fru P1* neurons likely release at least two neurotransmitters.

## Identification of *fru P1* neurons that express circadian clock genes

Research of circadian/sleep behaviors have identified sexual dimorphisms (reviewed in *Andretic and Shaw, 2005*; *Ho and Sehgal, 2005*; *King and Sehgal, 2020*; *Shafer and Keene, 2021*), with *fru P1* neurons and mating behavior implicated in mediating sex differences (*Fujii and Amrein, 2010*; *Fujii et al., 2007*; *Hanafusa et al., 2013*; *Sakai and Ishida, 2001*; *Tauber et al., 2003*). GO enrichment analyses on marker genes identified two clusters in the full data set, clusters 108 and 109, enriched with genes that have circadian functions (*Figures 1C and 7A*). The marker genes include *period* (*per*), *timeless* (*tim*), *Clock* (*Clk*), *vrille* (*vri*), and *Pdp1-epsilon* (*Pdp1*) (*Source data 3*). We also annotated other clusters with circadian maker genes (*Source data 3*), but 108 and 109 are the only clusters with more than one marker gene with known circadian functions in the full data set. The male data set has one cluster with *per*, *tim*, *Clk*, *vri*, and *Pdp1* marker genes (*Source data 3*), whereas there are no female clusters with multiple marker genes with known circadian functions. In the full data set, cluster

109 is male biased in cell number (>twofold more after normalization), though both clusters 108 and 109 contain cells from males and females (*Source data 5*). An examination of the expression of *per*, *tim*, *Clk*, *vri*, and *Pdp1* in all neurons did not reveal any additional clusters that are likely to be involved in circadian functions that might have been missed when only examining marker genes in clusters (*Figure 7—figure supplement 1A–C*).

To further examine the neurons in clusters 108 and 109, we performed a subclustering analysis, resulting in three subclusters that each have male and female neurons (*Figure 7B, C*). Expression of *per*, *tim*, *Clk*, *vri*, and *Pdp1* is different in the three subclusters (*Figure 7—figure supplement 1D*), as is expression of the top five maker genes (based on log-fold-change in expression, *Figure 7D*), suggesting the three subclusters are comprised of different circadian regulatory neurons, or the neurons could have distinct gene expression profiles due to slight differences in time of day when the neurons were processed for 10× Genomics library preparation. Based on expression data from a recent scRNA-seq study of clock neurons (*Ma et al., 2021*), we postulate that subcluster 0 is most similar to DN1p neurons given this cluster has *gl*, *VGlut*, *Rh7*, and *Dh31* as marker genes (*Figure 7B*, *Source data 6*). We postulate that subcluster 1 has similar expression to DN1p neurons, due to having *Rh7* as a marker gene (*Kistenpfennig et al., 2017*; *Ma et al., 2021*). Consistent with these observations, DN1s have been shown to express Fru$^M$ in adults and DN1ps show a sex difference in cell number in adults (*Fujii and Amrein, 2010*; *Hanafusa et al., 2013*). Marker gene expression in subcluster 2 indicates that neurons in this cluster are similar to s-LNvs or l-LNvs clock neurons, because *cry* is a marker gene (*Yoshii et al., 2008*). Previous studies of developing clock neurons, based on *per* and *tim* expression, indicate that two DN1 neurons, four small ventral LNs (s-LNvs), and two large ventral LNs (l-LNvs) are present at 48-hr APF (*Kaneko et al., 1997*). However, we are not able to visualize DN1s or l-LNv neurons at 48-hr APF in either sex with Per antibody staining. We do identify four s-LNvs in both sexes with Per antibody staining at 48-hr APF, three of which overlap with *fru P1* expression (*Figure 7E–E''*, *Figure 7—figure supplement 1E–E'*). This supports that subcluster 2 may be comprised of developing s-LNvs. However, there is no expression of *pdf* in subcluster 2, leaving open the possibility that these neurons may be a different subpopulation of circadian neurons that are present in pupae (*Helfrich-Förster, 1995*; *Kaneko et al., 1997*).

## Functional studies of *fru P1* neurons that express circadian clock genes

To visualize circadian neurons that overlap with *fru P1* neurons, we use the genetic intersection approach to visualize *Clk856 ∩ fru P1* neurons in the CNS. The *Clk856-Gal4* transgene was recently used to classify clock neuron expression in adults, using scRNA-seq (*Ma et al., 2021*). We did not detect *Clk856 ∩ fru P1 >sm.GDP.Myc* neurons at 48-hr APF in either sex, perhaps due to a lag in reporter gene expression, as noted above. When we examined 0- to 24-hr adults, we find *Clk856 ∩ fru P1 >sm.GDP.Myc* neurons that have cell bodies positioned in the DN1 and DN3 circadian regions. Additional cell bodies are detected in suboesophageal zone (SEZ) with projections in the SEZ or in the median bundle, and cell bodies are in several regions of the VNC (*Figure 7F*). When we quantified the number of *Clk856 ∩ fru P1 >sm.GDP.Myc* cell bodies in the brain, we observe that males have significantly more DN3 neurons at 0–24 hr (*Source data 7*). The male-biased number of *fru P1*-expressing DN3s has not been previously described, to our knowledge (*Figure 7F*, *Source data 7*). We note that it is uncertain if DN3 neurons would be present in our 48-hr APF scRNA-seq data set, as DN3 neurons are thought to arise later in pupal development (*Kaneko et al., 1997*). In 4- to 7-day adults, this sex difference in the number of *Clk856 ∩ fru P1* DN3 neurons does not persist, but we find a larger number of DN3s in both males and females. Now, we find a significant male bias in the number of *Clk856 ∩ fru P1* DN1 neurons at 4–7 days. This is in agreement with previous studies that have quantified *fru*-expressing DN1s and found a larger number in males (*Figure 7—figure supplement 1F–I*, *Source data 7*; *Hanafusa et al., 2013*). In addition, in 4–7 day adults we find equal numbers of LNds in both sexes.

To examine the behavioral role of *Clk856 ∩ fru P1* neurons, we activated these neurons using the intersectional genetic approach, with *UAS<stop<TrpA1$^{Myc}$ (Clk856 ∩ fru P1>TrpA1* Myc) (*von Philipsborn et al., 2011*). The TrpA1 channel has been shown to be activated by temperature and downstream of Gq and phospholipase C (*Kang et al., 2011*; *Kim et al., 2010*; *Kwon et al., 2008*; *Luo et al., 2017*; *Roessingh and Stanewsky, 2017*; *Zhong et al., 2012*). We assayed the effect on locomotor activity patterns using the *Drosophila* activity monitor (DAM, Trikinetics) system, in 12-hr

light:12-hr dark (LD) conditions. The TrpA1 channel is Myc tagged, so we further visualized *Clk856 ∩ fru P1* neurons in 4–7 day adults. When we count the number of *Clk856 ∩ fru P1* neurons in both sexes, we find significantly more male DN1s, LNds, and both types of SEZ neurons, as compared to females (*Figure 7—figure supplement 1*, *Source data 7*). Generally, when we find these significant sex differences in neuron numbers in *Clk856 ∩ fru P1>TrpA1* Myc or *Clk856 ∩ fru P1>sm.GDP.Myc*, the effect size is small (*Source data 7*).

To examine locomotor activity, we raised flies at 19°C, a temperature where the TrpA1 channel is not activated by temperature, and recorded activity in the DAM system at 25°C, a temperature where the TrpA1 channel is activated by temperature (*von Philipsborn et al., 2011*). When we compare activity between the sexes across genotypes, we find that the wild-type control, Canton S (CS), males are significantly more active than females (*Figure 7—figure supplement 2A*). This is not observed in the experimental comparisons of *Clk856 ∩ fru P1>TrpA1* Myc males and females, where the trend is that females are more active (*Figure 7—figure supplement 2A*). To evaluate activity across the 24-hr day, we examined actograms for all genotypes (*Figure 7—figure supplement 2B, C*). *Clk856 ∩ fru P1>TrpA1* Myc males exhibit little daytime activity (*Figure 7—figure supplement 2B*). *Clk856 ∩ fru P1>TrpA1* Myc females show robust activity in the early afternoon compared to all other genotypes (*Figure 7—figure supplement 2C*). This suggests that the *Clk856 ∩ fru P1* neurons have sleep-reducing roles in females, and sleep-promoting roles in males. These neurons may normally direct the dimorphism in daytime sleep, which has been referred to as the male daytime siesta (*Andretic and Shaw, 2005*; *Ho and Sehgal, 2005*). A previous study has also shown that DN1 neurons have a role in promoting the daytime siesta (*Guo et al., 2016*), and this population of neurons are activated in our *Clk856 ∩ fru P1* intersection.

Next, we determined if *Clk856 ∩ fru P1* neurons had a role in regulating circadian period. To test this, we conducted the DAM assay in 12-hr dark:12-hr dark (DD) conditions after entraining the flies in LD. Consistent with the LD data, actograms containing 2 days of LD data show reduced daytime activity in *Clk856 ∩ fru P1>TrpA1* Myc males and increased activity in the early afternoon in *Clk856 ∩ fru P1>TrpA1* Myc females (*Figure 7—figure supplement 3A, B*). Remarkably, these actograms also show that *Clk856 ∩ fru P1>TrpA1* Myc males have an observable activity shift over the 10 DD days, indicative of a shortening circadian period (*Figure 7—figure supplement 3A, B*). To further evaluate the circadian period in DD for all genotypes, we examined periodograms, period peaks, and period strength (*Figure 7—figure supplement 3C–E*). Strikingly, this analysis reveals a shortened period in *Clk856 ∩ fru P1>TrpA1* Myc males (mean period = 22.8 hr), and a longer period in *Clk856 ∩ fru P1>TrpA1* Myc females (mean period = 24.6 hr), compared to control genotypes (*Figure 7—figure supplement 3C, D*). When we examine period strength, we find that *Clk856 ∩ fru P1>TrpA1* Myc also exhibit the strongest period strength compared to all other genotypes. Altogether, we find that activating *Clk856 ∩ fru P1>TrpA1* Myc neurons has sexually dimorphic effects on circadian period, with males having a shortened period and females having a lengthened period. These circadian sex differences with *Clk856 ∩ fru P1>TrpA1* activation could be due to quantitative sex differences in *Clk856 ∩ fru P1* neuron number, sex differences in their connectivity, or sex differences in their intrinsic physiology.

## Neuropeptide and neuropeptide receptor expression in *fru P1* neurons

Out of the 49 annotated neuropeptide-encoding genes in the *Drosophila* genome (*Larkin et al., 2021*), we find that 47 genes are expressed in *fru P1* neurons in the full data set analysis (*Figure 8A*), and 18 genes are identified as marker genes (*Source data 3*). In addition, 64% of the clusters in the full data set (72 clusters) have at least one neuropeptide-encoding gene as a marker gene (*Source data 3*). Here, we focus on those with known roles in reproductive behaviors in females and males.

We first characterize clusters that express genes that encode neuropeptides with known roles in female-specific reproductive behaviors. For example, *Insulin-like peptide 7* (*Ilp7*)-expressing neurons are involved in oviduct function, and *Diuretic hormone 44* (*Dh44*)-expressing neurons have a role in sperm ejection (*Castellanos et al., 2013*; *Garner et al., 2018*; *Lee et al., 2015*). We find *Ilp7* is a marker gene in two clusters, in all three data sets (*Figure 8B*, *Figure 8—figure supplement 1A, B*). *Ilp7* has been shown to be expressed in *fru P1* neurons in both sexes, with female-specific neurons identified in the abdominal ganglion of the VNC, due to cell death in males (*Garner et al., 2018*). However, neither cluster with *Ilp7* as a marker gene in the full data set is female-specific nor shows a

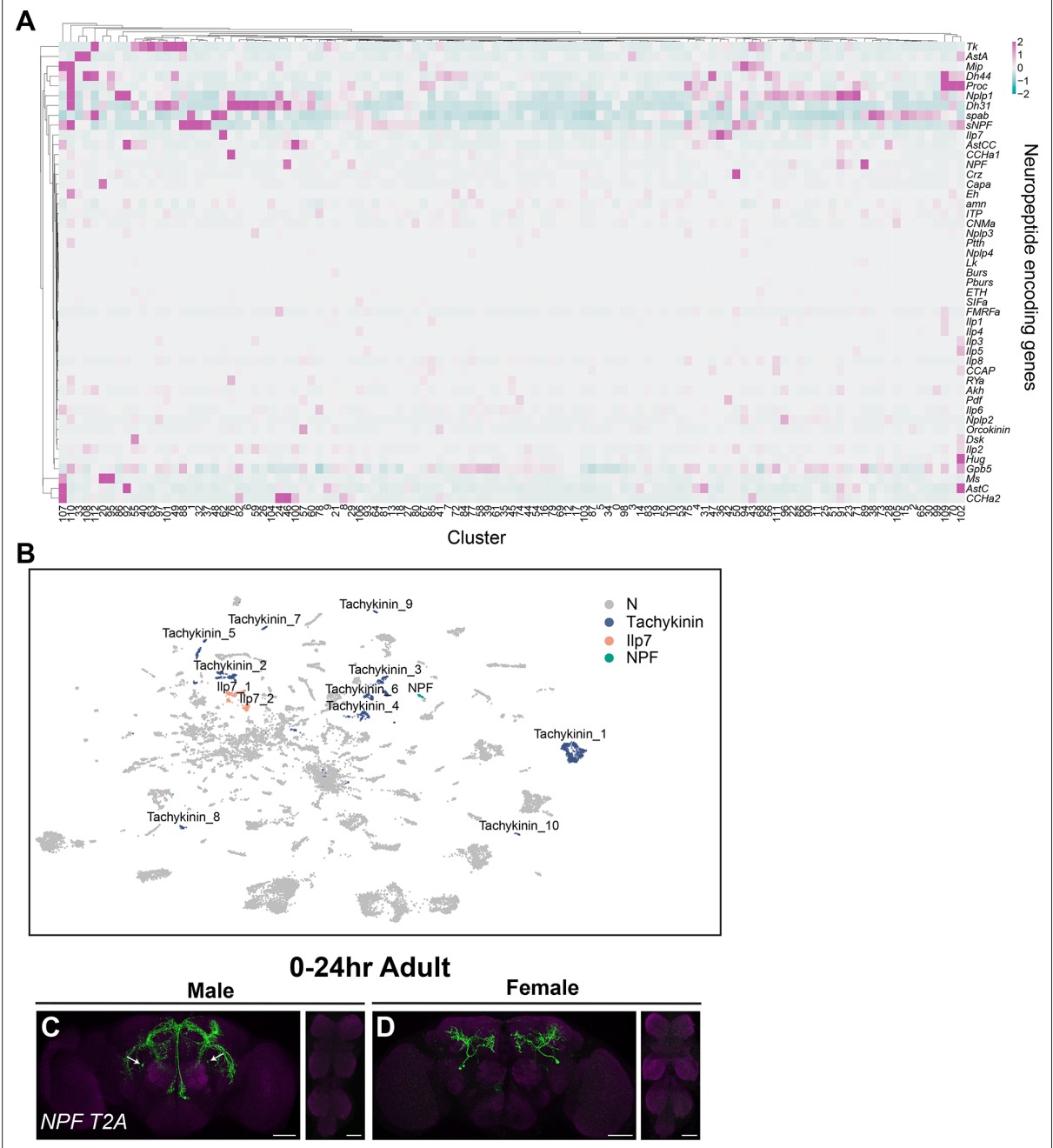

**Figure 8.** Annotation of neurons that express neuropeptides. (**A**) Heatmap of log-normalized gene expression for neuropeptide-encoding genes. Gene expression values were mean scaled and log normalized. (**B**) Uniform Manifold Approximation and Projection (UMAP) plot showing annotated clusters with neurons that produce neuropeptides with known roles in *fru P1* neurons for directing reproductive behavior, in the full data set (*Source data 3*). (**C, D**) Confocal maximum projections of brains and ventral nerve cord (VNC) showing *NPF ∩ fru P1* expression in 0- to 24-hr adult males and females, using an *NPF-T2A-Gal4* driver (*Deng et al., 2019*). White arrows on male brain indicate male-specific *NPF ∩ fru P1* neurons that have been previously identified (*Liu et al., 2019*). Scale bars = 50 µm.

The online version of this article includes the following figure supplement(s) for figure 8:

**Figure supplement 1.** Annotation of neurons that express neuropeptides in males and females.

sex bias in cell number (*Source data 5*), so it is possible that we did not identify the female-specific *Ilp7* neurons. *Dh44* has been shown to have a role in female sperm storage post-mating, with six cholinergic *Dh44*-expressing neurons in the Pars intercerebralis of the female brain shown to have a role in sperm ejection behavior (*Lee et al., 2015*). We find *Dh44* is a marker gene in two clusters in

both the male and female data sets and six clusters in the full data set (*Source data 3*). In the full data set, three of these clusters with *Dh44* as a marker gene also have marker genes that indicated they are cholinergic (*ChAT* and/or *VAChT* are marker genes, *Source data 3*). This previous work showed co-expression of *fru P1* with the gene that encodes the *Dh44* receptor (*Dh44-R1*) but did not examine if *Dh44*-expressing neurons were *fru P1* expressing (*Lee et al., 2015*). Using a *T2A-Gal4* knock-in in the *Dh44* gene and the genetic intersectional approach (*Deng et al., 2019*), we visualized the spatial expression of *Dh44 ∩ fru P1* neurons, finding expression at 48-hr APF and in 0- to 24-hr adults in both sexes (*Figure 8—figure supplement 1C–F*). In 0- to 24-hr adults, this *Dh44 ∩ fru P1* expression pattern includes neurons in the Pars intercerebralis, which may contain the neurons implicated to have a role in female sperm ejection. *fru P1* neurons that produce neuropeptides also have known roles in male courtship, copulation, and aggression behaviors (*Asahina et al., 2014*; *Liu et al., 2019*; *Tayler et al., 2012*; *Wohl et al., 2020*; *Wu et al., 2019*). For example, a subset of male-specific *Tackykinin* (*Tk*) and *fru P1* co-expressing neurons in the brain promote male aggression, though a larger number of *Tk*-expressing neurons have been identified in both sexes (*Asahina et al., 2014*; *Nässel et al., 2019*; *Wohl et al., 2020*). Here, we find that *Tk* is a marker gene in 10 clusters in the full data set, and five in both the male and female data set analyses (*Figure 8B*, *Figure 8—figure supplement 1A, B*, *Source data 3*). This suggests that beyond the male-specific *Tk-Gal4* and *fru P1* co-expressing neurons that have been well characterized for their role in aggression (*Asahina et al., 2014*; *Wohl et al., 2020*), there are additional *Tk*-expressing neurons that co-express *fru P1* in both sexes.

Male-specific *Neuropeptide F* (*NPF*) and *fru P1* co-expressing neurons have a role in regulating courtship drive, by inhibiting courtship in sexually satiated males (*Liu et al., 2019*). We annotate two clusters where *NPF* is a marker gene in both male and full data set analyses (*Figure 8B*, *Figure 8—figure supplement 1A*, *Source data 3*). *NPF* is not a marker gene in the female data set (*Source data 3*). To spatially visualize *NPF ∩ fru P1* neurons, we used the genetic intersection approach with an *NPF T2A-Gal4* knock-in (*Deng et al., 2019*). There were no visible *NPF ∩ fru P1* neurons at 48-hr APF in either sex. In 0- to 24-hr adult brains, we identify dramatically different sexually dimorphic neuronal projections, with some cell bodies in similar positions (*Figure 8C, D*). Based on location of the neurons, we potentially find the previously described male-specific *NPF* neurons in the brain using the intersectional approach (*Figure 8C, D*, white arrows). However, the *NPF ∩ fru P1* expression pattern differs from previous reports that used different *NPF-Gal4* or *LexA* insertions (*Liu et al., 2019*). Additionally, when we examined expression of all genes that encode neuropeptides, we find additional clusters that express the neuropeptides *Corazonin* (*Crz*) and *Drosulfakinin* (*Dsk*), which have been shown to be expressed in subpopulations of *fru P1* neurons and have impacts on male behaviors, but are not considered marker genes in our data sets (*Figure 8A*) (sperm transfer, *Tayler et al., 2012*; courtship inhibition, *Wu et al., 2019*).

Neuropeptide receptors are also widely expressed in *fru P1* neurons, with 104 clusters having at least one gene encoding a neuropeptide receptor as a marker gene in the full data set (*Source data 3*). Previous studies have found that *fru P1* neurons that express neuropeptide receptors have roles in male courtship behaviors. These include *Corazonin receptor* (*CrzR*) (sperm transfer, *Tayler et al., 2012*), *Cholecystokinin-like receptor at 17D3* (*CCKLR-17D3*) (courtship inhibition, *Wu et al., 2019*), and *SIFamide receptor* (*SIFaR*) (increased male–male courtship, *Sellami and Veenstra, 2015*). In addition, the neuropeptide receptor *Leucine-rich repeat-containing G protein-coupled receptor 3* (*Lgr3*) has been shown to expressed in *fru P1* neurons and have a role in female behavior (female receptivity and fecundity, *Meissner et al., 2016*). All these neuropeptide receptors are found as marker genes in our data set (*Source data 3*). We visualized *CrzR ∩ fru P1* and *Lgr3 ∩ fru P1* neurons using *T2A-Gal4* knock-in driver lines (*Deng et al., 2019*; *Figure 8—figure supplement 1G–N*). *CrzR ∩ fru P1* neurons in the abdominal ganglia have a known role in male copulation behavior (*Tayler et al., 2012*). We find that these neurons are in similar positions as previously shown in males and additionally visualize sexually dimorphic neurons in the brain as well as a small set of neurons in the female abdominal ganglia (*Tayler et al., 2012*; *Figure 8—figure supplement 1G–J*). The 48-hr APF *Lgr3 ∩ fru P1* neurons show expected median bundle expression, with more GFP-expression in females compared to males, consistent with *Lgr3* expression being regulated by Fru$^{M}$ (*Figure 8—figure supplement 1K–L*; *Meissner et al., 2016*). However, in 0- to 24-hr adult males, we observe male-specific *Lgr3 ∩ fru P1* neuronal projections in the lateral protocerebral complex and in the VNC that were not previously reported (*Figure 8—figure supplement 1M–N*), which may be due to differences in visualization

strategies. Given that 109 of the 113 clusters in the full data set express at least one neuropeptide or neuropeptide receptor as a marker gene, indicates that many *fru P1*-expressing neurons use neuropeptides for signaling (*Source data 3*).

## *nAChR* receptor subunit genes expressed in *fru P1* neurons

We find nicotinic acetylcholine receptor (nAChR) subunits are marker genes for the majority of clusters in the full data set (91 clusters out of 113) (*Source data 3*). Nicotinic acetylcholine receptor are pentameric receptors that assemble with combinations of 10 subunits [seven α nAChR subunits (1–7) and three β nAChR subunits (1–3)] (*Rosenthal and Yuan, 2021*). Of these 10 *nAChR* subunit genes, we find that nine are marker genes and all 10 are expressed in the full data set (*Source data 3*). Furthermore, they are broadly expressed with at least 97% of *fru P1* neurons expressing at least one *nAChR* receptor. Out of the 10 subunits, *nAChRα5* is the most widely expressed in all three data sets and *nAChRβ3* has the narrowest expression (*Source data 8*). The different subunit combinations confer different physiological binding properties (*Perry et al., 2021*). To evaluate subunit co-expression,

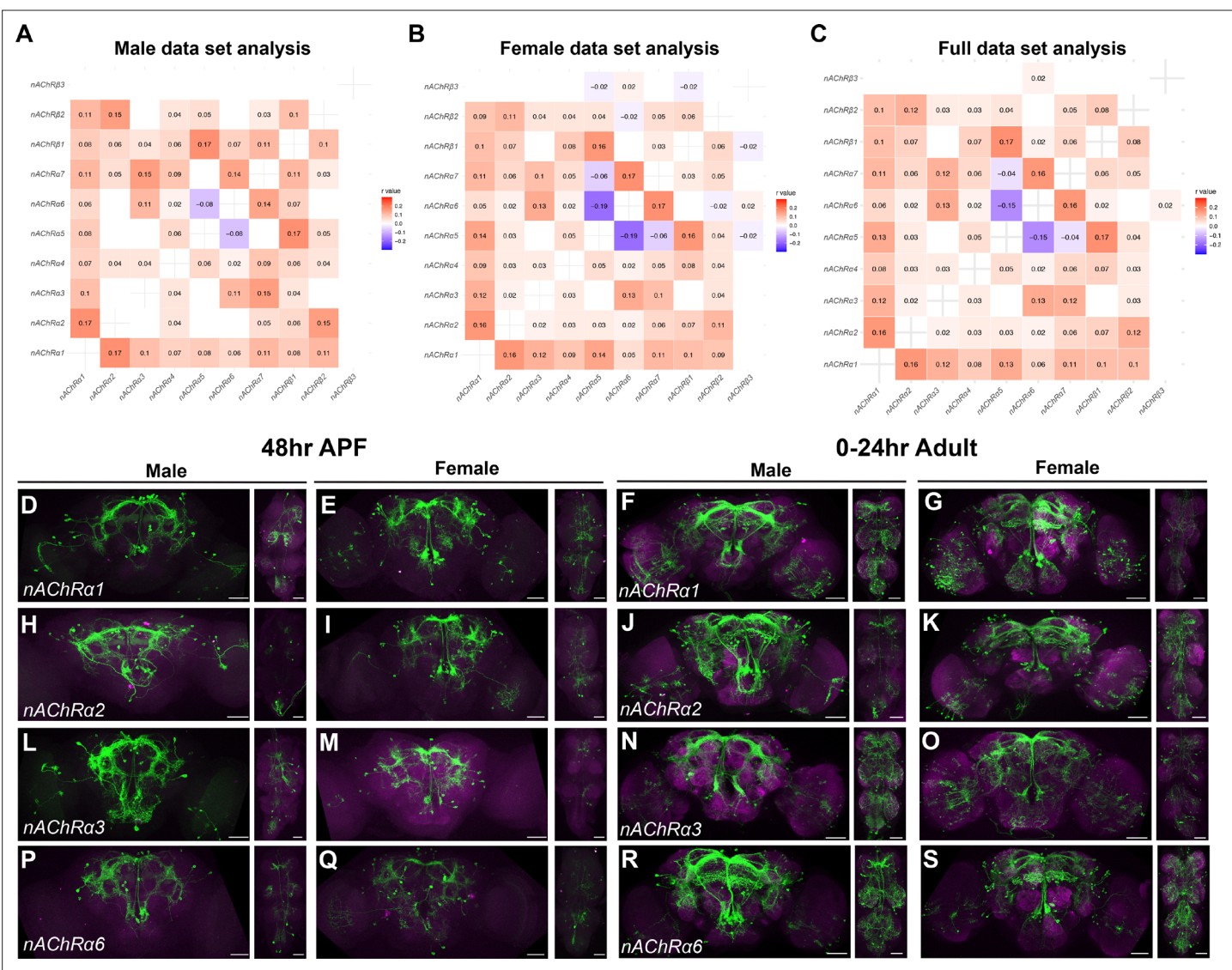

**Figure 9.** *nAChR* subunit gene expression in *fru P1* neurons. Heatmap showing correlation of *nAChR* subunit gene expression in neurons in the male data set (**A**), female data set (**B**), and full data set (**C**). Pearson's *r* values denoted on heatmap with color, according to legends. (**D–S**) Confocal maximum projections of brain and ventral nerve cord (VNC) for *nAChRα ∩ fru P1*-expressing neurons, with intersecting expression shown in green for males and females, as indicated. Male and female animals from 48-hr after puparium formation (APF) and 0- to 24-hr adults are shown, as indicated. Scale bars = 50 μm.

we examined the pairwise expression correlation between the 10 subunits for the three data set analyses and show significant correlations (*Figure 9A–C*). For all data sets, we find that the overall correlation trends are similar, with some additional significant correlations observed for the female data set (*Figure 9A–C*). We find the highest positive expression correlations are between: *nAChRα5* and *nAChRβ1*, *nAChRα6* and *nAChRα7*, and *nAChRα1* and *nAChRα2* for the full data set analysis (*Figure 9C*). In contrast to the analysis of subunit correlations performed in the adult mid-brain data set, we find anti-correlations between subunit expression data (*Figure 9A–C*; *Croset et al., 2018*). These negative correlations occur between *nAChRα5* and *nAChRα6* and *nAChRα5* and *nAChRα7*, with the strongest negative correlations in the female data set analysis (*Figure 9A–C*). Given that *nAChRα5* and *nAChRα6* have the strongest expression correlation in the adult midbrain data set and we observe this pair as having the strongest expression anti-correlation points to the importance of examining different neuronal populations using single-cell approaches.

We next visualized the spatial expression patterns for some *nAChRα* that have available Gal4 lines. We examined nAChRα1, nAChRα2, nAChRα3, and nAChRα6 ∩ *fru P1* neurons and find broad expression at 48-hr APF and in 0- to 24-hr adults of both sexes, which is consistent with our scRNA-seq expression data (*Figure 9D–S*). All *nAChR* ∩ *fru P1* spatial populations examined have similar expression patterns within each sex, which is consistent with their correlated co-expression (*Figure 9D–S*). The *nAChR* ∩ *fru P1* spatial populations are in all segments of the VNC in both sexes (*Figure 9D–S*). We also find that spatial expression of *nAChRα1*, *nAChRα2*, and *nAChRα6* ∩ *fru P1* neurons in 0- to 24-hr adult male brains contain previously characterized sexually dimorphic projections in the tritocerebral loop (*Figure 9D–S*). Notably, we find that *nAChRα3* ∩ *fru P1* expression is absent from the mushroom body in both sexes, whereas we observe mushroom body expression for all other *nAChR*s ∩ *fru P1* examined (*Figure 9D–S*). We also find that *nAChRα3* is not a marker gene in mushroom body clusters identified in our data sets (*Source data 3*), consistent with previous reports indicating its absence from the mushroom body (*Croset et al., 2018*; *Shih et al., 2019*). Our expression findings here, and the identification of several *nAChR*s in genomic studies examining regulation by Fru[M], expression in *fru P1* neurons, and expression changes post-mating, suggest that *nAChR*s are important for *Drosophila* reproductive behaviors (*Brovkina et al., 2021*; *Dalton et al., 2013*; *Neville et al., 2014*; *Newell et al., 2016*; *Newell et al., 2020*; *Palmateer et al., 2021*; *Vernes, 2014*).

## *Wnt* expression in *fru P1* neurons

Wnts have been shown to have a range of function in neurons, including axon guidance, synapse formation, neuronal plasticity, morphogenesis, and signaling in several species (*He et al., 2018*; *Kolodkin and Tessier-Lavigne, 2011*; *Salinas and Zou, 2008*; *Sanes and Yamagata, 2009*). Here, we further examine *Wnt* expression, given the GO enrichment analysis of marker genes in the full data

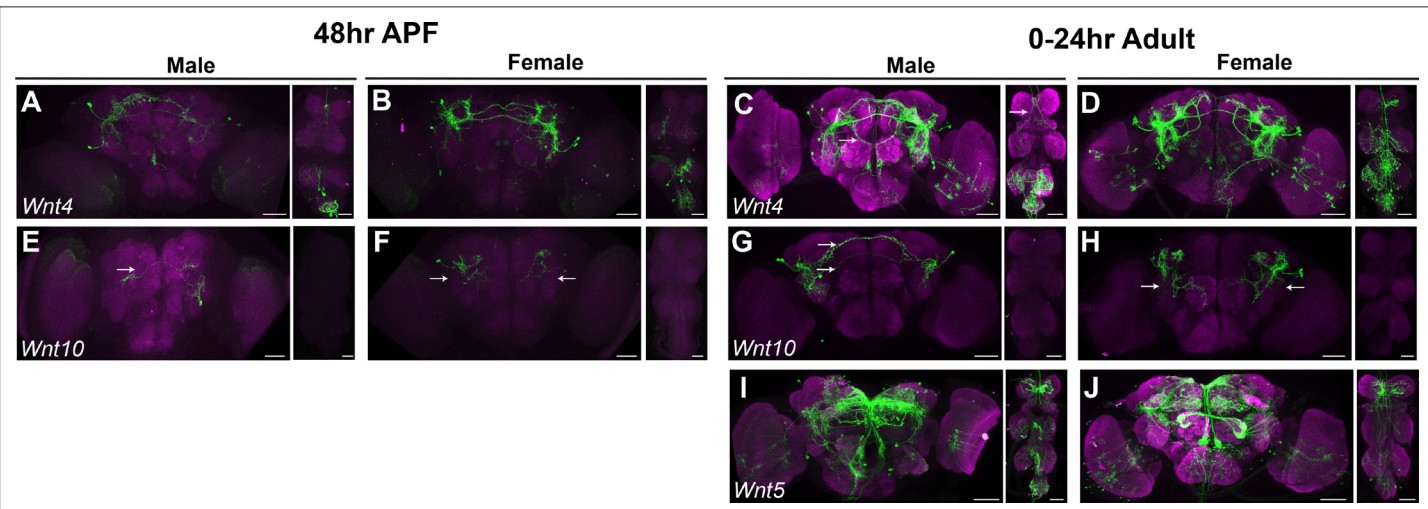

**Figure 10.** *Wnt* ∩ *fru P1* expression patterns. (**A–J**) Brain and ventral nerve cord (VNC) confocal maximum projections showing intersecting expression patterns (in green), for males and females, as indicated. The patterns of *Wnt4* ∩ *fru P1*, *Wnt10* ∩ *fru P1*, and *Wnt5* ∩ *fru P1* neurons are shown. Scale bars = 50 μm. The *Wnt5* stock was not available to perform expression analyses at 48-hr after puparium formation (APF) due to the stock being unhealthy.

set identified several wingless pathway terms including: 'WG ligand binding', 'wingless pathway', 'Beta-catenin independent Wnt signaling', and 'signaling by Wnt' (*Source data 4*). Furthermore, our previous genome-wide association study showed four Wnt pathway signaling genes had allele variants associated with differences in female remating behavior (*Newell et al., 2020*), adding further impetus to evaluate the Wnt pathway. We found three *Wnt* genes were marker genes, *Wnt4* (14 clusters), *Wnt5* (2 clusters), and *Wnt10* (4 clusters) (*Source data 3*).

We visualized the spatial expression patterns for *Wnt4 ∩ fru P1* and *Wnt10 ∩ fru P1* neurons and observe several sexual dimorphisms in their neuronal projection patterns at 48-hr APF and in 0- to 24-hr adults of both sexes (*Figure 10A–H*). We additionally visualized *Wnt5 ∩ fru P1* in 0- to 24-hr adults (*Figure 10I, J*). In 0- to 24-hr adult males, *Wnt4 ∩ fru P1* neuron projections look similar to previously identified sexually dimorphic vPR1/vPr-a ascending neurons (*Figure 10C*, white arrows), which have been suggested to have a role in sensory integration and motor output to control wing song (*Cachero et al., 2010*; *Yu et al., 2010*). *Wnt10 ∩ fru P1* neurons are observed only in the brain, and not the VNC, and show sexually dimorphic projections (*Figure 10E–H*, white arrows). Male *Wnt10 ∩ fru P1* neurons project across hemispheres at both 48-hr APF and 0- to 24-hr stages (*Figure 10E, G*, white arrows), whereas in female brains no projections between the brain hemispheres were observed (*Figure 10F, H*, white arrows). Sexually dimorphic midline crossing of gustatory sensory neurons was previously observed in the VNC and regulated by *fru* and *dsx* (*Mellert et al., 2010*; *Possidente and Murphey, 1989*). The observation that *Wnt10 ∩ fru P1* neurons also have sexually dimorphic midline crossing in the brain suggest this may be a mechanism to generate sex differences in behavior. In 0- to 24-hr adults, *Wnt5∩ fru P1* shows broader expression in the brain than *Wnt4 ∩ fru P1* and *Wnt10 ∩ fru P1* neurons, and also exhibits expression in all VNC segments (*Figure 10I, J*).

## Sex differences in gene expression between male and female *fru P1* neurons

We next examined differential expression between the sexes within individual cluster in the full data set analysis. For each gene, the expression in male neurons within a cluster was compared with expression in female neurons within the same cluster. This differential expression analysis reveals 6162 genes are significantly differentially expressed between males and females, at the cluster level (3504 higher in males, 2658 higher in females). When we reduce these gene lists to remove redundant genes (the same genes are differentially expressed in more than one cluster) and compare between males and females, we find 711 genes uniquely male biased in expression (always show higher expression in males within at least one cluster), 586 are uniquely female biased in expression (always show higher expression in females within at least one cluster) (*Source data 9*). There are 147 genes that are either male- or female biased, depending on the cluster (*Source data 9*). This analysis shows that the same genes are re-used in different neuronal clusters to impart sexual dimorphism, given the redundancy in the gene lists found across the clusters. When we compare these sex-biased genes to genes identified as sex biased in our previous cell-type-specific TRAP study at 48-hr APF, we find significant overlap between gene lists (*Source data 3*; *Palmateer et al., 2021*). One limitation to this analysis is if the clusters are comprised of more than one neuronal type, as this may impact the ability to detect sex-differential expression due to averaging the expression. We provide GO enrichment analysis of these lists (*Source data 4*); several of the enriched GO categories motivated the follow-up investigations presented below.

## Immunoglobulin-like domain superfamily (IgSF) expression in *fru P1* neurons

A Flymine analysis of the marker gene lists to find 'protein domain' enrichments identify 'Immunoglobulin-like domain superfamily' with the most significant enrichment in each of the three data sets (*Source data 4*; *Lyne et al., 2007*). In the full data set analysis, we find that all clusters have at least one marker gene that is a member of the Immunoglobulin-like domain superfamily, with most expressing a large and unique repertoire (*Source data 3*). Using a heatmap we visualized expression of IgSF-encoding genes, which shows the average expression of a gene in a cluster, regardless of marker gene status (*Figure 11*). Consistent with what we observe when we examine marker genes (*Source data 3*), each cluster exhibits unique expression combinations of IgSF-encoding genes (*Figure 11*).

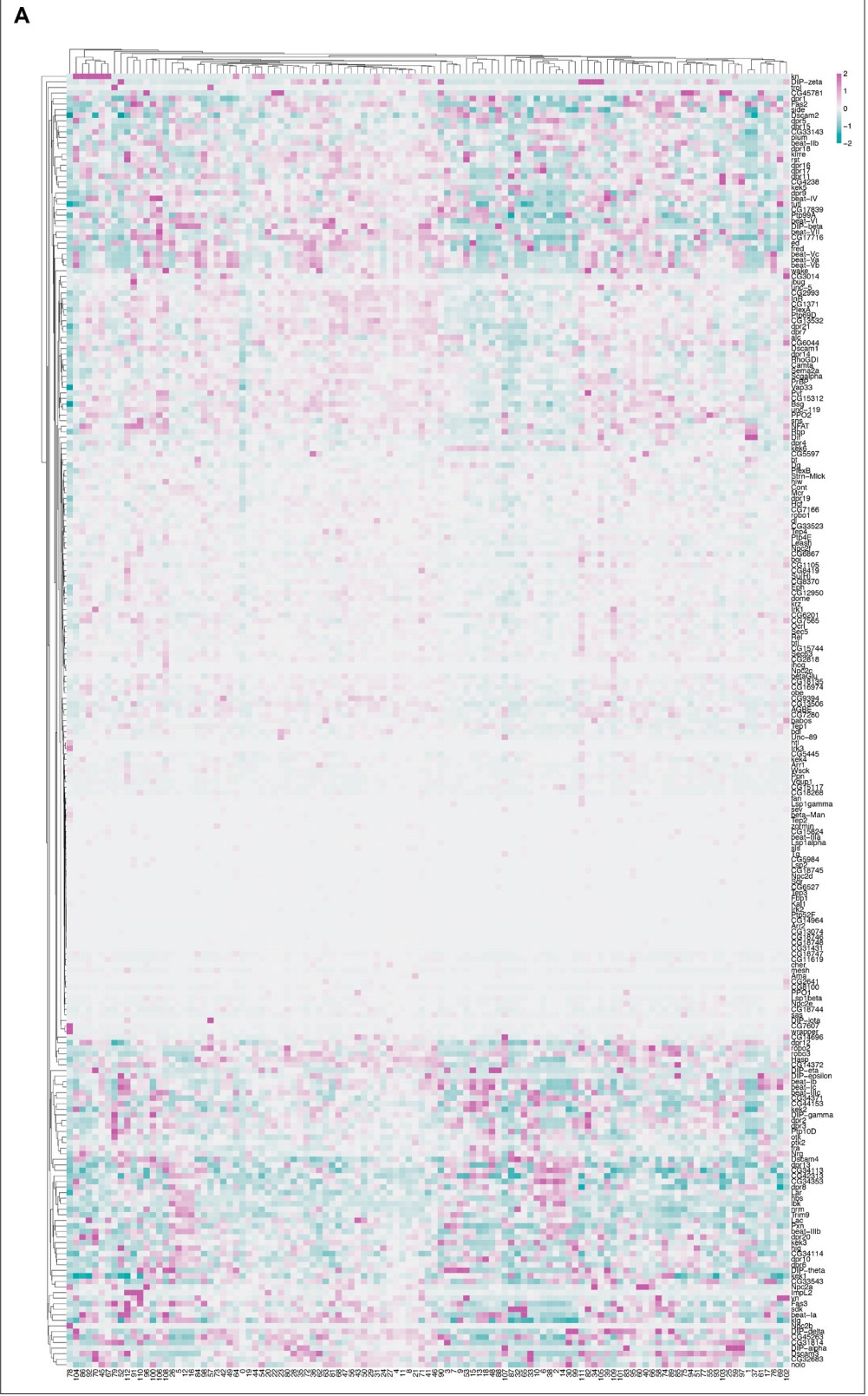

**Figure 11.** IgSF gene expression in *fru P1* neurons. (**A**) Heatmap of log-normalized gene expression for IgSF-encoding genes. Gene expression values were mean scaled and log normalized.

The online version of this article includes the following figure supplement(s) for figure 11:

**Figure supplement 1.** Sex differences in *dpr/DIP* gene expression in *fru P1* neurons.

Previously we examined expression of *dpr* and *DIP* genes in the male CNS from 48-hr APF using 10× single-cell genomic approaches, with *fru P1* neurons identified based on expression of *fru P1* and/ or a GFP reporter (*Brovero et al., 2021*). Similar to our previous study on males, we find that in both sexes the majority of *dpr*s show broad expression, whereas *DIP*s exhibit more restricted expression (*Figure 11—figure supplement 1*). With the addition of female data, we next determine if there are *dpr* and *DIP* expression differences between the sexes, for each cluster of the full data set analysis (*Figure 11—figure supplement 1*). This analysis revealed that there are sex differences in *dpr* and *DIP* expression for several clusters in our data set (*Figure 11—figure supplement 1*, *Source data 9*). Of the *dpr*s and *DIP*s that show a significant sex bias in expression within clusters, some show expression that is always higher in female neurons (uniquely female biased; *dpr*s *1*, *5*, *8*, *10*, *11*, *12*, *14*, *15*, *16*, *17*, and *20*, *DIP-alpha* and *DIP-beta*), whereas fewer show expression that is always higher in males (uniquely male biased; *dpr*s *7*, *18*, and *21*, *DIP*s *gamma*, *theta*, and *zeta*, *Figure 11—figure supplement 1*, *Source data 9*). In contrast, *dpr*s *2*, *3*, *6*, *9*, and *13* and *DIP-delta* and DIP-*eta* show cluster dependent differences in their significant sex-biased expression (*Figure 11—figure supplement 1*, *Source data 9*). Future in vivo expression analyses are needed to confirm these quantitative differences. Differences in *dpr/DIP* co-expression within individual *fru P1* neurons may offer a mechanism to generate different cell adhesion properties to mediate synaptic connections, as we previously proposed (*Brovero et al., 2021*).

## Transcription factor expression in *fru P1* neurons

During the stage of metamorphosis that we examined, cell fate decisions are being directed by transcription factor (TF) expression, with different combinations of TFs found in different neuronal populations (*Doe, 2017*; *Shirasaki and Pfaff, 2002*). Consistent with this, the top 'molecular function' GO enrichment terms for marker genes across all three data set analyses is 'DNA-binding transcription factor activity' (*Figure 1—figure supplement 6*, *Source data 4*). Out of the 628 *Drosophila* genes encoding TFs (Flybase), 240 are marker genes in the male data set analysis, 228 in the female data set analysis, and 253 in the full data set analysis (*Source data 3*). We next determined the enriched TF subcategories in Flymine, evaluating subcategories that contained more than 25 genes (*Lyne et al., 2007*). This identified three subcategories of TF marker genes: homeobox-like domain superfamily (74 genes), zinc finger C2H2 (74 genes), and helix–loop–helix (HLH) DNA-binding domain superfamily (25 genes). We next examined if certain subcategories have restricted or broad expression, to gain insight into which TFs might impart cluster-specific identities.

Most of the homeobox-like domain superfamily TF-encoding genes, including *abd-A*, *Abd-B*, and *slouch* (*slou*), show cluster specificity, with high expression in a large percent of neurons in the cluster, in only a subset of clusters (*Figure 12—figure supplement 1A*).There are some homeobox TF-encoding genes, such as *pipsqueak* (*psq*), *extradenticle* (*exd*), and *CG16779*, that are expressed more broadly, in nearly all clusters (*Figure 12—figure supplement 1A*), On the other hand, zinc finger C2H2 TF-encoding genes, which includes *fru*, has most genes displaying broad expression (*Figure 12— figure supplement 2*). Though, some zinc finger C2H2 TFs exhibit cluster-specific expression, such as *buttonhead* (*btd*), which is predominately expressed in cluster 101 (*Figure 12—figure supplement 2*). Genes that encode HLH DNA-binding domain superfamily TFs also have several showing broad expression across clusters (*Figure 12—figure supplement 3*), and others with more restricted expression. These analyses suggest that the homeobox-like domain superfamily may have a large role in directing differences in cell types among the *fru P1* neurons, given the overall number of genes with large differences in expression across the clusters (*Figure 12—figure supplements 1–3*).

We next determined if the 253 marker genes that encode TFs display significant sex-differential expression. There are 69 genes that encode TFs with cluster-specific sex-differential expression. Seventeen genes are uniquely male biased, 38 are uniquely female biased, and 8 are either male- or female-biased depending on the cluster (*Source data 9*). We used dot plots to visualize the expression of these TFs with cluster-specific differential expression between the sexes (*Figure 12—figure supplement 4A*), though future in vivo validation is needed to confirm these quantitative differences. This shows that 49 clusters exhibit significant sex-differential expression of at least one marker gene TF, with several clusters showing differential expression of multiple TFs (*Figure 12—figure supplement 4A*, black circles). We also visualized expression of the top three sex-differentially expressed TFs in each sex, based on log fold-change between the sexes (*Figure 12A*). The top three differentially

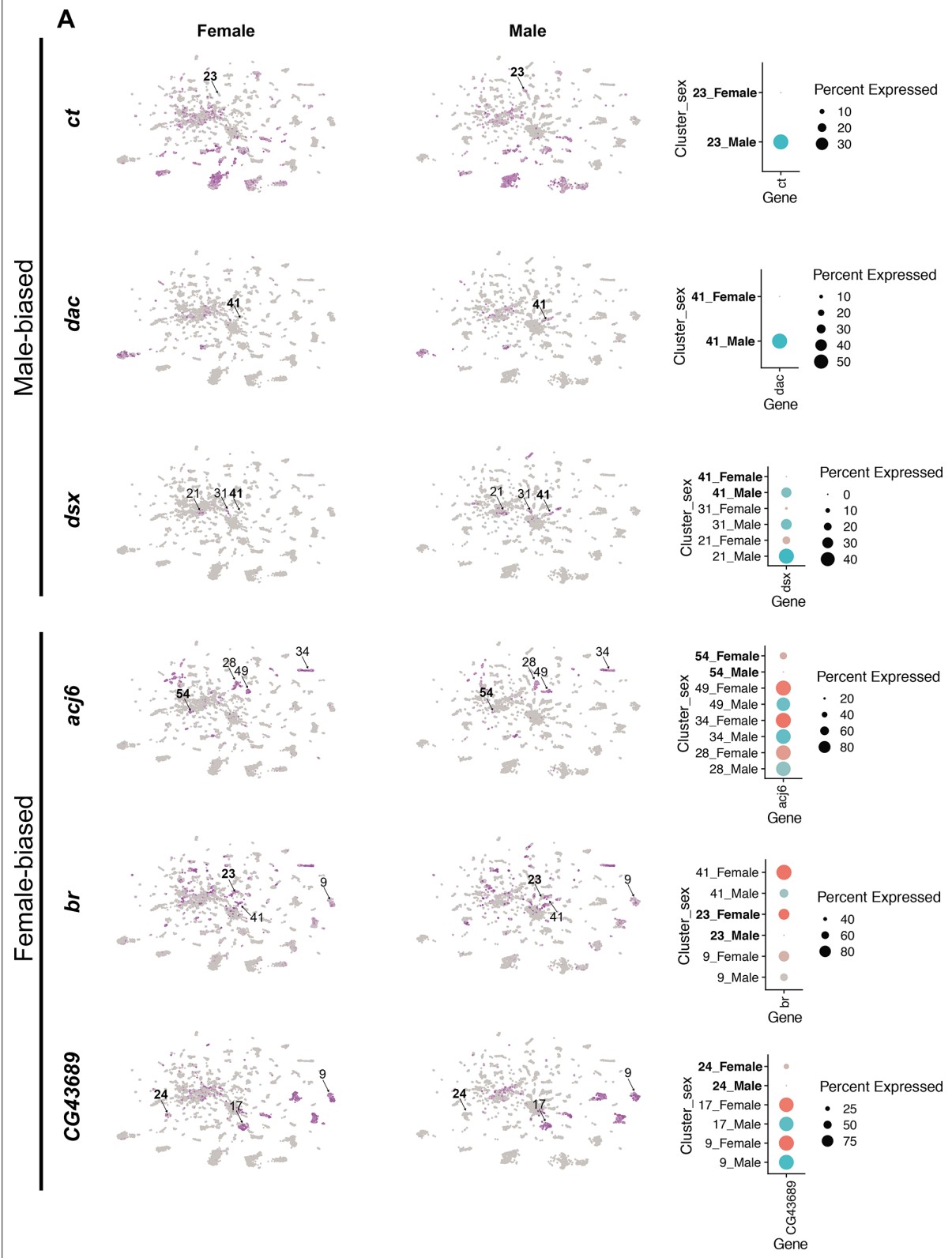

**Figure 12.** Sex differences in transcription factor (TF) expression in *fru P1* neurons. (**A**) Gene expression feature plots showing gene expression in neurons for TFs that have significant sex-biased gene expression within single-cell clusters between female (left; top three) and male (right; top three). Top sex-biased TFs determined by log FC in *Source data 9*. The top three male-biased TFs shown are: *ct, dac*, and *dsx*. The top three female-biased TFs shown are: *acj6, br*, and *CG43689*. Bold cluster numbers in the dot plot indicate the cluster where the gene had the sex-differential expression

*Figure 12 continued on next page*

*Figure 12 continued*
detection in the top three for each sex. A dot plot representation for all TF genes that have cluster-specific sex-biased marker gene expression is shown in *Figure 12—figure supplement 4*.

The online version of this article includes the following figure supplement(s) for figure 12:

**Figure supplement 1.** Homeobox-like domain superfamily transcription factor (TF) expression in *fru P1* neurons.

**Figure supplement 2.** Zinc finger C2H2 transcription factor (TF) expression in *fru P1* neurons.

**Figure supplement 3.** Helix–loop–helix (HLH) DNA-binding domain superfamily expression in *fru P1* neurons.

**Figure supplement 4.** Sex differences in transcription factor (TF) expression in *fru P1* neurons.

expressed TFs with male-biased expression are: *cut* (*ct*), *dachshund* (*dac*), and *dsx*. Those with female-biased expression are: *abnormal chemosensory jump 6* (*acj6*), *broad* (*br*), and *CG43689* (***Source data 9***). We show all clusters with significant sex bias in expression for these TFs using UMAP featureplots and dot plots for visualization (***Figure 12***). This differential TF expression between the sexes is likely a mechanism to direct sex differences in *fru P1* neurons.

## Conclusions

We performed scRNA-seq on pupal *fru P1* neurons from males and females to understand how sex differences are built into the nervous system to direct adult reproductive behaviors. The full data set analysis revealed the heterogeneity of *fru P1* neurons at 48-hr APF with 113 different molecular clusters identified. This exceeds the number of previously designated spatial cluster and morphological classifications. This suggests that there are more functional differences in neurons than predicted by Fru^M/*fru P1* spatial patterns (***Billeter and Goodwin, 2004***; ***Cachero et al., 2010***; ***Lee et al., 2000***; ***Manoli et al., 2005***; ***Stockinger et al., 2005***; ***Yu et al., 2010***). The analyses also revealed that nearly all clusters have neurons from both males and females. This suggests that male and female neurons that direct vastly different reproductive behaviors, have a core gene expression program that are overlaid with sex differences in gene expression. This is consistent with our previous TRAP-seq data sets, where we postulated that shared molecular properties underlie common behavioral needs. We proposed that sex differences in gene expression are overlaid on a sex-shared baseline gene expression program to direct male and female behavioral differences (***Newell et al., 2016***; ***Palmateer et al., 2021***). We also identify sex-specific clusters in our data set. We find a male-specific *dsx*-expressing cluster, which is consistent with the known male-specific *fru P1* and *dsx* co-expressing neurons (***Billeter et al., 2006***; ***Ishii et al., 2020***; ***Rideout et al., 2007***; ***Sanders and Arbeitman, 2008***), and we validate a newly identified female-specific *CCHa2*-expressing cluster (***Figure 2***, ***Figure 2—figure supplement 2***). Our results also point to sex differences in neuron number across the clusters, which is consistent with previous in vivo studies (for characterization of all *fru P1* neurons see ***Cachero et al., 2010***; ***Manoli et al., 2005***; ***Stockinger et al., 2005***; ***Yu et al., 2010***).

Using the enriched marker genes, we annotate 46 clusters, and provide information for all the clusters based on the distribution of the marker genes, with comparisons to other single-cell *Drosophila* studies and in-depth examinations of different GO groupings (***Source data 3***). For example, we use Hox gene expression to annotate clusters from the VNC (***Figure 3***, ***Source data 3***). We used known gene expression profiles to categorize *fru P1* neurons as γ and αβ KCs of the mushroom body, including some newly identified molecular subtypes (***Figure 4***, ***Source data 3***). The marker gene analyses led to the identification of two clusters that have circadian clock neurons, which we further subcluster to reveal additional populations by using known gene expression signatures from a previous single-cell RNA-seq studies (***Figure 7***; ***Ma et al., 2021***). We perform functional analysis of these *fru P1* circadian neurons and show that they have dimorphic roles in period length (***Figure 7—figure supplement 3***). The analyses of marker genes also show that neuropeptides (64% of clusters) and neuropeptide receptors (92% of clusters) genes have enriched expression in the majority of *fru P1* neurons, (***Figure 8***, ***Source data 3***), and indicate the importance of these signaling molecules in reproductive behaviors. Our previous GWAS analyses of natural variation of female remating found *Wnt* pathway genes contributing to differences (***Newell et al., 2020***). The *Wnt* pathway marker gene identification led to the identification of *Wnt* ∩ *fru P1* neurons, with sexual dimorphism in their projections (*Wnt 4*, *Wnt 5*, and *Wnt 10*; ***Figure 10***). The top protein domain enrichment across the marker genes was the 'Immunoglobulin-like domain superfamily', consistent with our identification of the IgSF genes being

expressed in *fru P1* neurons in our previous studies (see *Brovero et al., 2021*; *Goldman and Arbeitman, 2007*). An examination of the marker gene distribution and the overall expression levels shows that all clusters display a unique repertoire of IgSF genes (*Source data 3*; *Figure 11*). The diversity of IgSFs across the clusters may be a way to generate neurons with different adhesion properties to ultimately generate different neuronal connections. Another functional category enriched in the marker genes was 'DNA-binding transcription factor activity', with homeobox-like domain TF genes showing the largest expression differences across the clusters. We also annotate the clusters with respect to the neurotransmitters they produce, based on marker gene expression and overall gene expression levels (fast-acting and biogenic amines). One key finding is that 80% of the clusters have *nAChR* subunit genes as marker genes (*Figure 9*), and most *fru P1* neurons express at least one *nAChR* subunit gene, with *nAChRalpha5* expressed in 97% of neurons.

For many of the marker gene analyses, we provide in vivo spatial links that support the conclusions. Additional in vivo validation is certain to produce interesting new findings contributing new insights into the molecular-genetic and circuit basis of behaviors and will provide valuable information about how expression in scRNA-seq studies correlates with in vivo mRNA, protein, and reporter gene patterns. Our study used careful clustering and filtering criteria consistent with published *Drosophila* single-cell studies (*Allen et al., 2020*; *Croset et al., 2018*; *Davie et al., 2018*; *Konstantinides et al., 2018*; *Kurmangaliyev et al., 2020*; *Li et al., 2017*; *Li et al., 2022*; *McLaughlin et al., 2021*; *Özel et al., 2021*; *Simon and Konstantinides, 2021*; *Xie et al., 2021*). We expect new computational algorithms will likely improve single-cell data analyses. For example, after we started our analyses a new doublet detection algorithm (DoubletFinder) was released (*McGinnis et al., 2019*), which we show found additional potential doublets in our data set throughout the clusters (*Source data 2*). The data set will provide opportunities to further our biological understanding of behaviors and can also be reanalyzed to provide additional molecular-genetic insights as new algorithms are generated.

# Materials and methods

**Key resources table**

| Reagent type (species) or resource | Designation | Source or reference | Identifiers | Additional information |
|---|---|---|---|---|
| Genetic reagent (*D. melanogaster*) | fru P1-Gal4 | Provided by Baker lab PMID:15959468 | | fruP1-GAL4 |
| Genetic reagent (*D. melanogaster*) | Canton S | Ulrike Heberlein | Wild type | +/+;+/+;+/+ (Canton S) |
| Genetic reagent (*D. melanogaster*) | fru P1[FLP] | Donor: Barry Dickson, Howard Hughes Medical Institute, Janelia Research Campus | RRID:BDSC66870 | w[*]; TI{FLP}fru[FLP]/T+D6:D27+D6:D18M3, Sb[1] |
| Genetic reagent (*D. melanogaster*) | 10XUAS-IVS-mCD8::GFP | Donor: Gerald M. Rubin & Barret Pfeiffer, Howard Hughes Medical Institute, Janelia Research Campus | RRID:BDSC_32186 | w[*]; P{y[+t7.7] w[+mC]=10XUAS-IVS-mCD8::GFP}attP40 |
| Genetic reagent (*D. melanogaster*) | 10xUAS >stop > myr::smGdP-cMyc | Pfeiffer, B., Rubin, G. (2014.4.16). Recombinase and tester constructs and insertions. | RRID:BDSC_62125 | w[1118]; P{y[+t7.7] w[+mC]=10xUAS(FRT.stop)myr::smGdP-cMyc}attP40 |
| Genetic reagent (*D. melanogaster*) | UAS >stop > TrpA1Myc | Barry Dickson Stocks. Flybase ID: FBrf0234603 | RRID:BDSC_66871 | P{UAS(FRT.stop)TrpA1[myc]}VIE-260B |
| Genetic reagent (*D. melanogaster*) | Lgr3-Gal4 | Donor: Bruce Baker, Howard Hughes Medical Institute, Janelia Research Campus; Donor's Source: James Truman, Howard Hughes Medical Institute, Janelia Research Campus | RRID:BDSC_66683 | w[*]; P{y[+t7.7] w[+mC]=Lgr3-GAL4::VP16}attP40/CyO |
| Genetic reagent (*D. melanogaster*) | NPF-Gal4 | Donor: Paul Garrity, Brandeis University & Bowen Deng, National Institute of Biological Sciences; Donor's Source: Yi Rao, National Institute of Biological Sciences | RRID:BDSC_84671 | w[*]; TI{2 A-GAL4}NPF[2 A-GAL4]/TM6B, Tb[1] |
| Genetic reagent (*D. melanogaster*) | Wnt10-Gal4 | Donor: Hugo J. Bellen, Baylor College of Medicine; Donor's Source: Gene Disruption Project | RRID:BDSC_86484 | y[1] w[*]; TI{GFP[3xP3.cLa]=CRIMIC.TG4.2}Wnt10[CR01661-TG4.2]/SM6a |

*Continued on next page*

*Continued*

| Reagent type (species) or resource | Designation | Source or reference | Identifiers | Additional information |
|---|---|---|---|---|
| Genetic reagent (*D. melanogaster*) | Wnt4-Gal4 | Donor: Gene Disruption Project; Donor's Source: Hugo J. Bellen, Baylor College of Medicine | RRID:BDSC_67449 | y[1] w[*]; Mi{Trojan-GAL4.2}Wnt4[MI03717-TG4.2] |
| Genetic reagent (*D. melanogaster*) | Wnt5-Gal4 | Donor: Huey Hing, State University of New York, Brockport | RRID:BDSC_59034 | w[*] TI{w[+mC]=GAL4}Wnt5[Gal4] |
| Genetic reagent (*D. melanogaster*) | Vmat-Gal4 | Donor: Gene Disruption Project; Donor's Source: Hugo J. Bellen, Baylor College of Medicine | RRID:BDSC_66806 | y[1] w[*]; Mi{Trojan-GAL4.0}Vmat[MI07680-TG4.0] |
| Genetic reagent (*D. melanogaster*) | ChAT-Gal4 | Donor: Paul Garrity, Brandeis University & Bowen Deng, National Institute of Biological Sciences; Donor's Source: Yi Rao, National Institute of Biological Sciences | RRID:BDSC_84618 | TI{2 A-GAL4}ChAT[2 A-GAL4]/TM3, Sb[1] |
| Genetic reagent (*D. melanogaster*) | CrzR-Gal4 | Donor: Paul Garrity, Brandeis University & Bowen Deng, National Institute of Biological Sciences; Donor's Source: Yi Rao, National Institute of Biological Sciences | RRID:BDSC_84621 | w[*]; TI{2 A-GAL4}CrzR[2A-B.GAL4]/TM3, Sb[1] |
| Genetic reagent (*D. melanogaster*) | CCHa2-Gal4 | Donor: Paul Garrity, Brandeis University & Bowen Deng, National Institute of Biological Sciences; Donor's Source: Yi Rao, National Institute of Biological Sciences | RRID:BDSC_84602 | w[*]; TI{2 A-GAL4}CCHa2[2 A-GAL4]/TM3, Sb[1] |
| Genetic reagent (*D. melanogaster*) | Dh44-Gal4 | Donor: Paul Garrity, Brandeis University & Bowen Deng, National Institute of Biological Sciences; Donor's Source: Yi Rao, National Institute of Biological Sciences | RRID:BDSC_84627 | w[*]; TI{2 A-GAL4}Dh44[2 A-GAL4]/TM3, Sb[1] |
| Genetic reagent (*D. melanogaster*) | nAChRα1 Gal4 | Donor: Paul Garrity, Brandeis University & Bowen Deng, National Institute of Biological Sciences; Donor's Source: Yi Rao, National Institute of Biological Sciences | RRID:BDSC_84662 | TI{2 A-GAL4}nAChRalpha1[2A-AC.GAL4]/TM6B, Tb[1] |
| Genetic reagent (*D. melanogaster*) | nAChRα2 Gal4 | Donor: Paul Garrity, Brandeis University & Bowen Deng, National Institute of Biological Sciences; Donor's Source: Yi Rao, National Institute of Biological Sciences | RRID:BDSC_84663 | w[*]; TI{2 A-GAL4}nAChRalpha2[2 A-GAL4]/TM3, Sb[1] |
| Genetic reagent (*D. melanogaster*) | nAChRα3 Gal4 | Donor: Paul Garrity, Brandeis University & Bowen Deng, National Institute of Biological Sciences; Donor's Source: Yi Rao, National Institute of Biological Sciences | RRID:BDSC_84664 | w[*] TI{2 A-GAL4}nAChRalpha3[2 A-GAL4] |
| Genetic reagent (*D. melanogaster*) | nAChRα6 Gal4 | Donor: Paul Garrity, Brandeis University & Bowen Deng, National Institute of Biological Sciences; Donor's Source: Yi Rao, National Institute of Biological Sciences | RRID:BDSC_84665 | TI{2 A-GAL4}nAChRalpha6[2 A-GAL4]/CyO |
| Genetic reagent (*D. melanogaster*) | VGAT-Gal4 | Donor: Paul Garrity, Brandeis University & Bowen Deng, National Institute of Biological Sciences; Donor's Source: Yi Rao, National Institute of Biological Sciences | RRID:BDSC_84696 | TI{2 A-GAL4}VGAT[2 A-GAL4]/CyO |
| Genetic reagent (*D. melanogaster*) | VGlut-Gal4 | Donor: Paul Garrity, Brandeis University & Bowen Deng, National Institute of Biological Sciences; Donor's Source: Yi Rao, National Institute of Biological Sciences | RRID:BDSC_84697 | TI{2 A-GAL4}VGlut[2 A-GAL4]/CyO |
| Genetic reagent (*D. melanogaster*) | Clk856-Gal4 | Obtained from Michael Rosbash | | w; CLK856-Gal4/Cyo; MKRS/TM6B |

*Continued on next page*

*Continued*

| Reagent type (species) or resource | Designation | Source or reference | Identifiers | Additional information |
|---|---|---|---|---|
| Antibody | Rat polyclonal | *Sanders and Arbeitman, 2008* | RRID:AB_2569440 | anti-FruM; 1:200 in TNT |
| Antibody | Rabbit polyclonal | Obtained from Michael Rosbash | RRID:AB_2315101 | anti-Per; 1:500 in TNT |
| Antibody | Rabbit polyclonal | Abcam | RRID:AB_307014 | anti-Myc; 1:6050 in TNT |
| Antibody | Mouse monoclonal | Developmental Studies Hybridoma Bank | RRID:AB_2314866 | anti-NC82; 1:20 in TNT |
| Antibody | Mouse monoclonal | Developmental Studies Hybridoma Bank | RRID:AB_528440 | anti-Prospero(MR1A); 1:100 in TNT |
| Antibody | Rabbit polyclonal | Invitrogen | RRID:AB_221477 | anti-GFP 488; 1:600 in TNT |
| Antibody | Goat polyclonal | Invitrogen | RRID:AB_141373 | anti-rat 488; 1:1000 in TNT |
| Antibody | Goat polyclonal | Invitrogen | RRID:AB_10563566 | anti-rabbit 568; 1:500 in TNT |
| Antibody | Goat polyclonal | Invitrogen | RRID:AB_2535719 | anti-mouse 633; 1:500 in TNT |
| Software, algorithm | Zen | Carl Zeiss | RRID:SCR_013672 | |
| Software, algorithm | Illustrator | Adobe | RRID:SCR_010279 | |
| Software, algorithm | Seurat | *Stuart et al., 2019* | RRID:SCR_016341 | |
| Software, algorithm | Cell Ranger | 10 x Genomics | RRID:SCR_017344 | |
| Software, algorithm | GraphPad Prism | GraphPad | RRID:SCR_002798 | |
| Software, algorithm | ShinyR-DAM | *Cichewicz and Hirsh, 2018* | | |

## Fly strains and husbandry

Flies for scRNA-seq experiments were the genotype: *w[*]; P[y[+t7.7] w[+mC]=10XUAS-IVS-mCD8::GF] attP40/UAS-Gal4;fru P1-Gal4/+*. This strain expressed membrane-bound GFP in *fru P1* neurons in males and females. The flies were sex sorted at the white pre-pupal stage using the male gonad to distinguish between the sexes. We note that the *UAS-Gal4* present in this genotype was determined to be non-functional (*Feng and Cook, 2017*). The pupae were aged to 48 hr APF on 3% agar plates with Bromophenol blue for color contrast. All strains, unless otherwise indicated, are aged in a humidified incubator at 25°C on a 12-hr light and 12-hr dark cycle. Additional fly strains used in this study are listed in the Resources Table. The laboratory *Drosophila* media composition is: 33 l H$_2$O, 237 g agar, 825 g dried deactivated yeast, 1560 g cornmeal, 3300 g dextrose, 52.5 g Tegosept in 270 ml 95% ethanol and 60 ml propionic acid.

## Dissociation of pupal CNS for single-cell mRNA sequencing

The CNS (brain and VNC) from 20 freshly dissected male or female pupae were used per replicate (*n* = 2 replicates per sex). The dissected tissue was dissociated as previously described (*Brovero et al., 2021*). All dissections occurred within an hour after the lights turned on (ZT0). GFP positive cells (*fru P1*-expressing neurons) were enriched using FACS with a FACSAria SORP with BD FACSDiva *software*.

## 10× genomics library preparation and sequencing

scRNA-seq libraries were generated using Single Cell 3′ Library & Gel Bead Kit v3, Chip B Kit, and the GemCode 10× Chromium instrument (10× Genomics, CA), according to the manufacturer's protocol (*Zheng et al., 2017*). In brief, FACS enriched *fru P1*-expressing neurons were suspended in S2 medium with 10% fetal bovine serum and the maximum volume of cell suspension (46.6 µl) was mixed with single-cell Master Mix and loaded into an individual chip channel for each sex and replicate. For all samples, this process occurred between 4 and 5 hr after lights on (ZT4–ZT5). The library preparation steps were conducted as previously described, with modifications to polymerase chain reaction (PCR) thermal cycling steps for v3 chemistry (*Brovero et al., 2021*). For cDNA amplification PCR settings were: 98°C for 3 min, 12 cycles of (98°C for 15 s, 63°C for 20 s), 72°C for 1 min, held at 4°C. For sample index PCR amplification the settings were: 98°C for 45 s, 16 cycles of (98°C for 20 s, 54°C for 30 s, 72°C for 20 s), 72°C for 1 min, and hold at 4°C. All single-cell libraries were sequenced on an Illumina NovaSeq 6000 at Florida State University Translational Core Laboratory.

## scRNA-seq data processing and clustering in Seurat

CellRanger software (v.3.0.2) 'cellranger count' command was used to align sequencing reads to a customized *Drosophila melanogaster* (BDGP6.92) STAR reference genome, which contained the sequence for *mCD8-GFP* cDNA to generate a multidimensional feature-barcode matrix for each replicate. Using the inflection point of the barcode-rank plots, the CellRanger pipeline called the number of cells captured in each replicate (*Source data 1*). We captured 26,149 total cells across both sexes and replicates with 871,092,958 sequencing reads, reaching an average of 65% sequencing saturation (*Source data 1*). The feature-barcode matrices were next analyzed in Seurat (v3.2.2) (*Stuart et al., 2019*). Each expression matrix was filtered to obtain high-quality cells using the following criteria: cells with >5% mitochondrial transcripts (stressed/dead/dying cells), <200 genes (empty droplets), those expressing more than 4000 features (genes) and/or possessing more than 20,000 UMIs (potential doublets or triplets) were removed in each replicate. In addition, we provide the number of cells per replicate identified by DoubletFinder (version 2.0.3) in *Source data 2*. The male replicates and the female replicates were merged to form sex-specific data sets and additionally all replicates for both sexes were merged into a full data set. This yielded 7988 male cells, 17,530 female cells and 25,518 cells in total. We used default normalization and scaling steps for each data set as outlined in the Seurat 'Guided clustering tutorial' (https://satijalab.org/seurat/v3.2/pbmc3k_tutorial.html). The default 'NormalizeData' function was used to normalize the data and the data were subsequently scaled using the default 'ScaleData' function. We performed a principal component analysis (PCA) using the top 2000 highly variable features in each data set. We calculated statistically significant principal components (PCs) for each data set based on Jackstraw analysis. We used the number of significant PCs, up to where the first non-significant PC was met, for each data set (*Figure 1—figure supplement 2*). This resulted in 80, 104, and 107 PCs. We proceeded to use Seurat's standard workflow to reduce dimensionality and cluster cells by default 'FindNeighbors', 'FindClusters', and 'runUMAP' functions. We compared a range of cluster resolutions for each data set using the clustree R package (*Zappia and Oshlack, 2018*) before selecting an optimal resolution for each (resolution = 3.0, 0.7, and 1.2, *Figure 1—figure supplements 3 and 4*). To further validate the clustering resolutions chosen, we examined expression of *DIP-iota* in our data set, which we have previously shown to be expressed in a small subset of adult *fru P1* neurons (*Brovero et al., 2021*). We find that *DIP-iota* expression is restricted to a small subset of neurons in our UMAP plots for each data set (*Figure 1—figure supplement 8*), and is considered a significant marker gene for one cluster in the male and full data sets (*Source data 3*). We examined expression in the pupal CNS and find one neuron present in the male and two in the female VNC (*Figure 1—figure supplement 8*).

We also performed an analysis where we combined the male and female data sets using Seurat data integration to generate an integrated full data set (*Figure 1—figure supplement 5*). We used default 'FindIntegrationAnchors' and 'IntegrateData' steps as outlined in the Seurat 'Introduction to scRNA-seq data integration' tutorial (https://satijalab.org/seurat/articles/integration_introduction.html). Consistent with our initial merged full data set analysis, we performed PCA using the top 2000 highly variable features in each data set and calculated statistically significant PCs based on Jackstraw analysis. We selected the number of significant PCs, up to where the first non-significant PC was met, resulting in 125 significant PCs for this analysis (*Figure 1—figure supplement 5A*). We proceeded through the standard Seurat workflow as presented above using 'FindNeighbors', 'FindClusters', and 'runUMAP' functions, selecting an optimal cluster resolution that also produced a similar number of clusters to our merged full data set (resolution = 1.8). We find that the integration methodology performs marginally better in terms of replicate balance within cluster, though there are still clusters that have replicate contributions from only one technical replicate (integrated analysis: seven clusters; merged analyses: nine clusters). Given our biological question of comparing the expression of *fru P1*-expressing neurons between the sexes, we proceeded with the merged data in this study, as the integrated approach may limit true differences.

## Marker gene identification and sex-differential expression

Marker genes were identified per cluster for each data set using the Seurat 'FindAllMarkers' function, using the Wilcox rank sum test (min.pct = 0.25, logfc.threshold = 0.25). For each gene, the expression in a given cluster was compared with expression in neurons in the remaining clusters. Differentially expressed genes between the sexes per cluster were identified using the Seurat 'FindAllMarkers'

function, using the Wilcox rank sum test (min.pct = 0.25, logfc.threshold = 0.25). For each gene, the expression in a male neurons within a cluster was compared with expression in female neurons within the same cluster. The p values for marker gene and sex-differential expression were adjusted for multiple testing using the Bonferroni method and all results are presented in *Source data 3*. A marker was considered a significant marker gene if it was present in the marker gene list based on these criteria: min.pct = 0.25, logfc.threshold = 0.25, p_val <0.05.

## Sex differences in cell number within clusters

A cluster was considered 'sex-specific' if only cells from one sex were present in that cluster. Given the data set contains an unbalanced number of cells between the sexes in the full data set, the female cell numbers per cluster were divided by a scaling factor of 2.19. For a cluster to be considered 'sex-biased', cells from one sex were required to be >twofold higher in that cluster. For a cluster to be considered 'sex-biased, strong' the number of cells was >fourfold higher.

## Downsampling and random cell removal analyses

To perform random cell removal analyses, a random subset of 7988 female cells (to equal the number of male cells) was selected by the 'sample' R function. The Seurat 'Dimplot' function was used to visualize these data sets and the number of cells per sex in each cluster was quantified. To perform random downsampling analyses, a random subset of 7988 female cells were selected from the 17,530 filtered female cells using the 'sample' R function, as above.

The random subset of female cells was then merged with the Seurat object containing the 7988 filtered male cells into a new Seurat object using the 'merge' function. This was performed three times to create three independent random downsampled data sets. Each downsampled data set underwent the Seurat workflow described above. Consistent with our analysis of the full data set, we used the number of significant PCs up to where the first non-significant PC was met. This resulted in 99, 98, and 99 PCs. We chose cluster resolutions that yielded ~113 clusters, the number resolved in the analysis of the full data set, resulting in resolutions of 3.0, 3.0, and 3.1. To match these down sampled clusters with those in the full data set, we used the R package clustifyr (*Fu et al., 2020*). The 'seurat_ref' function was used to make a cluster reference from the full data set and the default 'clustify' wrapper function was used to perform Spearman correlations between the reference downsampled analysis clusters (*Source data 5*).

## Immunostaining and microscopy

Adult and 48-hr APF brains were dissected and imaged as previously described (*Palmateer et al., 2021*). Primary antibodies used were as follows: rat α-Fru$^M$ (1:200) (*Sanders and Arbeitman, 2008*), rabbit α-Myc (1:6050; abcam, ab9106), rabbit α-Per (1:500) (gift from Michael Rosbash), mouse α-Nc82 (1:20, DSHB), and mouse α-Prospero (1:100, DSHB). The secondary antibodies were as follows: goat α-rat 488 (1:1000, Invitrogen A11006), goat α-rabbit 568 (1:500; Invitrogen, A11036), rabbit α-GFP 488 (1:600; Invitrogen A21311), and goat α-mouse 633 (1:500; Invitrogen, A21052). Both primary and secondary antibodies were diluted in TNT (Tris-NaCL-Triton buffer; 0.1 M Tris–HCl, 0.3 M NaCl, 0.5% Triton X-100). All confocal microscopy was performed on a Zeiss LSM 700 system, with Zeiss Plan-Apochromat ×20/0.8 and ×0/1.4 objectives. The z-stack slice interval for all images was 1.0 μm. A 1 Airy Unit (AU) pinhole size was calculated in Zeiss Zen software (Black edition, 2012) for each laser line: 488 nm: 38 μm; 555 nm: 34 μm; and 639 nm: 39 μm. All images were acquired at 1024 × 1024 pixel resolution with bidirectional scanning.

*Clk856 ∩ fru P1>sm.GDP.Myc* and *Clk856 ∩ fru P1>TrpA1* Myc cell bodies were scored blinded, in each brain hemisphere for 0- to 24-hr adults and 4- to 7-day adults of both sexes (*Source data 7*). Data from both left and right hemispheres for each region within sex were pooled for analysis for each genotype and time point. Statistical testing within each quantified region was performed between the sexes within each genotype and time point using a Mann–Whitney *U*-test in GraphPad Prism (9.3.0).

## Subclustering male-specific/-biased *dsx*-expressing neuron clusters

All male specific or male biased that were enriched for *dsx* expression (*Source data 3*), clusters 21, 47, and 68 in the full data analysis were subset and re-analyzed. Only *dsx* expressing neurons in these clusters were maintained in this analysis (*dsx* expression >0 UMIs). Twenty-seven significant PCs were

used to perform new dimensionality reduction based on Jackstraw analysis. Cluster resolution was determined based on visual inspection of distinct UMAP populations being identified as individual clusters (resolution = 0.5). Differential expression was calculated between the clusters using the same *FindAllMarkers* criteria as above.

## Subclustering circadian neuron clusters

All clusters classified as 'circadian' in our full data set UMAP analysis were subset and PCA was performed only on these cells. The two significant PCs were used to perform new dimensionality reduction based on Jackstraw analysis. Cluster resolution was determined based on visual inspection of distinct UMAP populations being identified as individual clusters (resolution = 2.5). Differential expression was calculated between the clusters using the same *FindAllMarkers* criteria as above.

## Annotating mushroom body KC populations and subclustering

Cell clusters were annotated based on the presence genes that have been shown to be expressed in the mushroom body KCs and those specific to subpopulations. All of these genes were manually annotated for their presence (*Source data 3*), however, a cluster was only confidently annotated as a mushroom body KC subtype if at least two genes were enriched in expression in that cluster. For example, if a cluster had a marker gene with known general KC expression present (*ey* and/or *Dop1R2*) and additionally had *Fas2*, *sNPF*, and/or *trio* as a marker gene, the cluster was annotated as a KC subtype. These subtypes were determined based on the presence of *trio*, which shows high expression in only the γ KCs at 48-hr APF (*Awasaki et al., 2000*). αβ KCs were annotated based on marker genes for *ey* and/or *Dop1R2* with *sNPF* and/or *Fas2* in the absence of *trio*. Further, we also found clusters with *Fas2* and *sNPF* as marker genes with expression of *ey* or *Dop1R2* in the cluster, but not present as a marker gene. We also find one cluster per data set that has marker gene expression of *trio* in addition to one other KC-expressed gene, either *Fas2* or *Dop1R2*. Clusters with incomplete enriched marker gene expression but suggestive of being KC subtypes are annotated as lower confidence ab_KC clusters (indicated by *).

All high-confidence KC clusters in our full data set UMAP analysis were subset and PCA was performed only on these cells. The 21 first significant PCs were used to perform new dimensionality reduction based on Jackstraw analysis. Cluster resolution was determined based on visual inspection of distinct UMAP populations being identified as individual clusters (resolution = 0.5). Differential expression was calculated between the clusters using the same *FindAllMarkers* criteria as above.

## Locomotor activity assays and analysis

Wild-type CS, transgene controls without a *Gal4* driver (*w; UAS <stop < TrpA1^{Myc}; fru^{FLP}/+*), and experimental flies with expression restricted to *Clk856 ∩ fru P1* neurons (*w; UAS<stop<TrpA1^{Myc}/ Clk856-Gal4; fru^{FLP}/+*) were collected 0- to 6-hr post-eclosion and loaded into glass tubes that contained our standard laboratory food. Tubes containing the flies were loaded into DAM (Trikinetics, Waltham, MA). The light:dark (LD) assay was conducted at 25°C on a 12-hr:12-hr LD cycle and beam cross activity was recorded in one min. bins for 15 days. Incubator lights turned on at 8 am and off at 8 pm. Flies were aged to 5 days during the assay, therefore the first 5 days of data were removed from the analysis. This resulted in 10 days of data analyzed for all genotypes in the LD assay. In the dark:dark (DD) assay, flies were first entrained and aged at 25°C on a 12-hr:12-hr LD cycle for 7 days, allowing for 2 days of LD data for analysis. Next, we switched to a 12-hr:12-hr DD cycle for 10 days (constant darkness for 10 days). Beam cross activity for the DD assay was recorded in 1-min bins. Data were analyzed using ShinyR-DAM for both assays (*Cichewicz and Hirsh, 2018*). Dead flies were considered those with less than 50 beam cross events per day and were removed from the analyses, resulting in an *n* = 26–30 for all genotypes in LD and *n* = 23–30 for all genotypes in DD. ShinyR-DAM analysis of sleep was only performed on LD assay data. ShinyR-DAM measures sleep events using a 5-min sliding window, where 5 min of inactivity is considered a sleep event. Statistical tests were performed on ShinyR-DAM output data in R Studio and GraphPad Prism (9.3.0). One-way analysis of variance with Tukey-HSD post hoc testing was performed for comparisons across all genotypes. Comparisons between daytime and nighttime sleep were statistically compared using a Student's *t*-test for daytime vs. nighttime sleep. Circadian period analysis was performed in ShinyR-DAM using the DD assay data. The default ShinyR-DAM parameters were used as follows: Chi-Sq testing range of 18–30 hr, a Chi-Sq

period testing resolution of 0.2, and a rhythmicity threshold for filtering arhythmic individuals (Qp.act/Qp.sig) of 1. All ShinyR-DAM output data presented are provided in *Source data 7*.

## Acknowledgements

We thank the Florida State University College of Medicine Translation Core for running the sequencing core. Several stocks were obtained from the Bloomington *Drosophila* Stock Center (NIH P40OD018537). Several antibodies used in this study were obtained from the Developmental Studies Hybridoma Bank, created by the NICHD of the NIH and maintained at The University of Iowa, Department of Biology, Iowa City, IA 52242. We thank colleagues that generously provided reagents, including Drosophila strains and antibody resources.

## Additional information

### Funding

| Funder | Grant reference number | Author |
| --- | --- | --- |
| National Institute of General Medical Sciences | R01GM073039 | Colleen M Palmateer<br>Catherina Artikis<br>Savannah G Brovero<br>Benjamin Friedman<br>Alexis Gresham<br>Michelle N Arbeitman |
| National Institute of General Medical Sciences | R01GM116998 | Colleen M Palmateer<br>Catherina Artikis<br>Savannah G Brovero<br>Benjamin Friedman<br>Alexis Gresham<br>Michelle N Arbeitman |
| National Institute of General Medical Sciences | 1R35GM148205 | Michelle N Arbeitman |

The funders had no role in study design, data collection, and interpretation, or the decision to submit the work for publication.

### Author contributions

Colleen M Palmateer, Conceptualization, Data curation, Formal analysis, Validation, Investigation, Visualization, Writing - original draft, Writing - review and editing; Catherina Artikis, Benjamin Friedman, Alexis Gresham, Validation, Visualization; Savannah G Brovero, Methodology; Michelle N Arbeitman, Conceptualization, Data curation, Formal analysis, Supervision, Funding acquisition, Validation, Investigation, Writing - original draft, Project administration, Writing - review and editing

### Author ORCIDs

Colleen M Palmateer ⓘD http://orcid.org/0000-0002-7254-0829
Michelle N Arbeitman ⓘD http://orcid.org/0000-0002-2437-4352

### Decision letter and Author response

Decision letter https://doi.org/10.7554/eLife.78511.sa1
Author response https://doi.org/10.7554/eLife.78511.sa2

## Additional files

### Supplementary files

• MDAR checklist

• Source data 1. Sequencing metrics and barcode-rank plots for all replicates. Excel data table containing CellRanger barcode-rank plots and sequencing metrics for all replicates. After quality control filtering, the median number of genes per cell and median UMI counts per cell are presented.

- Source data 2. Distribution of cells per replicate and sex in Uniform Manifold Approximation and Projections (UMAPs) and select gene expression metrics. Excel data table containing cell number contributions of each individual replicate to clusters. This table contains the Seurat integration analysis. The table contains the number of cells and percent of total cells expressing *GFP*, *fru*, *roX1*, *roX2*, *elav*, *nSyb*, *noe*, *stg*, and genes indicative of fast-acting neurotransmitter (FAN) expression. The table includes DoubletFinder analyses.

- Source data 3. Marker genes and annotations. Excel data table containing marker gene lists and cluster annotations for all analyses. Statistical tests of marker gene list overlaps with lists of genes shown to be enriched or differentially expressed at 48-hr after puparium formation (APF) in previous *fru P1* Translating Ribosome Affinity Profiling (TRAP study *Palmateer et al., 2021*). This table also contains the Seurat integration analysis marker gene list.

- Source data 4. Gene ontology (GO) enrichment analyses. Excel data table containing GO enrichment results.

- Source data 5. Assigning sex bias to clusters in full and downsampled data sets. Excel data table containing cell number contributions by each sex per cluster. A scaling factor to normalize the number of male and female cells was applied to determine the sex bias in cell number. The table includes assessments of cluster data with random female cells removed, and a full Seurat re-analyses of data with downsampled female cell numbers. The table includes the cluster comparisons of downsampled analyses using clustifyr matching.

- Source data 6. Marker genes for subclustering analyses. Excel data table containing marker genes for *dsx*, Kenyon cell, and circadian clock neuron subclustering.

- Source data 7. *Clk856* ∩ *fru P1* cell body counts in the adult brain and *Drosophila* activity monitor (DAM) data. Excel data table containing the number of *Clk856* ∩ *fru P1* cell bodies per region in adult male and female brains. The DAM data underlying graphs and plots in *Figure 7—figure supplement 2*.

- Source data 8. *nAChR* expression. Excel data table containing the proportion of total neurons expressing *nAChR*s in all data sets.

- Source data 9. Sex differences in gene expression within cluster.
 Excel data table containing the differentially expressed genes between the sexes per cluster.

### Data availability

The Gene Expression Omnibus accession number for all scRNA-seq the data is https://www.ncbi.nlm.nih.gov/geo/query/acc.cgi?acc=GSE160370. Code used for the analyses presented can be found at: https://github.com/cpalmateer/fruP1SC_48hrAPF (copy archived at swh:1:rev:e82363ab1c3ef08dc4c067b155b28607bde3a861).

The following dataset was generated:

| Author(s) | Year | Dataset title | Dataset URL | Database and Identifier |
|---|---|---|---|---|
| Palmateer C, Arbeitman M | 2023 | A single-cell transcriptomic atlas of fruitless-expressing neurons in the *Drosophila* pupal central nervous system | https://www.ncbi.nlm.nih.gov/geo/query/acc.cgi?acc=GSE160370 | NCBI Gene Expression Omnibus, GSE160370 |

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
