## [Editor Report]

This study presents a valuable single-cell sequencing dataset of fruitless-expressing neurons in the male and female *Drosophila* nervous system. The quality data and convincing analyses allowed the authors to conclude that most neuronal types are present in both *Drosophila* sexes, suggesting that sex-specific versions of the transcription factor Fruitless can modify neural function in a sex-specific way without completely altering core neural identity. This work will be of interest to developmental biologists and neuroscientists with a focus on sex-specific differences.

---

## [Decision Letter]

**Decision letter after peer review:**

Thank you for submitting your article "Single-cell transcriptome profiles of *Drosophila* fruitless-expressing neurons from both sexes" for consideration by *eLife*. Your article has been reviewed by 3 peer reviewers, one of whom is a member of our Board of Reviewing Editors, and the evaluation has been overseen by Claude Desplan as the Senior Editor. The reviewers have opted to remain anonymous.

Essential revisions:

1) The reviewers have raised concerns regarding the "sex-specific" versus "shared but sexually dimorphic" cell types. They recommend a careful re-analysis of the data to address this exact question. In particular, they recommend generating a single atlas with both male and female cells under different integration or merging conditions. This would include:

a) a careful parameter selection, such as resolution, that will allow for evaluation of subclusters. They also recommend a comparison between integration and merging of male and female datasets

b) doublet elimination using appropriate algorithms (e.g. DoubletFinder or other)

c) elimination of cell cycle influence

After generating such an atlas, an experimental validation of differences in cell number or sex-specific markers in a few well-known clusters would be necessary to address the question of sex-specificity of clusters.

2) The reviewers have noticed that, in multiple cases, there are discrepancies between the paper's conclusions and established knowledge or previous datasets. There has to be a clear discussion of the reasons or experimental resolution.

3) The paper is long and detailed and the reviewers feel that there has to be a Conclusion Section that summarizes the most interesting observations and makes the paper accessible to a broader audience.

4) In many cases, the reviewers noticed images that were of low quality or resolution. This also has to be addressed.

*Reviewer #2 (Recommendations for the authors):*

I have some concerns about the way neuron number was normalized per cluster per sex. I do not have a better way to do this analysis but would instead like to see a specific example cell type/cluster pinned down via counting the number of real neurons. This could perhaps be done using a known cell type and antibody stains for a marker protein or a GAL4-reporter combination, to determine whether the difference in neural number between sexes for a particular cluster is real. This approach was used to evaluate female-specific cluster 107, but not to validate any differences in neuron number in a cluster that is present in both sexes. The conclusion that so many clusters contain different numbers of neurons between the sexes is an important finding and additional "ground truthing" would help support the authors conclusions.

Many of the figure panels that contain antibody stains are not of sufficiently high quality to really see what is going on, even in the separate higher res version of the Figure. This is true in several places. For example, for Figure 7E, please consider cropping or including a second set of panels with insets showing the region with expression. Many brains, exe. all of Figure 10, are not high enough resolution to see what the tiny arrows are pointing at.

I was not able to evaluate a section of Figure 5: there are no panels after 5S. The text refers to experiments in 5T and 5U that are not shown.

"However, we do not find broad expression of ChAT at 48hr 478 APF, which is not consistent with our single cell data that shows extensive expression of ChAT." I was hoping for some discussion of this discrepancy, or an experimental resolution. mRNA expression does not always match protein (or reporter protein) expression. This can be especially true for terminal genes like neurotransmitters or Rhodopsins, where there are cases of high mRNA expression long before protein expression. I would love to see HCR in situs for ChAT vs. an antibody stain or protein reporter. This examination could potentially be restricted to a specific cell type for clarity. Comparing the two would be an interesting addition, especially given such a distinct difference that is otherwise unexplained in the manuscript. At the very least, this result should be discussed.

I'm not entirely sure I understand the link between the section on circadian rhythm and the rest of the paper. The authors were able to identify and label DN1p in the scSeq data, but unless I misunderstand, do not focus on DN1p in the experiments on sexually dimorphic roles of activation of a subset of circadian rhythm neurons.

*Reviewer #3 (Recommendations for the authors):*

In addition, we have some comments and questions about the data analysis that could impact some of these "big picture" conclusions.

A. One of the big findings the authors report from their single cell analysis is the apparent detection of male-specific and female-specific neuron populations, a finding that would be of wide interest to those interested in sex differences. However, we feel that support from a further analysis would help make this result more convincing. The authors currently infer sex-specificity in some of their clusters based on the absence of cells from one sex after merging the datasets. This raises an obvious question – to what extent does this inference depend on clustering parameters, and how do you know what the optimal clustering strategy is? For example, in the combined-sex analysis, clusters 12/5/16 are considered separate clusters (Figure 1), and cluster 12 is designated as strongly male-biased and clusters 5 and 16 as strongly female-biased (Figure 2B). On the other hand, clusters 3 and tachykinin-1 are each considered a single cluster (Figure 1) and are considered sexually monomorphic (Figure 2B), but a look at Figure 2A shows that each of these clusters has a male half and a female half. Is there an objective reason for deciding when we are looking at a single monomorphic cluster with sex-biased gene expression, and when we are looking at a closely related pair of sex-limited clusters? The authors describe their statistical approach to determine the number of clusters; but that is going to be affected by sequencing depth, variation in staging, batch effects, cell cycle differences, etc. What is the most biologically significant clustering strategy, and how do you determine that? How do you decide, as a neurobiologist, whether you are under- or over-clustering your data, and when you hit that Goldilocks spot where your clusters are most likely to correspond to biologically meaningful cell types? These issues are worth discussing. (On a technical note, please use different colors in Figure 2B – the color for strong male bias looks almost the same as the color from strong female biased, which confused us until we figured that out).

There is a technical approach that could help with this. Merging combines the raw count matrices from different datasets, but an alternative is to integrate them (implemented through Seurat), which uses common anchors between cells across datasets to promote the identification of shared cell types. We think it would be worth seeing if this sex-specificity holds after integration. If it does, then that's two independent pipelines that identify sex-specific populations. We think that integrating, rather than merging, might help tighten up some of the sex difference analyses too. Looking at Figure 2A, it's clear that many of the clusters don't merge particularly well. The authors use this as evidence of variation in the extent of sexual dimorphism between populations (as labelled in Figure 2B). But we worry that this creates fuzzy boundaries between male and female populations of what may be the same neuron type. And this fuzziness could affect where the cluster boundaries are called. Comparing the bottom part of Figure 1F and 2A, we can see a cluster 6 that is called as a single cluster despite clearly separating into male and female populations in Figure 2A. This contrasts with Octopamine_1, which seems to be largely/entirely female-biased, while Octopamine_2 is male-biased. There are other examples like that. We wonder whether integration (rather than merging) will do a better job of identifying homologous cell types. The authors could then ask what is differentially expressed between male and female cell-types within a cluster.

B. A general conclusion the authors reach is that male and female fru+ neurons 'share common gene expression repertoires with sex-specific information overlaid on these core patterns' (lines 145-149; see also 1057). This conclusion rests on their observation that 'nearly all clusters are comprised of male and female neurons'. On the one hand, this is what UMAP does – it clusters cells by expression similarities. How do we know that a cluster containing both male and female cells on a UMAP actually corresponds to the same population in male and female brains? On the other hand, taken alongside the small number of 'sex-specific' clusters they resolve, it seems that the authors are arguing that the vast majority of the neurons are shared between the sexes, but that these shared types generally show sexual dimorphism in their expression. This is an interesting result, but it rests exclusively on the distribution of cells in UMAP space and the cluster identification boundaries. For the reasons outlined above, these features of the UMAP/clustering are heavily dependent on the input parameters. Without some sort of orthogonal validation, it's hard to know how robust this result is.

C. A potential problem with the sex difference analyses, which the authors themselves recognize (line 249), is how to exclude the possibility that much of the difference is driven by sex differences in development time and/or cell cycle phase. The data discussed on lines 970-974 seems to confirm that the differences in developmental timing could be a significant contributor to male- and female-specific gene expression. The authors show some of their stainings at two different time points, which helps for those specific genes, but it's a potential issue that limits the interpretation of the transcriptomic data itself and the authors should probably highlight it. On the analysis side, there's not much that can be done about development time, but have the authors tried to account for cell cycle?

D. Organizing the section on transcription factors (starting at line 1023) by TF superfamily may not be the most informative approach, since the type of DNA binding domain is not directly related to developmental function. It may be better to organize this section by the degree of specificity: start with which TFs, regardless of their family, are the most specific to candidate cell types – and which TFs are specific to particular clusters that were singled out for detailed analysis in previous sections (e.g. KC cells, clock neurons, which TFs show correlation with particular neurotransmitters, etc). Better biological insights are likely to be obtained this way. Figure 12 and its supplements are near-impossible to take in. It would be better to replace the main-text figure with examples of highly specific TFs mapped on UMAPs, perhaps in relation to cluster annotation developed previously in this paper and move the big grid to the supplement. Showing a separate panel with 5-10 most sex-biased TFs for each sex would be more informative than annotating them on the large cluster*TF grid.

[Editors' note: further revisions were suggested prior to acceptance, as described below.]

Thank you for resubmitting your work entitled "Single-cell transcriptome profiles of *Drosophila* fruitless-expressing neurons from both sexes" for further consideration by *eLife*. Your revised article has been evaluated by Claude Desplan (Senior Editor) and a Reviewing Editor.

The manuscript has been improved but there are some remaining issues that need to be addressed, as outlined below:

When comparing the merged and integrated dataset, the authors argue that they are highly concordant. Nevertheless, based on Figure 1 and Figure Supplement 5C-D, it is pretty obvious that integration worked better since there are a number of library-specific groups of datapoints in the merged UMAP. It would make it easier to visualise the integration of the libraries, even if the points are shuffled. In R, this is implemented in DimPlot of Seurat, by the option shuffle=TRUE.

We appreciate that the authors do not want to repeat the analysis, but they should definitely compare how the two approaches affect their analysis and not only rely on the visual (which, furthermore, is not convincing).

This can indeed lead to serious misinterpretations of the data when clusters that are library-specific are simply the product of batch effect. For example, the authors identify more mushroom body subtypes (9 or 13 depending on the clustering) than what has been reported in the literature (7). Are these clusters reliable? Or do they result from batch effect? Comparing Figure 1, Figure Supplement 5C-D, and Figure 4A, it seems very plausible that cluster γ_KCs_3 is library-specific. This problem might also occur in other neuronal structures, where the cell type composition is not as well known as the mushroom bodies.

---

## [Author Response]

Essential revisions:1) The reviewers have raised concerns regarding the "sex-specific" versus "shared but sexually dimorphic" cell types. They recommend a careful re-analysis of the data to address this exact question. In particular, they recommend generating a single atlas with both male and female cells under different integration or merging conditions. This would include:a) a careful parameter selection, such as resolution, that will allow for evaluation of subclusters. They also recommend a comparison between integration and merging of male and female datasets

We performed an additional analysis where we integrated the data sets using the Seurat FindIntergrationAnchors function and the IntegrateData functions, as suggested. The authors did not do an integrated analysis initially given we did not see large batch effects across the replicates, and all the samples were generated using the same 10x Genomics platform and sequencing methodologies. The Integrated results are highly consistent with our original merged data analyses. We have added a new figure (Figure 1 —figure supplement 5), to provide a side-by-side visual of these UMAP plots for the reader. The number of cells contributed from each sex in this integration analysis has been added to Source Data 2 and reveals highly similar trends to that of our presented analysis. The integrated analysis also reveals overlap between the neurons from males and females across clusters, few sex-specific clusters, and clusters with sex-biases in cell number. Further, when we examine the sex-specific clusters derived from the integrated analysis, we find that the male-specific cluster has *dsx* as a top marker gene, serving as a ground truth in both analyses, given there are known small populations of male-specific neurons that express both *dsx* and *fru* (PMID: 16950103, PMID: 32314964, PMID: 12745633, PMID: 17716899, PMID: 18599032)*.* Given the methodologies produced similar results, and the merged analyses was appropriate given there were no large batch effects or changes in methodologies in making the libraries, the authors continue to move forward with the original merged data presented and the substantial analyses on that data set. The integrated analyses are available in the supplemental materials and uploaded to GEO. A summary of this analysis and a figure (Figure 1—figure supplement 5) have been added to the manuscript, lines 221-225. The methodology for this analysis has also been added to the Materials and methods section, lines 1176-1187.

To address the concern about parameter selection, we clarify our approaches and provide in vivo evidence that the resolution chosen was appropriate. In the analyses presented, we used the JackStraw statistical approach to decide on the number of principle components (p <0.05). We next compared several resolutions and examined the resulting trees using Clustree, to visually inspect the additional clusters produced at each resolution. We decided to use the male-specific *dsx* cluster as a ground truth, as we know these neurons exist biologically and are a small population in vivo. Subclustering of the full data set did not reveal additional male-specific *dsx*-expressing clusters, giving us further confidence in the resolution chosen. We acknowledge that higher resolution clustering will result in more clusters, though we did not find substantial additional cluster separations with more resolution: the larger clusters were no longer sub-dividing by increasing the resolution. The data set presented here has more clusters than the published full brain data set, further suggesting we used resolution aligned with what is expected for a smaller neuronal data set. Finally, it does not appear that we used too high a resolution, given our biological validation of two clusters comprised of few neurons: the female-specific *CCHa2* cluster, and the sex-shared *DIP-iota*-expressing cluster a new figure, Figure 1 —figure supplement 8 (*DIP-iota* described in lines 1169-1175). The in vivo results showed there were few neurons that are *CCHa2* ∩ *fru P1* or *DIP-iota* ∩ *fru P1*, consistent with scRNA-seq clustering data. Further, we also present in vivo validation of large clusters by examining genes that are broadly expressed in large clusters, such as the *nAchR*-encoding genes (Figure 9, lines 894-912).

b) doublet elimination using appropriate algorithms (e.g. DoubletFinder or other).

We applied DoubletFinder to further evaluate our filtering criteria. DoubletFinder does identify additional potential doublets that were not filtered using our initial criteria. We have added the DoubletFinder results to Source Data 2 and add this point to our discussion (lines 1099-1102). The potential additional doublets are not concentrated in any one cluster or replicate, indicating that doublets are not a large confound across the cluster analyses presented and are not the only cells in any cluster. As with any genomic analyses, researchers need to establish their criteria using the tools and genome releases available at the time and move forward with their extensive annotation and validations. Additionally, our initial filtering criteria to remove potential doublets were rigorous. We eliminated outliers by maintaining cells in our analysis that met the following criteria: >5% mitochondrial transcripts (dead/dying/stressed cells), <200 genes (empty droplets), those expressing more than 4,000 features (genes) and/or possessing more than 20,000 UMIs (potential doublets or triplets) (Lines 172-177 and 1149-1157). These parameters are similar to those used in *Drosophila* cell atlases (PMID: 32314735, PMID: 29671739, PMID: 31746739, PMID: 36289550, PMID: 31915294, PMID: 32339165, PMID: 32396065).

c) elimination of cell cycle influence.

Animals used in this study were 48 hours after puparium formation (APF). At this time point, all neuroblasts divisions have ceased, except for four mushroom body neuroblasts (PMID: 1728583), so cell division is unlikely to be having a global impact on this data set. To further address this concern, we examined expression of *stg*, a cell-cycle gene with elevated expression in mitotic neuroblasts. *stg* expression was only found in 0.3% of neurons in our data set (lines 193-197 and Source data 2). Furthermore, we observe that a large percentage of neurons express the post-mitotic genes *elav* (77%) and *nsyb* (93%). These results suggest that there is minimal cell-cycle influence on gene expression in our data set.

After generating such an atlas, an experimental validation of differences in cell number or sex-specific markers in a few well-known clusters would be necessary to address the question of sex-specificity of clusters.

In this study we find two high confidence sex-specific clusters. These clusters were identified after both downsampling and data normalization approaches. One is the known male-specific *dsx*-expressing cluster (PMID: 16950103, PMID: 32314964, PMID: 12745633, PMID: 17716899, PMID: 18599032) and the other is the newly identified female-specific *CCHa2*-expressing cluster validated in this study. Therefore, there is in vivo experimental evidence for these small sex-specific clusters.

One limitation of scRNA-seq studies is that the workflows may be performed with neurons/cells from several animals and the workflows may results in loss of neurons/cells. Therefore, the number of neurons/cells present in any cluster may differ from what is present in the animal. However, even given this caveat we observed a high correlation in the number of neurons in each cluster between the replicates from each sex (male replicates r=0.93; female replicates r=0.88)(Source data 2). This indicates that there is limited technical variation in terms of the cells captured in each replicate, providing confidence in the comparisons made based on cell numbers.

As further validation that cluster size reflects in vivo biology, we show that clusters comprised of few neurons have correspondingly low numbers in the animal (*CCHa2*, *DIP-iota,* and circadian neurons*,* Figure 2 and Figure 1 —figure supplement 8, Figure 7 and Figure 7 —figure supplement 1) and clusters comprised of many neurons have a high number in animals (several *nAchR* subunit genes; Figure 9).

2) The reviewers have noticed that, in multiple cases, there are discrepancies between the paper's conclusions and established knowledge or previous datasets. There has to be a clear discussion of the reasons or experimental resolution.

We address all the reviewer comments. We have also added commentary about limitations of our study to address additional concerns.

3) The paper is long and detailed and the reviewers feel that there has to be a Conclusion Section that summarizes the most interesting observations and makes the paper accessible to a broader audience.

We have added a more comprehensive conclusions section with a summary of our results.

4) In many cases, the reviewers noticed images that were of low quality or resolution. This also has to be addressed.

We appreciate this being pointed out. Higher resolution figures have been provided or will be provided upon publication. We are planning on submitting for the reviewers high quality images using the Zipped related manuscript file to avoid issue with individual file sizes, as needed.

Reviewer #2 (Recommendations for the authors):I have some concerns about the way neuron number was normalized per cluster per sex. I do not have a better way to do this analysis but would instead like to see a specific example cell type/cluster pinned down via counting the number of real neurons. This could perhaps be done using a known cell type and antibody stains for a marker protein or a GAL4-reporter combination, to determine whether the difference in neural number between sexes for a particular cluster is real. This approach was used to evaluate female-specific cluster 107, but not to validate any differences in neuron number in a cluster that is present in both sexes. The conclusion that so many clusters contain different numbers of neurons between the sexes is an important finding and additional "ground truthing" would help support the authors conclusions.

We have added biological validation that shows the number of neurons per cluster reflects biology. We now show the small sex-shared *DIP-iota*-expressing cluster (Cluster 57) reflects the in vivo observations of few neurons in the animal, with sex-differences in number (Figure 1 —figure supplement 8, lines 1169-1175). We show that one neuron is restricted to the TN1 segment of the VNC in males and two in females at 48hr APF. In addition, there are several neurons in the male abdominal ganglion portion of the VNC that are absent in females, which is reflected in the male-bias in cell number in the cluster. Cluster 57 in the full data set contains 91 males cells and 46 female cells (Source Data 5). Given that 20 animals were used per replicate, this also validates that we are capturing very small neuronal populations in our data. As noted above, the two sex-specific clusters (female-specific *CCHa2* and male-specific *dsx*) also show that the number of neurons found in a cluster reflects in vivo biology. Additionally, we note above that large clusters with genes expressed in large percentage of the neurons have broad expression in vivo (*nAchR* subunits). We also perform independent validation for the circadian neurons in both sexes using antibody staining and Gal4 tools (Figure 7 and Figure 7 – supplement 1).

Furthermore, the results presented are consistent with studies that have shown that there are differences in *fru P1* neuron number between the sexes (PMID: 15959468, PMID: 15935765, PMID: 20832311, PMID: 20832315). Several studies have also shown sex differences in neuron number in *fru P1* neuron sub-populations defined by immunostaining with antibodies and Gal4 driver intersectional strategies (PMID: 20832311, PMID: 20832315, PMID: 18786359, PMID: 16281036).

Additionally, we provide strong evidence that the number of cells per cluster is unlikely due to batch effects, which would reflect technical issues, given the high correlation of numbers of cells per cluster across the replicates (male replicates r=0.93; female replicates r=0.88) (Source data 2).

Many of the figure panels that contain antibody stains are not of sufficiently high quality to really see what is going on, even in the separate higher res version of the Figure. This is true in several places. For example, for Figure 7E, please consider cropping or including a second set of panels with insets showing the region with expression. Many brains, exe. all of Figure 10, are not high enough resolution to see what the tiny arrows are pointing at.

We appreciate this feedback and have made sure to provide higher resolution figures to address this issue. We have asked to provide higher quality images for review through the submission system and will be certain that these are used for publication.

I was not able to evaluate a section of Figure 5: there are no panels after 5S. The text refers to experiments in 5T and 5U that are not shown.

We apologize for our error in these figure callouts, this has now been resolved.

"However, we do not find broad expression of ChAT at 48hr 478 APF, which is not consistent with our single cell data that shows extensive expression of ChAT." I was hoping for some discussion of this discrepancy, or an experimental resolution. mRNA expression does not always match protein (or reporter protein) expression. This can be especially true for terminal genes like neurotransmitters or Rhodopsins, where there are cases of high mRNA expression long before protein expression. I would love to see HCR in situs for ChAT vs. an antibody stain or protein reporter. This examination could potentially be restricted to a specific cell type for clarity. Comparing the two would be an interesting addition, especially given such a distinct difference that is otherwise unexplained in the manuscript. At the very least, this result should be discussed.

We acknowledge that this is a limitation of our study and now provide discussion of this result. HCR analyses will take several months to years of work with multiple HCR probes. This is a large undertaking, for a manuscript that is already lengthy. Follow-up with such an approach would be very interesting future work that we now discuss in our conclusions section.

I'm not entirely sure I understand the link between the section on circadian rhythm and the rest of the paper. The authors were able to identify and label DN1p in the scSeq data, but unless I misunderstand, do not focus on DN1p in the experiments on sexually dimorphic roles of activation of a subset of circadian rhythm neurons.

Here, we have implemented existing Gal4 tools to provide a more in-depth biological validation of the scRNA-seq dataset. We are capturing these neurons in our cell atlas at 48hr APF and sought to assess their function in behaving adult flies, when locomotor behaviors can be examined in a circadian context. Given this comment, we have modified and shortened this section to highlight the major findings. Based on the immunostaining staining results, the in vivo behavioral studies performed with the TrpA1 transgene expression modify activity of DN1, DN3, and LNds. We hope we understood the question being asked.

Reviewer #3 (Recommendations for the authors):In addition, we have some comments and questions about the data analysis that could impact some of these "big picture" conclusions.A. One of the big findings the authors report from their single cell analysis is the apparent detection of male-specific and female-specific neuron populations, a finding that would be of wide interest to those interested in sex differences. However, we feel that support from a further analysis would help make this result more convincing. The authors currently infer sex-specificity in some of their clusters based on the absence of cells from one sex after merging the datasets. This raises an obvious question – to what extent does this inference depend on clustering parameters, and how do you know what the optimal clustering strategy is? For example, in the combined-sex analysis, clusters 12/5/16 are considered separate clusters (Figure 1), and cluster 12 is designated as strongly male-biased and clusters 5 and 16 as strongly female-biased (Figure 2B). On the other hand, clusters 3 and tachykinin-1 are each considered a single cluster (Figure 1) and are considered sexually monomorphic (Figure 2B), but a look at Figure 2A shows that each of these clusters has a male half and a female half. Is there an objective reason for deciding when we are looking at a single monomorphic cluster with sex-biased gene expression, and when we are looking at a closely related pair of sex-limited clusters? The authors describe their statistical approach to determine the number of clusters; but that is going to be affected by sequencing depth, variation in staging, batch effects, cell cycle differences, etc. What is the most biologically significant clustering strategy, and how do you determine that? How do you decide, as a neurobiologist, whether you are under- or over-clustering your data, and when you hit that Goldilocks spot where your clusters are most likely to correspond to biologically meaningful cell types? These issues are worth discussing. (On a technical note, please use different colors in Figure 2B – the color for strong male bias looks almost the same as the color from strong female biased, which confused us until we figured that out).There is a technical approach that could help with this. Merging combines the raw count matrices from different datasets, but an alternative is to integrate them (implemented through Seurat), which uses common anchors between cells across datasets to promote the identification of shared cell types. We think it would be worth seeing if this sex-specificity holds after integration. If it does, then that's two independent pipelines that identify sex-specific populations. We think that integrating, rather than merging, might help tighten up some of the sex difference analyses too. Looking at Figure 2A, it's clear that many of the clusters don't merge particularly well. The authors use this as evidence of variation in the extent of sexual dimorphism between populations (as labelled in Figure 2B). But we worry that this creates fuzzy boundaries between male and female populations of what may be the same neuron type. And this fuzziness could affect where the cluster boundaries are called. Comparing the bottom part of Figure 1F and 2A, we can see a cluster 6 that is called as a single cluster despite clearly separating into male and female populations in Figure 2A. This contrasts with Octopamine_1, which seems to be largely/entirely female-biased, while Octopamine_2 is male-biased. There are other examples like that. We wonder whether integration (rather than merging) will do a better job of identifying homologous cell types. The authors could then ask what is differentially expressed between male and female cell-types within a cluster.

We address our clustering and analysis parameters in detail above. Based on the reviewers’ suggestions we used the integration functions in Seurat. We agree that there are challenges in discerning between what could be considered single monomorphic clusters with sex-biased gene expression and closely related pairs of sex-specific clusters. Given this, we felt it best to assign a cluster resolution that revealed known sex-specific populations of *fru P1*-expressing neurons and stopped at a resolution where we no longer saw large clusters splitting into smaller clusters, using Clustree (see above). We acknowledge the additional potential limitations reviewer 3 raises in the conclusions section of the manuscript.

The colors in Figure 2B and in Figure 2 supplement 1 have been changed.

B. A general conclusion the authors reach is that male and female fru+ neurons 'share common gene expression repertoires with sex-specific information overlaid on these core patterns' (lines 145-149; see also 1057). This conclusion rests on their observation that 'nearly all clusters are comprised of male and female neurons'. On the one hand, this is what UMAP does – it clusters cells by expression similarities. How do we know that a cluster containing both male and female cells on a UMAP actually corresponds to the same population in male and female brains? On the other hand, taken alongside the small number of 'sex-specific' clusters they resolve, it seems that the authors are arguing that the vast majority of the neurons are shared between the sexes, but that these shared types generally show sexual dimorphism in their expression. This is an interesting result, but it rests exclusively on the distribution of cells in UMAP space and the cluster identification boundaries. For the reasons outlined above, these features of the UMAP/clustering are heavily dependent on the input parameters. Without some sort of orthogonal validation, it's hard to know how robust this result is.

The authors point to previous studies in the field to provide orthogonal validation. First, results from our laboratory using the cell-type-specific, RNA sequencing approach called Translating Ribosome affinity purification (TRAP) showed that there are a large number of genes that are expressed in *fru P1* neurons from both males and females (core genes), and a much smaller set of male- and female-biased genes. This was true across two independent studies, examining three developmental time points (PMID: 27247289, and PMID: 33901168). That there is a much larger core set of genes expressed in *fru P1* neurons, than those with sex-biased expression supports the idea of shared core expression networks, overlaid with sex-specific information. The TRAP results are completely independent of cluster-based tools and were performed using independent statistical methodologies.

Additional evidence that neurons are “shared” between the sexes comes from a cell lineage study that mapped *fru* neurons to their neuroblast lineage origins, with high resolution. This allowed detailed comparisons and showed they have shared origins (PMID: 27618265)

Furthermore, several studies from our lab and others in the field have shown that *fru P1* neurons identified using genetic intersectional approaches identify populations in both males and females with highly similar spatial patterns (PMID: 20832315, PMID: 33901168, PMID: 20832311, PMID: 33616528). The intersectional approaches rely on expression of individual genes that have overlapping expression with *fru P1*, so this also provides support to the idea that cells in the same spatial position have shared gene expression. The finding of homologously positioned neurons in males and females using intersectional approaches is true across a large number of independent genes.

Text has been added to the manuscript on Lines 261-276

C. A potential problem with the sex difference analyses, which the authors themselves recognize (line 249), is how to exclude the possibility that much of the difference is driven by sex differences in development time and/or cell cycle phase. The data discussed on lines 970-974 seems to confirm that the differences in developmental timing could be a significant contributor to male- and female-specific gene expression. The authors show some of their stainings at two different time points, which helps for those specific genes, but it's a potential issue that limits the interpretation of the transcriptomic data itself and the authors should probably highlight it. On the analysis side, there's not much that can be done about development time, but have the authors tried to account for cell cycle?

As the reviewer indicates, the authors note the potential impacts of sex-differences in developmental timing at this stage. We have provided evidence that the cell cycle phase is not a major source of variation in this data set (see above).

D. Organizing the section on transcription factors (starting at line 1023) by TF superfamily may not be the most informative approach, since the type of DNA binding domain is not directly related to developmental function. It may be better to organize this section by the degree of specificity: start with which TFs, regardless of their family, are the most specific to candidate cell types – and which TFs are specific to particular clusters that were singled out for detailed analysis in previous sections (e.g. KC cells, clock neurons, which TFs show correlation with particular neurotransmitters, etc). Better biological insights are likely to be obtained this way. Figure 12 and its supplements are near-impossible to take in. It would be better to replace the main-text figure with examples of highly specific TFs mapped on UMAPs, perhaps in relation to cluster annotation developed previously in this paper and move the big grid to the supplement. Showing a separate panel with 5-10 most sex-biased TFs for each sex would be more informative than annotating them on the large cluster*TF grid.

The reviewer makes a good point and modify the manuscript figure 12 and provide a more focused analysis of 6 genes (3 per sex). We provide the TF superfamily cluster*TF grid results in the supplemental materials, as a reader may find them informative for a particular question.

[Editors' note: further revisions were suggested prior to acceptance, as described below.]

The manuscript has been improved but there are some remaining issues that need to be addressed, as outlined below:When comparing the merged and integrated dataset, the authors argue that they are highly concordant. Nevertheless, based on Figure 1 and Figure Supplement 5C-D, it is pretty obvious that integration worked better since there are a number of library-specific groups of datapoints in the merged UMAP. It would make it easier to visualise the integration of the libraries, even if the points are shuffled. In R, this is implemented in DimPlot of Seurat, by the option shuffle=TRUE.We appreciate that the authors do not want to repeat the analysis, but they should definitely compare how the two approaches affect their analysis and not only rely on the visual (which, furthermore, is not convincing).This can indeed lead to serious misinterpretations of the data when clusters that are library-specific are simply the product of batch effect. For example, the authors identify more mushroom body subtypes (9 or 13 depending on the clustering) than what has been reported in the literature (7). Are these clusters reliable? Or do they result from batch effect? Comparing Figure 1, Figure Supplement 5C-D, and Figure 4A, it seems very plausible that cluster γ_KCs_3 is library-specific. This problem might also occur in other neuronal structures, where the cell type composition is not as well known as the mushroom bodies.

The authors recognize that the reviewer wants to ensure that the computational analyses are robust. We agree that batch correction methodologies are critical for analyzing data sets that exhibit large degrees of variation between libraries or use different scRNA-seq chemistry platforms. While batch corrections are important in these circumstances, the use of the integrated analysis approach needs to be weighed against the outcome that the approach limits detection of true biological differences due to the algorithmic method of converging the data sets. Integrated analyses may have the unintended consequence of limiting the detection of true biological differences between the sexes in *fru P1*-expressing neurons, the primary goal of this study.

At the start of our analyses, we examined if the data have batch effects by visual inspection and by comparing the number of cells contributed per replicate for each cluster (Source data 2). We find high concordance in terms of number of cells contributed to each cluster between the replicates, as detailed in our previous response. In the full data set, using the merged analysis approach, we do identify 9 clusters that exhibit replicate-specific data from either sex when we performed data merging (Source data 2). However, we still argue that our technical replicates exhibit only a small degree of batch effects, based on the criteria above and using the merged approach is superior for this study. As suggested, the authors have additionally performed integrated analyses and find that this analysis has 7 clusters that exhibit replicate-specific data from either sex (Source data 2). Therefore, with respect to potentially improving replicate-specific population of a cluster, the integrated analyses does not perform that differently. We have added these points to the manuscript (lines: 222-228; 1181-1198).

We proceeded with the merged analyses for the data presented in the paper to ensure that we are able to best address our biological question, even with the potential small batch effects. This is reasonable as the data from the replicates overlaps for the majority of the clusters, and the integrated analysis also has replicate-specific data. Furthermore, integrated analyses may obscure detecting true biological differences, limiting our study results. We provide additional data for our reader to interrogate the integrated analysis further, by including the marker gene lists for that analysis in this manuscript (Source data 3), though we argue the merged data set analyses are better overall. Given the recommendation to improve the visualization, the authors have changed the UMAP figures that show the replicates using the option “shuffle = TRUE” (Figure 1 —figure supplement 1I, Figure 1 —figure supplement 5C-D). We agree that this is a better method to visualize the data for each replicate.

We further address the issue that using the merged analyses may skew the mushroom body results. The mushroom body analysis is mentioned by the reviewers, so the authors visualized those clusters by replicate, as well as cross-checked with Source data 2. We conclude that γ_KCs_3 is not library-specific. It is largely female-biased but neurons are contributed by all replicates in the merged data set (Source data 2). In addition, this recommendation prompted us to reexamine the subclustering results for the mushroom body analyses. Again, we find that no clusters are library specific and have added this visualization for transparency (Figure 4 —figure supplement 1K). Based on this visual inspection, we do find that male-specific subcluster 4 does exhibit a slight batch effect between the libraries based on distance of cells within that cluster, but there are cells from each replicate in the cluster.